# Hippocampal-entorhinal cognitive maps and cortical motor system represent action plans and their outcomes

Irina Barnaveli [1] ✉, Simone Viganò [1,2], Daniel Reznik [1], Patrick Haggard [3] & Christian F. Doeller [1,4] ✉

Efficiently interacting with the environment requires weighing and selecting among multiple alternative actions based on their associated outcomes. However, the neural mechanisms underlying these processes are still debated. We show that forming relations between arbitrary action-outcome associations involve building a cognitive map. Using an immersive virtual reality paradigm, participants learned 2D abstract motor action-outcome associations and later compared action combinations while their brain activity was monitored with fMRI. We observe a hexadirectional modulation of the activity in entorhinal cortex while participants compared different action plans. Furthermore, hippocampal activity scales with the 2D similarity between outcomes of these action plans. Conversely, the supplementary motor area represents individual actions, showing a stronger response to overlapping action plans. Crucially, the connectivity between hippocampus and supplementary motor area is modulated by the similarity between the action plans, suggesting their complementary roles in action evaluation. These findings provide evidence for the role of cognitive maps in action selection, challenging classical models of memory taxonomy and its neural bases.

Effective goal-directed actions are a key feature of animal behaviour. Theories of anticipatory behaviour suggest that actions are represented in terms of their outcomes[1,2]. Action and outcome are often directly linked: for example, extending the ankle joint will accelerate the car, and bring me closer to my destination. However, humans (but also some other animals, see Passingham[3], Wise & Murray[4], and Yamazaki et al.[5]), have a remarkable ability to rapidly learn and then exploit arbitrary associations between actions and outcomes[6]: think, for example, of the different keypress actions required to get a subway ticket from the ticket machine. Although this may be difficult when you first arrive in a new city, it rapidly becomes fluent and automatic. Current computational models of goal-directed behaviour are based on predictive models that connect motor commands with their outcomes[7–9]. The range and diversity of action-outcome behaviour is best captured by a modular organisation, in which several independent controllers exist for different action types or 'motor primitives'[10–13]. While such computational models clearly and explicitly define how goal-directed actions might be controlled, they do not specify how the relations between different action-outcome modules are organised and structured in the brain[11]. How do we search for the correct action program among the many action plans stored in memory? How do we compare them and select the most appropriate one for the outcome we desire?

A system like a look-up table[14] relating actions to outcomes might offer a simple solution to these questions. However, the wide repertoire of human actions and action-outcome associations implies that serially searching the look-up table to retrieve candidate action-outcome relations, compare different alternatives, and then select the

[1]Department of Psychology, Max Planck Institute for Human Cognitive and Brain Sciences, Leipzig, Germany. [2]Center for Mind/Brain Sciences, University of Trento, Rovereto, Italy. [3]Institute of Cognitive Neuroscience, University College London, London, UK. [4]Kavli Institute for Systems Neuroscience, NTNU, Trondheim, Norway. ✉e-mail: barnaveli@cbs.mpg.de; doeller@cbs.mpg.de

correct one would be impractically slow[15]. Moreover, action-outcome relations vary with context[16,17] and across time[18,19]. This would require constantly updating the look-up table, which might be highly inefficient. Modular theories of action control propose a responsibility-weighting process that assigns responsibility to each motor primitive according to how well it would minimise the forward prediction error for the current desired outcome[10,11]. Importantly, the motor primitives are seen as discrete, independent modules. The only sense in which different modules are related to each other is that the outputs of multiple modular controllers can be optimally blended according to some cost function[12,13].

Here, we investigate a different alternative, namely that action-outcome associations are represented in an abstract map-like structure (a so-called 'cognitive map'). This approach centres around the representation of relations between multiple actions, in contrast to the emphasis in current motor control models on control processes required for any one action. Specific brain circuits support the representation of such relational knowledge. The hippocampal-entorhinal system has historically been associated with encoding of spatial information characterized by place-specific activity[20,21]. More recently, the same system is now also thought to underlie the construction of relational representations of more abstract, non-spatial forms of knowledge[22–25]. Several neuroimaging studies confirmed map-like representations of sensory and conceptual information in the hippocampal-entorhinal system[26–39].

In the current work, we reasoned that the same neural machinery could form the representational scaffolding for encoding the relations between action-outcome models in the brain, a domain not previously linked to cognitive mapping. We aimed to show that information about different possible actions, and the relations between them, might be acquired from an interactive experience with the environment. Crucially, the relations between actions could be abstracted from specific sensory and motor experiences and represented in the hippocampal-entorhinal system in a way that would allow selection between alternative actions based on their outcomes. We therefore formulated two specific research questions. First, we asked whether the relation-based representations in the hippocampal-entorhinal system[23], typically interpreted as signatures of 'cognitive mapping' in spatial memory[40], are also present when participants have to represent and evaluate arbitrary and discrete action-outcome associations. Second, we investigated whether the hippocampal-entorhinal system interacts with neocortical motor regions essential for planning the movements as participants mentally map and evaluate action-outcome associations. According to this view, hippocampal cognitive maps would directly guide our motor interactions with the world by linking perception to action. To address these questions, we combined functional magnetic resonance imaging (fMRI) with an immersive virtual reality game that involved selecting between alternative actions on the basis of multiple outcome dimensions. Subsequent testing aimed to probe whether participants had acquired a cognitive map of the relations between multiple action-outcome associations.

## Results

### Participants successfully learned to associate arbitrary actions to their outcomes in a virtual reality setting

Forty-six human participants completed a 3-day experiment, involving 2 days of training in immersive virtual reality (VR) and in front of a computer screen, followed by a day involving two fMRI scanning sessions (Fig. 1). During the first 2 days, participants were trained in VR to select and execute a range of action combinations defined by precisely changing the position of two virtual joysticks (Fig. 1A, B, C). The execution of these action combinations triggered the launching of a virtual ball towards participants multiple times (see Supplementary Video 1 and 2; see Methods). The relative distance from the participant at which the ball eventually landed, and the proportion of its approach

trajectory for which it remained visible, depended on how the participant acted on the two joysticks. The mapping between these ball parameters and the actions that moved the joysticks into different positions was non-linear and varied between participants. Specifically, participants learned to associate 5 arbitrary actions for each joystick with two outcome dimensions: the probability of the ball falling close enough to be caught ("probability of catching", Joystick 1) and the probability of the ball remaining fully visible throughout its trajectory ("probability of visibility", Joystick 2) (Fig. 1D, E; see Methods, Action-outcome space). These two outcome dimensions were specifically chosen because of their obvious relevance to the common visuomotor tasks of interception and catching[41–44]. The action-outcome relations allowed us to position each combination of action-outcome associations as a point in a two-dimensional abstract action-outcome space. Furthermore, we created 'landmarks' wthin this action-outcome space, analogous to landmark objects in navigation studies[45,46], so as to facilitate learning of the relations between multiple action-outcome associations. Each landmark position (i.e., action combination) was associated with a specific coloured ball, while all other positions in the action-outcome space were associated with a grey ball (Fig. 1E; see Methods).This design allowed us to test for the existence of specific behavioural and neural signatures characteristic of a 2D relational structure associating actions to outcomes across these two dimensions.

Participants' task in the VR during the two training days was to infer and learn the mapping between joystick actions and the outcomes of these actions by experiencing a set of discrete and non-continuous action-outcome pairings. They began with Guided Exploration of the action-outcome environment (Supplementary Video 1; see Methods), followed by the Goal-directed Action Task. This task tested participants' knowledge about relations between acquired action-outcome associations. Participants were asked to select and produce appropriate actions from different alternatives in order to achieve a desired outcome (Supplementary Fig. 1A; Supplementary Video 2; see Methods). We showed that participants successfully acquired knowledge about the links between actions and their outcomes, and could use this knowledge for action selection (Supplementary Fig. 1B; see Methods).

### Participants successfully encoded action-outcome relations in a two-dimensional mental space

After VR training, we tested participants with four additional tasks to determine whether they had merely learned a sparse set of individual action-outcome associations, or alternatively had formed a more abstract and more complete structure (cognitive map) encoding multiple action-outcome associations, and the relations between such associations. To do so, we used two relational Rating Tasks, and two pairwise Comparison Tasks (Fig. 1F, G, H). In the Rating Tasks, participants were shown pairs of action combinations (Rating Task 1) or coloured balls (Rating Task 2) (Fig. 1G; see Methods). Using a slider, participants estimated how similar they thought the presented pairs of stimuli were in terms of designated outcomes – the probabilities of catching the flying ball and its visibility during the entire flight. We found that the estimated subjective similarity scores of the sampled stimuli correlated significantly and positively with the distances between the actual positions of these stimuli in the abstract action-outcome space (Fig. 1E; Fig. 2A, C, see Methods). The results of the rating tasks suggested that participants had organised action-outcome associations into a relational mental structure in which similar associations lie closer together, i.e., a cognitive map.

Our analyses investigated how participants structured and used the knowledge about action-outcome associations in order to select appropriate actions to achieve expected outcomes. To answer this question, we conducted two additional tasks during training and later during the fMRI scanning sessions. Participants were again presented

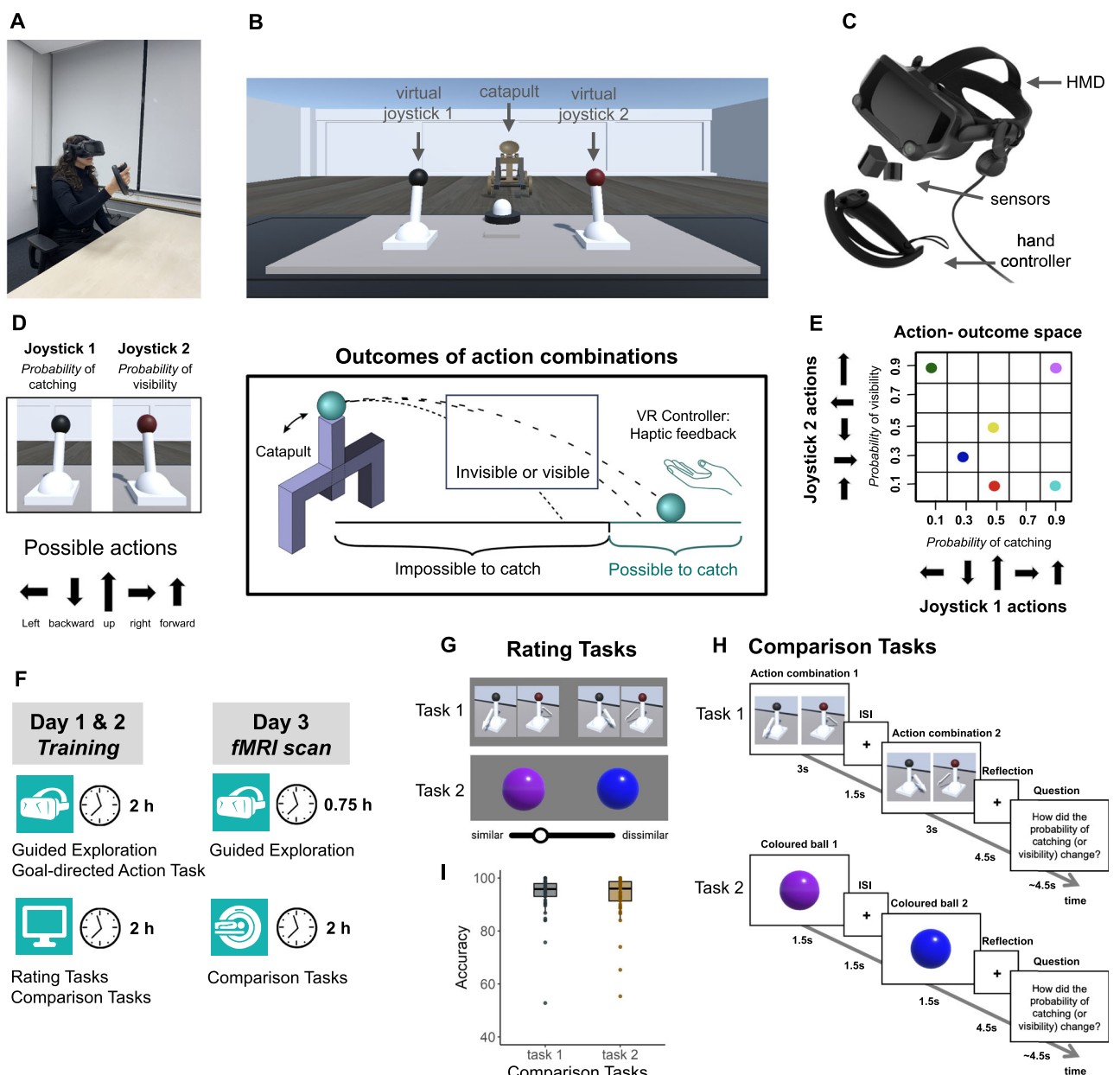

**Fig. 1 | Experimental design and cognitive tasks.** Participants were seated at a table in the experimental room (**A**), which corresponded to a virtual room (**B**) that was projected onto the head-mounted display (HMD) they were wearing (**C**). The table in the virtual room was equipped with two virtual joysticks and a pressable button in front of the joysticks. A hand controller device (**A, C**) was given to the participants' right hand, displaying the hand movements into the immersive VR environment. **D** Participants were trained to associate different joystick actions with two outcomes: the probability to catch the ball (outcome dimension 1), and the probability for the ball to remain visible throughout its entire flying trajectory (outcome dimension 2). The exact probabilities for each outcome dimension were never explicitly revealed to participants. The mapping of outcomes within each dimension to the corresponding joystick actions was randomized across participants. **E** Two-dimensional mapping of action-outcome associations. Each of the five actions of joystick 1 was associated with one of the five probabilities to catch the ball. Similarly, each of the five actions of joystick 2 was connected to one of the five probabilities of visibility of the ball. In addition, six different combinations of actions changed the colour of the ball, each producing a uniquely coloured ball, while all other combinations produced balls of the same grey colour. The six unique balls provided the orienting points in the action-outcome space and were therefore referred to as the 'landmark outcomes'. **F** Experimental session structure. The first

two days consisted of training in VR and in front of a computer screen. On the third day, each of the two scanning sessions was preceded by a short VR training to refresh the knowledge of action-outcome mapping. **G, H** After VR training on two days, participants' knowledge of action-outcome associations was tested in four tasks. In the two Rating Tasks (**G**), participants were simultaneously presented with pairs of action combinations (Rating Task 1) or coloured balls (Rating Task 2). Using the slider, they estimated how similar the shown stimuli are overall in terms of the associated outcomes, that is, the probabilities of ball catching and visibility. In the two Comparison Tasks (**H**), participants were sequentially presented with pairs of action combinations (Comparison Task 1) or coloured balls (Comparison Task 2). On each trial, participants were instructed to consider the associated outcomes of the stimuli of the pair during the reflection period. They then answered the subsequent question or statement on how the outcomes changed from one stimulus to the other. The two Comparison Tasks were performed in the scanner on the third day. **I** Behavioural performance in the scanner for the two Comparison Tasks. Dots represent data from $n$ = 46 participants. boxplots show median and upper/lower quartile with whiskers extending to the most extreme data point within 1.5 interquartile ranges above/below the quartiles; Source data are provided as a Source Data file.

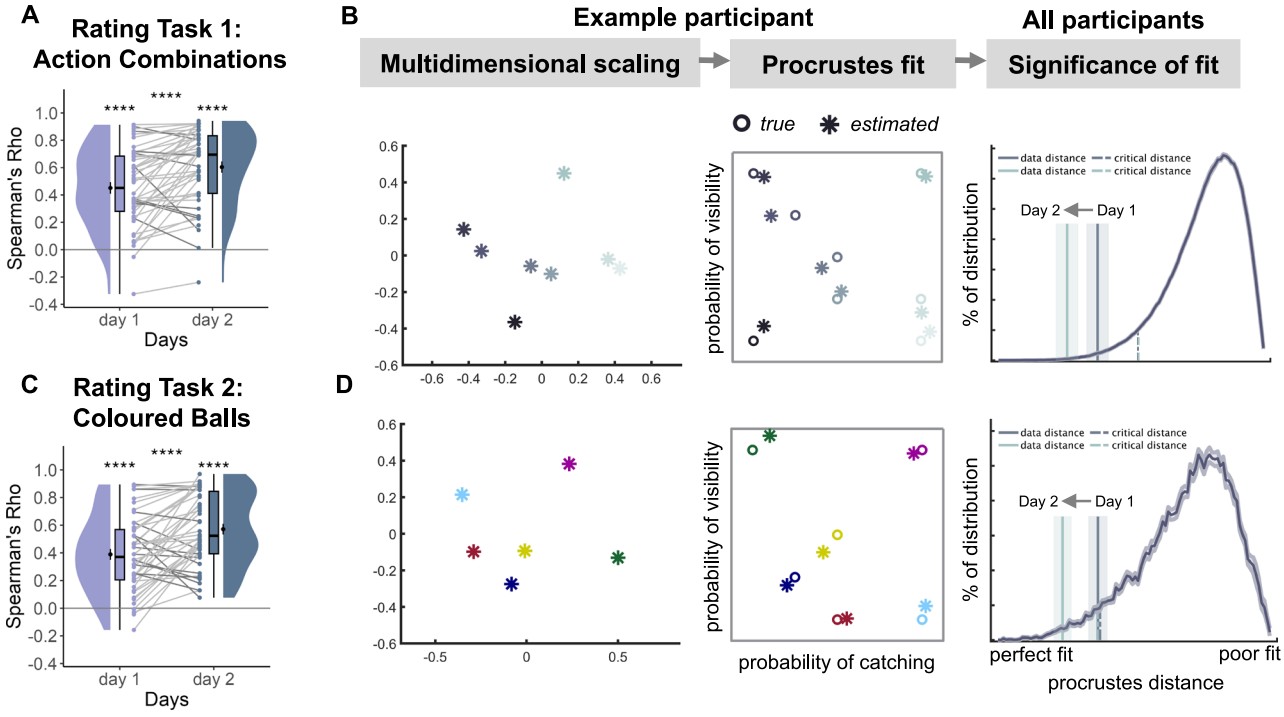

******p < 0.0001

**Fig. 2 | Reconstructing a map-like representation from the behavioural data.** The estimated pairwise similarity scores of action combinations (**A**) and coloured balls (**C**) from the Rating Tasks were correlated with the distances between their positions in the action-outcome space (Task 1: M = 0.451, SD = 0.288, Spearman's rho; t(45) = 10.63, p < 0.001, Cohen's d = 1.56, 95% CI= [0.888, 2.246]; Task 2: M = 0.389, SD = 0.27, Spearman's rho; t(45) = 9.79, p < 0.001, Cohen's d = 1.44, 95% CI= [0.776, 2.11]; two-sided t-test). Dots indicate data from individual participants. For both tasks, this correlation was higher on the second day compared to the first day (Task 1: M = 0.604, SD = 0.282, Spearman's rho; t(45) = 14.52, p < 0.001, Cohen's d = 2.14, 95% CI= [1.396, 2.886]; two-sided t-test; Across-day effect: t(45)= −4.28, p < 0.001, Cohen's d = −0.535, 95% CI= [−0.801, −0.269]; two-sided paired t-test; Task 2: M = 0.571, SD = 0.261, Spearman's rho; t(45) = 14.84, p < 0.001, Cohen's d = 2.18, 95% CI= [1.437, 2.939]; two-sided t-test; Across-day effect: t(45)= −4.82, p < 0.001, Cohen's d = −0.686, 95% CI= [−1.001, −0.372]; two-sided paired t-test). **B, D** To visually evaluate how participants' responses reflected the structure of the action-outcome space, we reconstructed their implied mental space from the pairwise similarity scores by applying multidimensional scaling (MDS) and obtained coordinate positions (indicated by snowflakes) that represent the structure of these similarity estimates. The resulting coordinates were mapped using Procrustes analysis to match the original positions (indicated by circles) of action combinations or coloured balls in the action-outcome space (see Bellmund et al.[107] and see Methods). The corresponding left and middle panels demonstrate the data

of an example participant from day 2. To evaluate the significance of mapping, we fitted MDS coordinates using the same approach to the sets of coordinates with shuffled stimulus – position assignments (curved solid lines). The corresponding right panels show the data for all participants. The vertical solid lines represent the mean Procrustes distances between original and reconstructed positions for each training day. The vertical dashed lines indicate mean critical Procrustes distances obtained from the shuffled distributions for each training day. The shaded areas indicate the standard error of the mean, calculated across participants. The results revealed that the mapping was indeed better than would be expected by chance for action combinations (Day 1: data distance: M = 0.368, SD = 0.273; critical distance: M = 0.526, SD = 0.009; Day 2: data distance: M = 0.257, SD = 0.261; critical distance: M = 0.528, SD = 0.011) as well as coloured balls (Day 1: data distance M = 0.371, SD = 0.219; critical distance: M = 0.382, SD = 0.028; Day 2: data distance: M = 0.247, SD = 0.205; critical distance: M = 0.372, SD = 0.04), pointing towards organization of multiple action-outcome associations in an abstract two-dimensional map. **A, C** Dots represent data from n = 46 participants. boxplots show median and upper/lower quartile with whiskers extending to the most extreme data point within 1.5 interquartile ranges above/below the quartiles; black circles with error bars correspond to mean ± SEM; distributions depict probability density functions of data points. Source data are provided as a Source Data file. ****p < 10^-4; Bonferroni corrected for tests on both days.

with pairs of action combinations (Comparison Task 1) or coloured balls corresponding to particular action outcomes (Comparison Task 2), but this time in a sequential order. Participants were instructed to compare the two stimuli based on their expected outcomes and respond to a question or a statement regarding how the associated outcomes differ between these stimuli (Fig. 1H; see Methods). Participants demonstrated high level of accuracy in their responses (Task 1: M = 93.7%, SD = 7.99%; Task 2: M = 92.9%, SD = 8.80%; Fig. 1I; Supplementary Fig. 2A), thus confirming that they had acquired and then used knowledge about the relation between items located in an action-outcome space. See Supplementary Fig. 2B for Spearman's correlation between the performance in the Comparison Tasks and the correspondence of similarity estimates from the Rating Tasks to the action-outcome space.

Collectively, these behavioural results indicate that over the two days of training in the virtual environment, participants successfully

learned to associate actions with overall task outcomes. Crucially, they additionally developed an internal abstract structure (cognitive map) that accurately represents how different actions relate to each other in terms of their outcomes.

## Entorhinal cortex represents the relational structure of different action plans

The behavioural performance observed during training days suggested a map-like organization of action-outcome relations, prompting us to investigate its neural underpinnings. To this end, we first focused on the first Comparison Task (see Fig. 1H), in which participants were instructed on each trial to recall the outcomes of a sequentially presented pair of action combinations and to answer a question or statement about the difference between the outcomes of the two action combinations. Since the action combinations could be conceptualized as particular positions in the action-outcome space, we

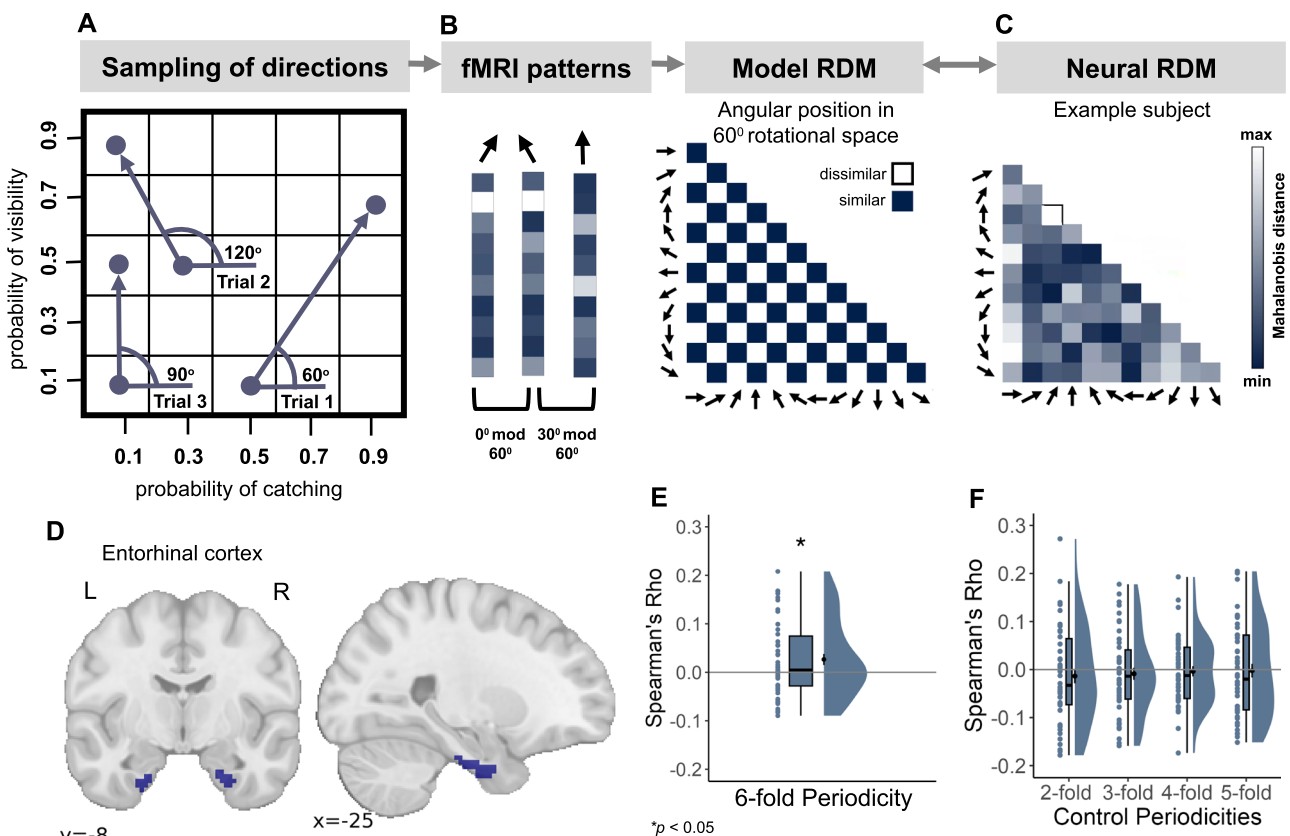

**Fig. 3 | Grid-like representation of the abstract action-outcome space.**
**A−C** Logic of analysis. The marked positions in the abstract action-outcome space correspond to different combinations of actions presented sequentially as pairs in the first Comparison Task. Each arrow corresponds to a direction in this abstract space, relating the first to the second action combination of a pair. The action-outcome space was sampled in 30° steps using 12 different directions in the first Comparison Task (**A**). To capture the directional information, we modelled the reflection phase of the task, where participants assessed the change in outcomes from one action combination to the other. Assuming grid-like activity in the entorhinal cortex (EC), we anticipated greater similarity between fMRI patterns for directions differing by multiples of 60° (with a remainder of 0° when dividing their angular difference by 60°) compared to directions whose angular difference results in a remainder of 30° (see Methods) (**B**). Following this logic, the 60° direction (Trial 1) should be more similar to the 120° direction (Trial 2) than to the 90° direction (Trial 3). The modelled representational dissimilarity matrix (RDM) was compared to the neural RDM using Spearman's correlation. Matrix for an example participant is shown in **C. D** Bilateral ROI mask for EC from the Julich brain atlas. **E** EC exhibited a significant 6-fold periodicity of pattern similarity at the group level (t(45) = 2.32, p = 0.011, Cohen's d = 0.34, 95% CI= [0.003, 0.049]**;** one-sided t-test). **F** There was no statistically significant effect in control periodicities (2-fold: t(45)= −0.9, p = 0.813, 95% CI= [−0.043, 0.016]; 3-fold: t(45)= −0.71, p = 0.762, 95% CI= [−0.033, 0.016]; 4-fold: t(45)= −0.33, p = 0.626, 95% CI= [−0.024, 0.017]; 5-fold: t(45)= −0.16, p = 0.563, 95% CI= [−0.031, 0.026]; one-sided t-test). Only four of these are shown in the figure, as the 7-fold and 8-fold periodicities are equivalent to the 5-fold and 4-fold periodicities due to the size of the action-outcome space. **E, F** Dots represent data from n = 46 participants. boxplots show median and upper/lower quartile with whiskers extending to the most extreme data point within 1.5 interquartile ranges above/below the quartiles; black circles with error bars correspond to mean ± SEM; distributions depict probability density functions of data points. Source data are provided as a Source Data file. *p < 0.05.

assumed that the relation between the first and the second action combination of a pair would correspond to a particular vector in that space (Fig. 3A). This allowed us to test for the existence of a specific neural representation associated with cognitive maps, namely a hexadirectional (6-fold) modulation of the activity in entorhinal cortex (EC), interpreted as an fMRI proxy measure of a putative population response of grid cells[47] (Fig. 3B−D). Following the Representational Similarity Analysis (RSA) approach[48] used in previous studies[34,35,37,49], we investigated the activity patterns evoked by pairs of action combinations. Specifically, we tested whether the similarity of activity patterns increased when the angles between the implicit directions of the different pairs within the putative action-outcome space representation were a multiple of 60° from each other, compared to when they were not (Fig. 3B; see Methods).

We found a significant positive correlation in the bilateral EC (t(45) = 2.32, p = 0.011, Cohen's d = 0.34, 95% CI= [0.003, 0.049]; one-sided t-test; Fig. 3E; see Methods for detailed information on the performed statistical tests, including the criteria for choosing between one- or two-sided tests). Analyzing the two hemispheres separately, we

observed a significant effect in the left (t(45) = 2.54, p = 0.011, Cohen's d = 0.37, 95% CI= [0.005, 0.05]; one- sided t-test), but not in the right hemisphere (t(45) = 0.75, p = 0.228, 95% CI= [−0.012, 0.02]; one-sided t-test; Supplementary Fig. 5A). We did not observe a statistically significant correlation between the pattern similarity in the EC and the alternative models assuming 2- to 8-fold periodicities (all p > 0.564; Fig. 3F). We further found no statistically significant relation between the similarity of the directions within the 2D abstract action-outcome space and their starting positions (t(45) = 1.02; p = 0.155, 95% CI = [−0.012, 0.039]; one-sided t-test), ending positions (t(45) = 0.36, p = 0.349, 95% CI = [−0.022, 0.032]; one-sided t-test), or a model combining both starting and ending positions in that space (t(45) = 0.34, p = 0.369, 95% CI = [−0.029, 0.042]; one-sided t-test; see Methods for details). Moreover, the effect could not be explained by the mean distance between starting and ending positions of each direction in the action-outcome space, as the mean was controlled to be similar for all directions (see Methods and Supplementary Fig. 3). The effect was confirmed using a cross-validated RSA (see Methods and Supplementary Fig. 4). Therefore, we concluded that the

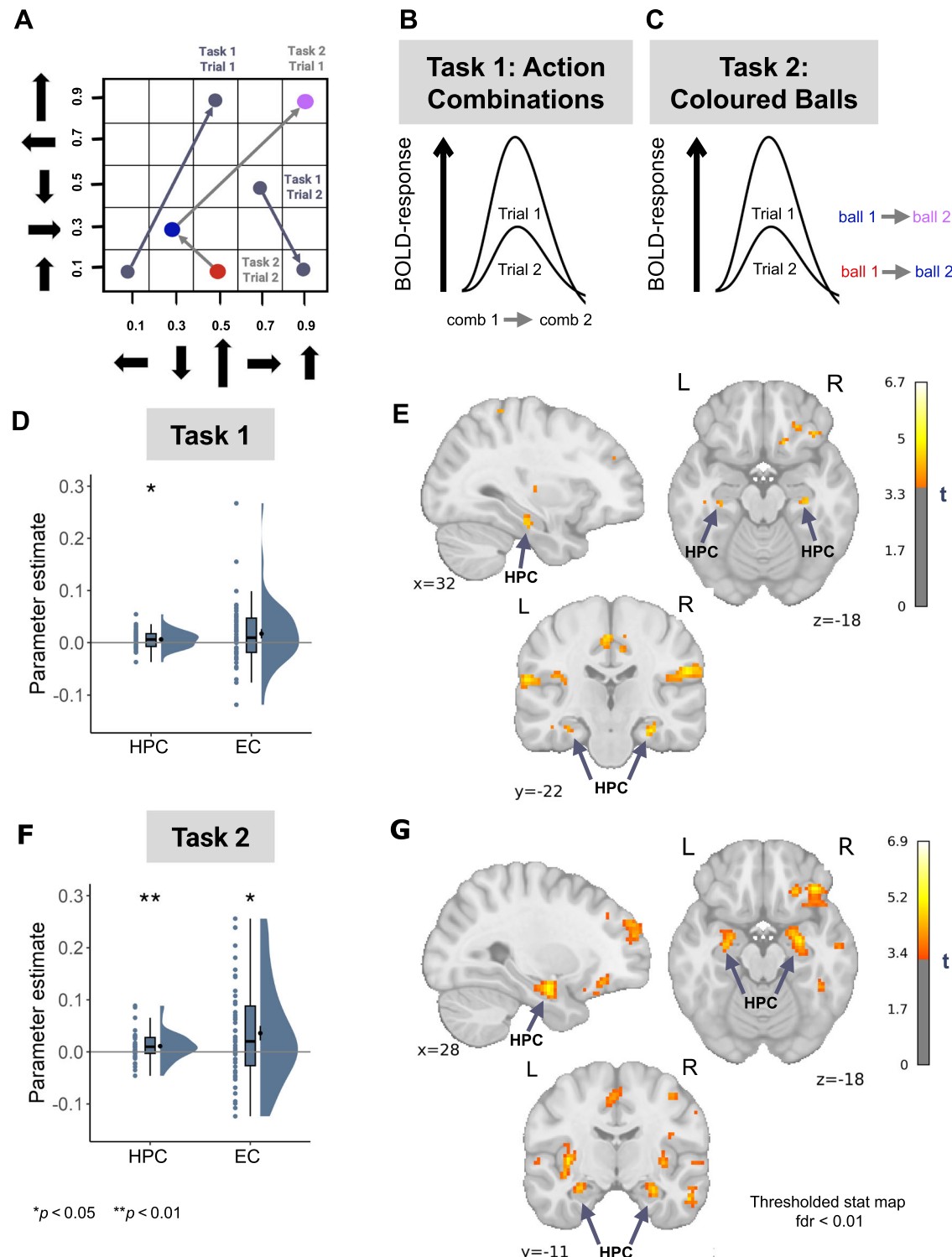

hexadirectional modulation in EC during the first Comparison Task indeed represents the abstract action-outcome space relating multiple action-outcome associations to each other, and that a vector within this space is the basis of the comparison. The control ROI analyses conducted in the hippocampus (HPC), supplementary motor area (SMA), premotor cortex (PMC), and lingual gyrus (LG) did not show a statistically significant effect (all p-values > 0.625; Supplementary Fig. 5B; see Methods, Defining ROIs). The results of a whole-brain searchlight analysis confirmed the main effect in the EC and additionally revealed a large cluster in the medial prefrontal cortex (mPFC) (Supplementary Fig. 5C).

These results indicate that the human brain develops an integrated representation of multiple action-outcome associations as an abstract map whose dimensions reflect the key components of task-relevant variation in action-outcome relations, the relational structure of which is reflected in a hexadirectional modulation of EC activity.

### Hippocampal activity reflects proximity in the action-outcome space

A second distinctive aspect of map-like relational representation within the hippocampal-entorhinal system is a sensitivity to the distance between items or positions in an abstract space (e.g., Morgan et al.[50]).

**Fig. 4 | Distance representations of the abstract action-outcome space.**
**A–C** Logic of analysis. The BOLD response in the hippocampal-entorhinal system was expected to show different levels of adaptation depending on the distance between the pair of stimuli in the action-outcome space. In particular, we predicted stronger adaptation for such trials where the two action combinations (**B**) or two coloured balls (**C**) of a pair are located closer to each other in the abstract 2D space, compared to when they are located farther apart. We modelled the presentation of the second stimulus in each pair, conceiving them as positions within the action-outcome space. **D** We observe hippocampal fMRI adaptation for trials with shorter distances between action combinations from the first Comparison Task (t(45) = 2.52, p = 0.015, Cohen's d = 0.37, 95% CI= [0.001, 0.011]; one-sided t-test). This effect was also observed on a whole-brain level (MNI peak voxel coordinates: 30, −23, −20; peak voxel t(45) = 4.87; two-sided test) (**E**). Interestingly, in their study, Morgan et al.[50] found evidence of fMRI adaptation to the physical distance between real-world landmarks in the hippocampus (HPC) as well as the additional brain regions, including the superior temporal sulcus, a region that similarly showed distance-dependent modulation of activity in our analysis (Supplementary Table 1). **F** In the second Comparison Task, in both HPC and entorhinal cortex (EC) we also show a stronger adaptation for coloured balls located closer to each other in the abstract 2D space (HPC: t(45) = 2.76, p = 0.005, Cohen's d = 0.40, 95% CI= [0.002, 0.018]; EC: t(45) = 2.57, p = 0.013, Cohen's d = 0.38, 95% CI= [0.007, 0.063]; one-sided t-test). Furthermore, the hippocampal distance-dependent adaptation was confirmed on a whole-brain level (MNI peak voxel coordinates: 27, −8, −18; peak voxel t(45) = 5.155; two-sided test) (**G**). **D, F** Dots represent data from n = 46 participants. boxplots show median and upper/lower quartile with whiskers extending to the most extreme data point within 1.5 interquartile ranges above/below the quartiles; black circles with error bars correspond to mean ± SEM; distributions depict probability density functions of data points. Source data are provided as a Source Data file. *p < 0.05; **p < 0.01; Bonferroni corrected for tests in both ROIs; Whole-brain maps were false discovery rate (FDR)-corrected using a voxel-level threshold of p < 0.01.

Previous studies have shown modulation of the hippocampal-entorhinal response by spatial[29,51] and conceptual[32,33] distance between visual objects in memory. Therefore, we hypothesised that similar mechanisms should be present when comparing pairs of action combinations in the first Comparison Task. Specifically, we expected the hippocampal-entorhinal activity to show adaptation according to the distance in the action-outcome space between the two action combinations (Fig. 4A, B; see Methods). That is, when the comparison involves two action combinations that occupy closer positions within the putative 2D action-outcome space, the BOLD response for the second combination of this successively presented pair should be reduced compared to cases where two action combinations of a pair are more distant from each other.

We found evidence for fMRI BOLD adaptation in the HPC (t(45) = 2.52, p = 0.015, Cohen's d = 0.37, 95% CI= [0.001, 0.011]; one-sided t-test), but not in the EC (t(45) = 1.89, p = 0.055, 95% CI= [−0.001, 0.035]; one-sided t-test; Fig. 4D). Consistent with the hexadirectional modulation of activity patterns in the EC, the hippocampal effect was more apparent in the left hemisphere (Left HPC: t(45) = 2.73, p = 0.009, Cohen's d = 0.4, 95% CI= [0.001, 0.012]; Right HPC: t(45) = 1.93, p = 0.064, 95% CI= [−0.0002, 0.01]; one-sided t-test; Supplementary Fig. 6A). The distance along the individual dimensions in isolation did not show a statistically significant adaptation in the HPC (all p > 0.103; see Methods, Distance-based adaptation to action combinations: control analyses). Furthermore, the statistically significant effect was not observed in control ROI analyses performed in the SMA, PMC and LG (all p > 0.068; Supplementary Fig. 6B). The BOLD adaptation effect in the HPC was further confirmed by whole-brain analysis (FDR-corrected using a voxel-level threshold of p < 0.01; MNI peak voxel coordinates: 30, −23, −20; peak voxel t(45) = 4.87; two-sided test; Fig. 4E). Additional clusters were observed in orbito-frontal cortex, frontal pole, inferior parietal lobule, cingulate cortex as well as premotor and secondary somatosensory cortices (Supplementary Table 1).

Crucially, the hippocampal BOLD adaptation effect was replicated with data from the second Comparison Task (see Fig. 4A, C). In this task, instead of action combinations, participants were instructed to compare pairs of coloured balls, and yet the HPC represented the distance between ball pairs within an underlying 2D map of action-outcome associations (Fig. 4F, G; Supplementary Fig. 6C; Supplementary Table 2; see Methods).

These findings indicate that the hippocampal activity scales with the distance between different action combinations in an abstract space that integrates action-outcome associations, thus providing additional evidence for a map-like representation of abstract action-outcome knowledge in the human brain. Similar to action combinations, the distance between the six coloured balls representing landmark outcomes of actions is also encoded in the hippocampal-entorhinal system, showing that a representation of an action-outcome space is used to compare outcomes even when the comparison does not directly involve actions themselves. This suggests retrieval of a map-like representation of abstract action-outcome associations irrespective of the stimulus format.

## Supplementary Motor Area encodes individual actions, including in the absence of action execution

Having observed the existence of neural signatures in the hippocampal-entorhinal system indicating relational representation of the action-outcome associations, we also sought to explore whether and how this representation interacts with brain regions specialised in encoding motor information. Here, we expected to find representations primarily linked to the motor implementation of actions, existing in parallel to the 2D representation of action-outcomes in the hippocampal-entorhinal system.

Recent studies suggest a close link between these two systems during skill learning tasks[52,53]. As a first step, we identified the brain regions that supported the representation of motor information related to action plans. To do this, we focused on the first Comparison Task (Fig. 1H), in which the arrow cues on each joystick were the equivalent of a motor action that the participants had performed during the immersive VR training. We reasoned that participants were likely to use a predictive forward model[7,10] to deduce outcomes from associated actions, and therefore, in addition to the spatial map-like representation of action-outcome relations, participants are expected to have another relevant representation of the motor plans required to perform these actions. More specifically, we hypothesised that motor regions such as PMC and SMA could be involved in mental simulation of actions subsequent to the visual presentation of action-related cues[1,10,54–56]. We therefore again performed a BOLD adaptation analysis on the data from the first Comparison Task where participants compared the outcomes of each of two action combinations, represented by joystick settings (Fig. 1H). Specifically, we expected a modulation of fMRI activity as a function of whether the two action combinations shared a common action (e.g., move joystick to the right) in the trial or did not share any action (see Methods). Trials with no common action between pairs were expected to elicit little or no adaptation, whereas trials with a common action between pairs were expected to show more adaptation, due to their reliance on a similar neural representation (see Fig. 5A, B and Methods). In contrast with our previous analyses, in this case, we did not have a preexisting hypothesis about the direction of BOLD modulation. This was due to the divergent findings in the literature, demonstrating both repetition suppression and repetition enhancement for similar motor actions[57,58]. We observed a significant increase in activation within the SMA, but not in the PMC, in the trials with shared actions across two action combinations (SMA: t(45)= −3.07; p = 0.005, Cohen's d = −0.45, 95% CI= [−0.029, −0.006]; PMC: t(45)= −1.78, p = 0.078, 95% CI= [−0.013, 0.0008]; two-sided t-

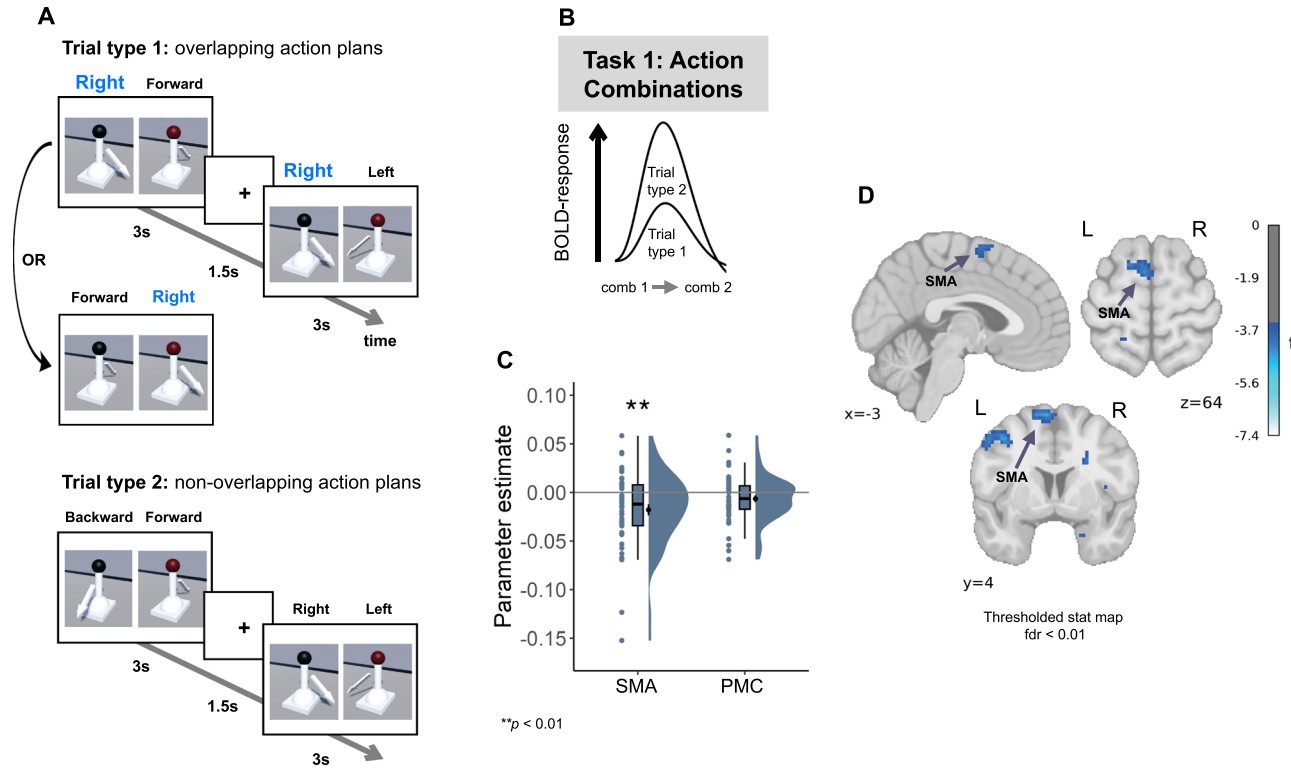

**Fig. 5 | Representation of individual actions in the SMA. A, B** Logic of analysis. Trials from the first Comparison Task were categorized as similar (Trial type 1) or dissimilar (Trial type 2) depending on whether the two action combinations of a pair shared the common action: in the current example of overlapping action plans, the common action would be moving the joystick to the right (**A**). Action could be shared between two action combinations by the same or different joysticks. We assumed that the shared motor information should elicit the modulation of BOLD response: a positive value would suggest activity suppression as a function of similarity (shown in **B**), while a negative value would indicate increased activity as a function of similarity. **C** Supplementary motor area (SMA) showed increased activity in the trials with similar action combinations (t(45)= -3.07; $p = 0.005$,

Cohen's d = -0.45, 95% CI= [-0.029, -0.006]**;** two-sided t-test). Dots represent data from $n = 46$ participants. Box plots show median and upper/lower quartiles with whiskers extending to the most extreme data point within 1.5 interquartile ranges above/below the quartiles; black circles with error bars correspond to mean ± SEM; distributions depict probability density functions of data points. Source data are provided as a Source Data file. **D** The action similarity-dependent increase in activity in the SMA was confirmed on the whole-brain level (false discovery rate (FDR)-corrected using a voxel-level threshold of $p < 0.01$; MNI peak voxel coordinates: -6, 4, 67; peak voxel t(45) = −4.673; two-sided test). **$p < 0.01$; Bonferroni corrected for tests in two ROIs.

test; Fig. 5C). We did not observe a statistically significant modulation of fMRI activity in the control ROIs (HPC, EC and LG, all $p > 0.153$; Supplementary Fig. 7). We further confirmed the repetition enhancement effect in SMA using a whole-brain analysis (FDR-corrected using a voxel-level threshold of $p < 0.01$; MNI peak voxel coordinates: -6, 4, 67; peak voxel t(45) = −4.673; two-sided test; Fig. 5D; Additional significant clusters included the inferior parietal lobule, supramarginal gyrus, primary motor cortex, PMC, inferior and superior temporal gyri, lateral occipital cortex, inferior and superior frontal gyri; Supplementary Table 3). The effect was unlikely to be driven by the distance in the 2D action-outcome space because i) the actions were arbitrarily assigned across participants (see Methods); ii) the inherent residual correlation between distance and action similarity after this arbitrary assignment was relatively low (on average 0.09); and iii) the results remained consistent and significant after we repeated the analysis introducing both regressors (distance in 2D space and number of shared actions) in the same General Linear Model (GLM; all p-values < 0.011; see Methods, Action similarity-based adaptation: control analyses).

Taken together, the results of this analysis indicate that cueing actions by observing the corresponding joystick settings elicits the activation of distinct action plans in the SMA. Given that actions and outcomes of each joystick are fully correlated in this task, we cannot completely separate action- from outcome-specific representations within the SMA. However, additional analyses show that SMA activity is modulated by similar actions of two different joysticks that do not

produce the same outcomes, suggesting that the SMA contains action-related representations (see Supplementary Fig. 8). Furthermore, in contrast to the hippocampal-entorhinal system, SMA showed no indication of a two-dimensional representation of action-outcome space based on the dimensions supplied by our task (Supplementary Fig. 6B). These findings, together with positive evidence for SMA activation by observational cueing of action plans (see Fig. 1H), is consistent with SMA representation of individual action-outcome links, without the relational organization of multiple such links into an abstract space. Indeed, models of action sequences suggest that SMA may be structured in a way that supports chunking and chaining, rather than coding relations between different action-outcome associations[59]. The findings from the current analysis allowed us to examine how this motor-related cortical area coordinates with the HPC.

### Coordination between hippocampus and SMA during evaluation and comparison of different action plans

The existence of two parallel systems, (i) a map-like representation of action-outcome relations in the hippocampal-entorhinal system and (ii) a representation of individual actions in the SMA, allowed us to test the hypothesis that the hippocampal-entorhinal abstract map interacts with individual action representations in the motor system to relate multiple individual action-outcome associations. Motivated by the distance-based modulation of hippocampal activity in the action-

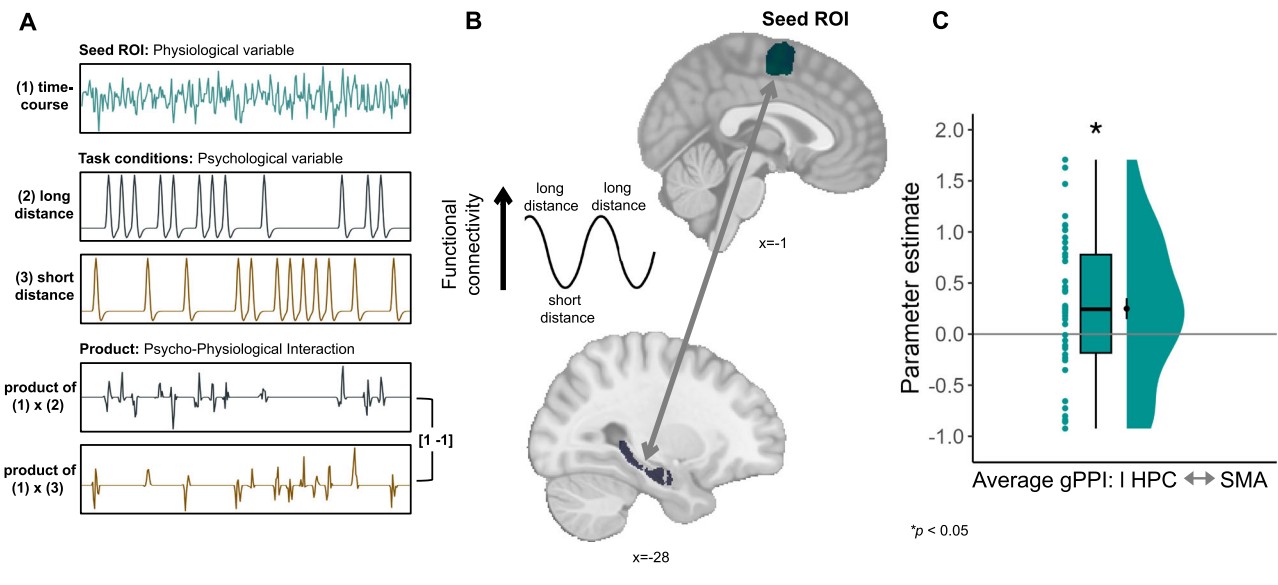

**Fig. 6 | Interaction between map-like representations in the hippocampus and individual action representations in SMA. A** Logic of the generalized psycho-physiological interaction (gPPI) analysis. (1) Extracted time-series from a ROI used as a seed region. (2) Onset regressors for each condition, convolved with a hemodynamic response function (HRF) to predict condition-specific activity. The last two regressors constitute the product of (1) and each of the task conditions (2,3), modelling the PPI term (see Methods for details). The two PPI regressors were contrasted between the two conditions to obtain condition-dependent change in the correlation between activity in the seed ROI and each of the other voxels in the brain. **B, C** In line with the modulation of hippocampal activity during the first Comparison Task based on the distance in the action-outcome space, we expected a modulation in the connectivity between the hippocampus (HPC) and the supplementary motor area (SMA), suggesting a reciprocal communication between motor regions and the hippocampus during the action comparison process. The analysis was repeated twice, once using the HPC and once using the SMA as seed regions and extracting the parameter estimates from the other region. At the group level, the connectivity between these regions increased for trials with long distances compared to those with shorter distances (t(45) = 2.42, p = 0.021, Cohen's d = 0.35, 95% CI= [−0.241, 0.956]; two-sided t-test). Dots represent data from n = 46 participants. boxplots show median and upper/lower quartile with whiskers extending to the most extreme data point within 1.5 interquartile ranges above/below the quartiles; black circles with error bars correspond to mean ± SEM; distributions depict probability density functions of data points. Source data are provided as a Source Data file. *p < 0.05.

outcome space, we aimed to investigate whether the connectivity between the HPC and SMA follows the same pattern of modulation.

We conducted a generalized psychophysiological interaction analysis[60] (gPPI) on the data from the first Comparison Task (Fig. 1H) with the HPC and SMA as seed ROIs and the distance between action combinations in the abstract action-outcome space (Fig. 1E) as a psychological component (Fig. 6A, B; see Methods). Importantly, we did not formulate a hypothesis about the direction of modulation of the connectivity between HPC and SMA. The averaged results for both seeds showed that functional connectivity between the left HPC and SMA was increased as a function of distance between action combinations in the action-outcome space (t(45) = 2.42, p = 0.021, Cohen's d = 0.35, 95% CI= [−0.241, 0.956]; two-sided t-test; Fig. 6C). The distance-based modulation of functional connectivity between the two ROIs did not change even when the connectivity was examined for each seed ROI separately, without averaging the results across seed ROIs (all p < 0.05; Supplementary Fig. 9A, B). We used LG as a control region and did not observe a statistically significant task-based change in its connectivity with left HPC or with SMA (all p > 0.252; Supplementary Fig. 9C, D). The whole-brain maps for the modulation of connectivity are shown in Supplementary Fig. 9E, F and Supplementary Tables 4, 5, confirming the mutual communication between HPC and SMA (uncorrected for multiple comparisons; statistical significance threshold defined at a single voxel level (z_α = 1.96); HPC as a seed: MNI peak voxel coordinates: 3, 1, 48; peak voxel t(45) = 3.175; two-sided test; SMA as a seed: MNI peak voxel coordinates: −26, −15, −18; peak voxel t(45) = 2.802; two-sided test). This analysis also revealed the involvement of additional regions not predicted by us, such as the posterior parietal cortex (MNI peak voxel coordinates at 62, −40, 31 and 5, −37, 53). These findings underscore that the hippocampal-

entorhinal map-like representation of action-outcome relations interacts with individual action representations in SMA during the action comparison process.

## Discussion

The human capacity to develop a diverse and highly complex repertoire of action plans is truly remarkable. Many of our behaviours are rooted in associations between actions and their outcomes, formed and leveraged in a flexible way. For example, we readily grasp that the same keypress action may cause very different outcomes depending on whether we act on a computer keyboard, a computer mouse, a radio or in some other context. However, the complexity of human action-outcome repertoires necessitates a sophisticated cognitive process of goal-directed selection. Further, multiple alternative choices are typically available, so that action selection requires comparing across available action-outcome representations[15], which is a non-trivial and taxing representational problem. To overcome this problem, multiple action options should be represented in a way that enables efficient comparison. We propose an approach suggesting that action-outcome associations could be organised in a map-like structure in the hippocampal-entorhinal system[23], potentially supporting efficient action selection within the rich human behavioural repertoire. Some action-outcome relations might be particularly suited for map-like representations simply because action and outcome are directly linked through proportional physical interactions described by the continuous quantities and dimensions of the external environment. The well-known relation between motor cortical activity and muscle force is one example of such a direct mapping[61–63]. Here we focused instead on whether the brain uses representation in the form of a cognitive map. Such maps are particularly efficient when action-

outcome structures involve non-continuous, arbitrary relations between abstract action categories and task-level outcomes.

We combined fMRI with immersive virtual reality and a sequential picture viewing task to demonstrate that (i) the hippocampal-entorhinal system can represent the relational structure of arbitrary action-outcome associations using cognitive map structures similar to those previously reported for spatial navigation, (ii) this system interacts with cortical motor regions essential for planning, selecting and executing motor actions. First we show that a two-dimensional action-outcome space underlies behavioural performance. This space could be reconstructed from our participants' judgements about how similar two action-outcome associations were to each other. Crucially, we further provide fMRI evidence for a map-like representation of action-outcome relations. Specifically, the map is reflected by a hex-adirectional signal in the EC on the one hand, representing the general structure of the abstract action-outcome space, and by the scaling of hippocampal activity with the distances between action-outcome associations that are compared according to the map.

### Learning and representation of relations between multiple action-outcome associations as a cognitive map in the hippocampal-entorhinal system

Our results are consistent with studies showing that the ability to infer relations between multiple concepts and events, known as 'cognitive mapping', is supported by the hippocampal-entorhinal system[20,22,23,64,65]. The discovery of map-like coding of space emerged from navigation studies[20,21]. Beyond the involvement of these representations in mental simulations of navigational trajectories and goals[49,66], map-like organizational principles also extend to coding perceptual information in different modalities[26,31–33,35,67,68]. However, studying more abstract spaces poses a challenge, as abstract dimensions often also involve the manipulation of sensory features of visual objects or auditory cues, which are themselves relational. Nevertheless, highly abstract forms of knowledge, such as social[36,37] or value[38] information, can also be captured by spatial map-like representation[22,25]. That is, participants may acquire map-like representations of stimuli that vary conceptually rather than perceptually.

In our study, in contrast, information about action-outcome associations is first derived from sensory and motor experiences within the virtual environment. Next, information about the relations between several action-outcome associations is integrated into the general abstract structure of the cognitive map. Ours is, to our knowledge, the first study to investigate the acquisition and use of cognitive maps for goal-directed actions, and the first to demonstrate the construction of such cognitive maps of non-spatial content from an interactive experience in immersive virtual reality. Our approach recalls Krakauer et al.'s[69] concept of a de novo learning task where a new motor controller is formed from scratch, shaping how we select and execute our actions. Indeed, our participants learned new ways to act upon the incoming bodily and environmental information through (i) grabbing and moving virtual joysticks with a VR controller and observing their virtual hand as visual feedback, and (ii) acquiring arbitrary mappings between each joystick's actions and two different types of outcome dimensions (Fig. 1E).

We selected six ball colours to label specific landmark positions in action-outcome space. Participants learned that each of these ball colours was produced by a unique combination of two joystick actions and therefore corresponded to the unique conjunction of two outcomes on the outcome dimensions (Fig. 2C, D). Our results showed that colour alone could reactivate the cognitive map in the hippocampal-entorhinal system (Fig. 4F, G), demonstrating the flexibility of such map-like structures to associate new and arbitrary additional information with their dimensions, in this case the specific colours of several landmark balls. Given that the cognitive map we have identified emerged without value-based reinforcement or

supervised learning, and without specification of any cost function[10], we suggest that, outside the laboratory, action-outcome knowledge may continuously accumulate in such map-like relational structures. These maps are then available when selecting the action that produces the most desired outcome.

### Representation of individual actions in the SMA in interaction with the hippocampal-entorhinal cognitive maps

We further investigated how these abstract maps might be used to select specific actions. We showed that in contrast to the hippocampal-entorhinal system, which represented the relational structure between multiple action-outcome associations (Figs. 3 and 4), SMA representations were restricted to individual action-outcome associations (Fig. 5). This finding aligns with previous research emphasizing the key role of the SMA as a motor structure linking movements to their arbitrary consequences[70,71] and preparing motor commands that activate desired response-stimulus associations[72]. As part of the cortical circuits for action monitoring[54,55,73], SMA has long been thought to play a crucial role in voluntary, self-initiated actions[74,75]. For example, Kühn et al.[76] showed a role of SMA in integrating motor signals for intentional action with sensory information about external consequences, resulting in the subjective experience of agency over action outcomes that accompanies goal-directed actions[77]. Our training in the virtual environment involved the coordination of abstract knowledge about different action items with motor information related to these items. Since our first Comparison Task during the scanning session involved the presentation of joystick settings as symbolic visual cues while participants evaluated action outcomes, we suggest the SMA may provide a forward simulation[11] of the corresponding action-outcome link. This forward representation of individual action-outcome linkages occurred in parallel to the hippocampal-entorhinal abstract mapping of action-outcome relations.

This finding indicates that cognitive maps do not operate in isolation, but in coordination with other brain regions from different functional domains, and cooperate to construct an abstract map[35,78–80]. In the context of motor planning, hippocampal-entorhinal maps could be used to predict and relate the outcomes of actions through interaction with motor structures such as SMA. Indeed, we observed a coupling between the two systems, with systematic modulation of the connectivity between the HPC and SMA. We speculate that the forward simulation of individual action-outcome links by SMA may allow the hippocampal-entorhinal system to position these links in an abstract cognitive map for subsequent comparison. Alternatively, the hippocampal-entorhinal system could integrate information about individual action-outcome links from SMA during the formation of the map and later serve as a reference system to access these individual action-outcome representations in SMA (see Supplementary Fig. 10). Our PPI methods cannot identify the directionality of the interactions between hippocampal-entorhinal system and SMA. Further research could address this question, for example, by measuring the flow of information using methods that combine high spatial resolution with high temporal sensitivity, such as intracranial recordings. The map-like representations might therefore represent how agents interact with their environment in a very general sense, well beyond the specific case of spatial navigation. By supporting action selection, cognitive maps could contribute to optimising the acquisition and exploitation of wide repertoire of action plans.

### Evidence for a cognitive map representation in mPFC

In addition to the hippocampal-entorhinal system, our analyses also revealed the involvement of the medial prefrontal cortex (mPFC) and the lateral orbitofrontal cortex (lOFC) in the representation of the abstract action-outcome space (see Supplementary Fig. 5C and Supplementary Tables 1 and 2). Neural representations interpreted as cognitive maps have been found in the mPFC in humans[26,47] and non-

human primates[81]. lOFC has been shown to be specifically involved in the construction of such a map (e.g., Costa et al.[82], rodent study), perhaps reflecting the direct projections from the hippocampus to these prefrontal regions[80,83,84]. Other work has suggested that the OFC, particularly its lateral part, represents the expected outcomes based on environmental statistics[85] in many tasks[86] and is involved in disambiguating the unobservable states in a cognitive map of task space[87]. The mPFC and OFC, together with the hippocampal-entorhinal system, could therefore participate in acquiring and structuring associative representations[88]. These representations may play an important role in guiding behaviour when multiple action alternatives are possible.

### Revision of the role of the hippocampal-entorhinal system in action planning

Historically, memory taxonomies distinguished between two memory systems, for explicit and implicit memory[89]. Skilled motor performance was clasically attributed to learning within an implicit, procedural memory system. This literature typically used tasks focused on ongoing sensory guidance of movement execution, such as pursuit tracking[90], and generally ignored the problem of outcome-based action selection. Explicit, declarative processes in the hippocampal-entorhinal system did not contribute[91,92]. Our findings now show that action-outcome representations in the hippocampal-entorhinal system are in fact well-suited for flexible action selection, since they represent action-outcome information in a format that supports relations between multiple options. Interestingly, earlier work had suggested hippocampal involvement in novel arbitrary mapping of visual stimuli to motor actions in human patients[93] as well as in Macaque monkeys[4]. Contemporary research in rodents[52,94] and humans[95,96] also suggests that the hippocampal-entorhinal system is indeed involved in instrumental learning, forming causal associations between actions and their consequences. This aligns with the well-established role of the hippocampal-entorhinal system in relational binding[97–101] and sensitivity to context changes[102,103]. Further, a recent case study of a patient with bilateral hippocampal loss confirmed deficits in flexible action selection[104]. Notably, the most profound impairment was evident in a task which required forming of arbitrary action-outcome associations, supporting the idea that the hippocampal formation may be implicated in learning novel and complex action-outcome mappings.

Our findings extend this understanding by demonstrating that the hippocampal-entorhinal system may integrate multiple action-outcome associations into a common representational format (a cognitive map). This format makes the relations between multiple alternative actions more efficient to compute. Classically, cognitive motor control studies focused on procedural aspects of representing an individual action, recalling the concepts of the procedural memory literature. However, more recent studies recognised an additional major contribution of explicit learning processes in action representation and control[69,105,106], but did not consider how many such representations might be organised into an overall structure. Here, we have investigated this process of relating multiple actions (and their linked outcomes) to each other, as opposed to simply planning an individual action. We show that relations between multiple action-outcome pairs are supported by map-like representations in the hippocampal-entorhinal system. Our findings suggest that multiple arbitrary action-outcome associations are represented in a similar way to other forms of semantic knowledge[4], requiring a continuous circumstantial learning and updating of these associations. Thus, while the neuroscientific literature has generally focused on procedural aspects of action memory, our study highlights that efficient instrumental action may also depend on more explicit forms of knowledge about the overall organization of action-outcome relations.

In conclusion, we have shown that the hippocampal-entorhinal system plays a key role in relating and selecting alternative action plans using map-like representations. A key advantage of this format for neural representation is its multimodality: the representation is agnostic about which features are used to define dimensions of the map, in this case the task-related features such as probability of ball catching and ball visibility. Where multiple criteria are essential for selecting appropriate actions, goal-directed behaviour might particularly rely on the dimensional representations of a cognitive map. Instead of separately representing each action option and comparing the outcome to some decision criterion, a unified representation of multiple action plans via relevant outcome dimensions could increase the efficiency of action selection. Moreover, this system could enable parallel evaluation of various options, avoiding inefficient serial processing. Therefore, our results pose a challenge to the classical declarative vs. procedural distinction in memory. Instead, our results suggest that goal-directed action planning skills rely on multiple neural systems that link action generation, motor planning, and memory.

## Methods
### Participants
Fifty-two participants were recruited via the internal database recruitment system of Max Planck Institute for Human Cognitive and Brain Sciences in Leipzig. The study was approved by the local Ethics Committee of Leipzig University, Germany (protocol number 112/21-ek). The participants gave written informed consent before the experiment and were compensated for their participation with 12 euros an hour for behavioural training as well as for the fMRI scanning sessions. All participants were right-handed and had normal or corrected-to-normal vision with no blindness to colours. We did not perform an a priori power analysis to determine sample size, as there are no studies examining the emergence of hippocampal-entorhinal abstract map-like representations in motor action planning and comparison. The sample size was chosen on the basis of experiments investigating hippocampal-entorhinal map-like representations in different domains, which have obtained meaningful results with smaller pools of participants. We gathered data from both sexes but did not evaluate gender differences, as we lacked a hypothesis regarding how cognitive maps might vary across the gender spectrum. Four participants were excluded from the analysis due to technical problems with VR. An additional 2 participants were excluded due to signal dropout in the EC. In total, 46 participants (22 female, age range 19-35 years, mean age 26.6 years, standard deviation 4.8 years) entered the analysis. The image of the individual demonstrating the VR setup (Fig. 1A) is published with their explicit informed consent.

### Experimental procedure
The experiment comprised two days of training, conducted both in immersive VR and in front of a computer screen, followed by a third day where participants were scanned using the MRI (see Fig. 1F). Over the two days, the training procedure was mostly similar and consisted of six different tasks, in which participants learned to associate different arbitrary motor actions with their outcomes. The VR training started with the Guided Exploration Task, followed by the Goal-directed Action Task. The subsequent training on a computer screen consisted of two Rating Tasks and two Comparison Tasks. On the third day, participants completed two one-hour fMRI sessions in which they performed two Comparison Tasks from the previous training days. Both fMRI sessions were preceded by a short VR training to refresh their knowledge.

### Exclusion criteria
On the second day of training, we applied two exclusion criteria: participants were excluded if they achieved 50% or less accuracy in either the Goal-directed Action Task or in any of the two Comparison Tasks.

## Stimuli

**VR environment.** Participants were seated at a table in the experimental room wearing a HMD that projected a larger virtual room partially replicating the physical environment. The table in the virtual room, matching the height and position of the real table, was equipped with two virtual joysticks and a pressable button in front of the joysticks (see Fig. 1 and Supplementary Videos 1 & 2). On the opposite side of the virtual room, at a distance of 21 virtual meters from the participants, we placed a catapult that threw the balls at them. Participants were given a hand controller device, which they used to control joysticks and catch the ball. Hand movements were displayed in the VR environment, allowing participants to grasp objects in a similar way to real life. To maximise detection of the motor actions involved in the tasks, we provided the controller for the right hand only, avoiding the inconsistent use of different hands to perform these actions. The room and the 3D objects used in the experiment were created and presented using a cross-platform game engine developed by Unity Technologies (version 2018.4.30f1, https://unity.com/).

**Action-outcome space.** Participants were trained in immersive VR to execute arbitrary action combinations by changing the settings of two virtual joysticks (e.g., move joystick 1 to the right and move joystick 2 forward), triggering a launching of a virtual ball towards them multiple times (see Fig. 1D and Supplementary Videos 1 & 2). The ball's landing point varied in location and distance, occasionally landing within reach of the participant to be caught. The term 'probability of catching' was used to describe how often the ball landed close to the participant. In addition, the ball sometimes disappeared shortly after being launched and reappeared before landing. The time interval during which the ball would disappear was determined as the period between the first and last quarter of its total flying time. The frequency with which the ball remained visible for its entire trajectory was referred to as 'probability of visibility'. Participants learned to associate different joystick actions with these two types of outcomes related to ball trajectory. The first outcome dimension, the probability of catching the ball, was controlled by joystick 1 using five different actions, each of which was associated with one of the five catching probabilities: 0.1, 0.3, 0.5, 0.7 and 0.9. Similarly, joystick 2 controlled the second outcome dimension, the probability of ball visibility, with five different actions being associated with five probabilities of visibility. This mapping allowed the arrangement of joystick actions along their outcome dimensions, which would result in a two-dimensional mapping of action-outcome associations. Thus, each action combination of two joysticks was conceived as a position in an *abstract* two-dimensional action-outcome space. Therefore, we assumed that comparing multiple action combinations of the two joysticks with each other would be equivalent to comparing multiple positions located at specific directions and distances in this 2D action-outcome space. Crucially, both outcome dimensions were chosen because of their relevance for successful interaction in the ball-catching task and clear linkage to neuroscientific accounts of action control. Thus, the probability of catching refers to body contact-related action relevance[41,42], while the visibility of the ball is highly relevant to feedback-guided execution control[43,44]. Participants were never given explicit information about the exact probabilities for each dimension and had to rely on their sensory and motor experiences in VR.

Performing any combination of actions would usually result in launching of a grey ball. However, six of the possible action combinations produced a ball of a particular colour: blue, green, purple, turquoise, red and yellow. These coloured balls represent 'landmark outcomes', distinct from the two outcome dimensions related to the probability of catching a ball and its visibility. This distinction arises because each colour is produced by a specific combination of otherwise independent actions, making these colours a joint outcome of both joysticks. The six unique colours mark six different positions in the action-outcome space and can be thought of as landmarks, possibly helping to structure this abstract space by providing prominent reference points. Crucially, in addition to acting as reference points for the map, the landmark outcomes were introduced to explore the multimodal nature of abstract cognitive maps, investigating whether these maps would assimilate a range of very different types of stimuli associated with joystick actions.

The actions of each joystick were randomly assigned to the corresponding outcome dimension in such a way as to ensure that no participant had exactly the same action-outcome mapping. Furthermore, for each individual mapping, similar actions were never assigned to similar values on both dimensions. For example, if moving the first joystick to the left would result in a minimum probability of catching the ball, moving the second joystick to the left would not result in a minimum probability of its visibility (Fig. 1E). The six landmark positions within the action-outcome space, associated with specific probabilities of ball catching and its visibility, were kept the same for all participants. However, the six colours were uniquely mapped to these positions for each participant. This randomized approach ensured that the dimensions of the action-outcome space were not influenced by the specifics of the action or any potential colour bias. Crucially, the possibility of thinking about action-outcome associations in a two-dimensional way was never revealed to the participants.

**Action performance and ball catching.** Every action began with the joystick positioned upright and required moving it along one of the five directions, until the joystick reached the final position (e.g. moved to the right). Five distinct actions of each joystick were defined as moving the corresponding joystick to the left, backward, up, right, and forward (see Fig. 1D). If the joystick was not moved in any of the possible directions, or if at any time the movement did not follow the clearly identifiable direction, the action was considered invalid and had to be repeated from the upright position of the joystick. After correctly executing an action combination, a ball was launched at the participant. The area where it was possible to catch the ball was defined as a semicircle in front of the participants with a radius of ca. 1 meter. The balls, whether falling within this catchable area or farther away, could land on the floor or table at varying locations and distances from the participant. Each ball was thrown with a projectile motion calculated on the basis of the distance it had to cover. Participants were asked to catch the ball each time it came close enough. When the ball was caught, the controller in participants' hand vibrated for 2 seconds during which the ball remained attached to the hand. If participants missed the catchable ball, they received a visual feedback. The next ball was thrown either when the current one disappeared from the participants' hand or when it touched the room floor.

## Behavioural training
### VR Tasks
**Familiarization phase.** On the first day of the experiment, participants were familiarized with the controls of the VR environment while wearing the HMD and holding a VR controller in their right hand. Following the experimenter's instructions, participants completed a 5-minute training session in which they learned how to perform the actions by grasping and correctly changing the settings of the virtual joysticks, how to activate the catapult to throw the balls and how to catch the approaching ball.

**Guided Exploration Task.** After a short familiarization with the VR environment, participants performed a Guided Exploration Task, which consisted of two parts on the first training day and only the first part on the second and third days.

In the first part they learned action-outcome associations, while the second part taught them the combinations of actions that produce each of the six uniquely coloured balls. Crucially, participants' task was

not to maximise the number of balls they caught, but to learn the relations between different actions and their outcomes. For that they were instructed to find out and remember (i) which actions of joystick 1 were associated with which probabilities of catching the ball and (ii) which actions of joystick 2 were associated with which probabilities for the ball to remain visible for its entire flying trajectory. The exact probabilities were never made explicit to the participants and they had to rely on their own judgement throughout the entire experiment. In each trial participants were given two joysticks in an upright position and were shown two arrows: one arrow next to each of the two joysticks (Supplementary Video 1). The arrows indicated which combination of actions they had to perform in the current trial. The actions were executed by moving two joysticks and leaving them in one of the five different positions. Importantly, before activating the catapult, participants could always perform the same action again or execute a different action without any time constraint. The catapult could be activated by touching the virtual button between the joysticks, but only if any combination of actions was successfully performed. After touching the button, both joysticks disappeared and the catapult proceeded to launch 10 balls consecutively at the participants. However, to ensure that participants remembered which actions were performed in the current trial, the catapult stopped after having thrown the ball 5 times and the joysticks reappeared in an upright position in front of the participants. They performed the same actions as at the beginning of the trial, making the catapult throw the ball a further 5 times. The first part of the task consisted of 25 trials, spanning all 25 possible combinations of actions in a randomized order and prompting participants to experience every possible combination of two task-relevant outcomes equally often.

In the second part of the task, participants began with a 5-minute free exploration period to discover combinations of actions that produced coloured balls. Each time participants performed the combination of actions that would launch one of the coloured balls, the hinting word 'coloured' appeared on the small rectangle on the table. The rectangle then changed colour to magenta. To speed up the task, any ball was launched only once. Following exploration, participants were given instructions for each trial, asking them to produce a ball of a particular colour. After performing the action combination associated with the target ball, the ball again was launched 10 times in a row. If an incorrect combination was made, arrows indicated the correct combination on the subsequent trial, ensuring that participants performed the correct actions. The second part of the task required the production of each of the six coloured balls only once if the corresponding combination of actions was performed correctly, and a second time if the production was incorrect. The similar balls would not be produced on adjacent trials. The minimum number of trials to complete this part of the task was 6 (M = 22.46; SD = 4.91; Two participants excluded from only this analysis due to missing data in this part of the task).

**Goal-directed action task.** After learning the action-outcome associations in the Guided Exploration Task, we tested participants' knowledge using a Goal-directed Action Task with multiple-choice questions (Supplementary Fig. 1 and Supplementary Video 2). In this task, participants had to infer actions from their associated outcomes and execute them, thereby triggering the catapult to launch the ball 10 times in a row. The purpose of this task was to examine whether participants have formed multiple predictive forward-inverse models[7,10], linking outcomes to actions, and could compare these models in order to guide selection of the appropriate action required to achieve a task-relevant outcome. The task was divided into pairs of trials, where the first trial in each pair instructed participants to perform actions associated with a particular coloured ball. The second follow-up trial allowed them to use the previous ball as a reference point and choose the actions among the restricted alternatives to achieve a desired

outcome based on the instructions. Specifically, the second trial presented participants with two arrows next to each of the joysticks, pointing to the two alternative options which could be performed using the corresponding joystick. The instruction asked participants to perform one of the action options with each joystick, resulting in a specific change in the associated outcomes compared to the coloured ball from the previous trial. To provide an example, the instruction could prompt participants to "select a combination of actions that results in a higher probability of catching the ball and a lower probability for the ball to remain visible compared to the blue ball" (Supplementary Fig. 1A). The blue ball, in this case, would be the ball that participants produced in the preceding trial. To avoid making the action selection process too easy, the two action options per joystick rarely included an action similar to the previous trial. This only occurred in cases where the instruction asked participants to produce a catching probability or a visibility probability similar to the previous coloured ball. If participants performed an incorrect action combination, an additional subsequent trial would be included, showing the correct action combination for the coloured ball or for the follow-up trial, ensuring further learning of action-outcome associations and the relations between them.

This task explicitly tested participants' knowledge of the coloured balls (landmark outcomes) and the two outcome dimensions associated with actions. Although the second follow-up trial in each pair of trials could be solved without using the coloured ball as a reference point, we deliberately created questions that referred to the coloured balls. This was done to encourage participants to relate the coloured balls to the outcomes associated with other neutral combinations of actions. In this way, we ensured that participants remembered the unique combinations of actions that produced the coloured balls, while at the same time ensuring that they learned the general relational structure of the actions and their outcomes.

The order of the colours for the instructed production of coloured balls was shuffled, with each ball being produced 4 times throughout the task. Combined with the follow-up trials, it resulted in a total of 48 trials, if each response was correct (Day 1: M = 54.71; SD = 5.41; Day 2: M = 49.36; SD = 1.95). The correct combinations of actions were randomly selected in a way to ensure that participants experienced every possible combination of actions at least once. For each correct answer, participants received a point, collecting a maximum of 48 points in the task.

### Computer Tasks
Similar to the VR tasks, the computer tasks were built and presented using a game engine developed by Unity Technologies (version 2018.1.9f2, https://unity.com/).

**Rating Tasks.** The VR training was followed by the two Rating Tasks presented on a computer screen, where participants had to estimate the similarity between pairs of stimuli (Fig. 1G). In the first Rating Task, participants were shown pairs of action combinations, each combination consisting of two images of joysticks in an upright position with arrows indicating one of the five possible actions. In the second Rating Task, participants were presented with pairs of the coloured balls. All images were presented on a light grey background with a single adjustable horizontal slider in the lower part of the screen. The labels 'very dissimilar' and 'very similar' were displayed to the left and right of the slider.

The first Rating Task instructed participants to imagine the outcomes of both actions in each combination and to rate overall how similar the two presented combinations of actions were in terms of the associated outcomes. Equally, the second Rating Task instructed participants to imagine the probabilities of catching and visibility associated with each of the coloured balls in a pair and to rate the overall similarity of the balls. Participants indicated their response without any time limit by adjusting the horizontal slider with a computer mouse

and confirming their choice by pressing a key on a keyboard, after which the stimuli disappeared from the screen for 2 seconds.

We sampled 8 action combinations corresponding to the 8 positions in the action-outcome space. The positions, consistent across all participants, were sampled from different parts of the space, covering each quadrant and the central part of it. All possible pairs of the 8 action combinations were presented twice in a randomized order with each combination displayed once on the left and once on the right side of the screen. We ensured that every pair was sampled at least once before any of the pairs was presented for the second time. This resulted in a total number of 56 trials. Using the same logic, all possible pairs of 6 coloured balls were presented twice in a randomized order, with each ball presented once on the left and once on the right side of the screen, resulting in a total number of 30 trials.

The order of the tasks was counterbalanced in a way that 50% of the participants were first asked to rate the similarity between action combinations, while another 50% started with rating the similarity between coloured balls.

**Comparison tasks.** In the following two Comparison Tasks, participants were instructed to compare the action combinations (Comparison Task 1) or coloured balls (Comparison Task 2) with respect to their associated outcomes (Fig. 1H). Participants were then asked to respond to a question or statement regarding the change in these outcomes between the stimuli.

Specifically, on each trial of the first Comparison Task, participants were sequentially presented with a pair of action combinations, each combination consisting of two images of joysticks with arrows cueing one of the five actions. Participants were instructed to explicitly judge how the two associated outcomes changed from the first to the second action combination of the pair. Each combination was displayed on the screen for a duration of 3 seconds, followed by a 1.5-second break that included a fixation cross before the presentation of the next combination. Subsequently, participants were given a 4.5-second reflection period to consider how the two outcomes changed from one action combination of the pair to the other and during which they again saw a fixation cross on the screen. This reflection period was succeeded by a question or a statement on the change in the outcomes. In half of the trials, participants were presented with a question regarding either the first or the second outcome dimension, such as "how did the probability of catching change?" or "how did the probability of visibility change?". Three options were shown below the question: 'increased', 'decreased' or 'stayed the same'. Participants could select one of these options by pressing one of the three keys: 1, 2 or 3. As participants did not know in advance which of the two types of question they would receive, they were instructed to consider the change in both types of outcome during the reflection period. In the other half of the trials, they were shown a two-sentence statement, such as "probability of catching decreased; probability of visibility increased". Each sentence described the change in one of the two outcomes, thereby integrating both outcome dimensions in a single task. Participants were asked to determine whether both sentences were true, if only one sentence was false, or if both sentences were false. Again, three options were presented below the statement: 'both true', 'one false' or 'both false'. The choice could be indicated by pressing one of the three corresponding keys. Both questions and statements were presented randomly across 24 trials in each run. Participants completed two runs of the task on the first day of training and three runs on the second day, resulting in 48 and 72 trials, respectively. For the fMRI version of the task, we selected a special set of 12 types of action combination pairs (special 'directions'. see Methods, fMRI tasks). During the behavioural training, these 12 types of pairs were randomly presented alongside the other pairs in each run. In addition, across all runs on each day, we ensured that each position within the

action-outcome space was sampled by covering it with the first action combination of the pairs.

The second Comparison Task was constructed similarly to the first. Participants were presented sequentially with a pair of coloured balls on each trial and were instructed to judge the change in outcome dimensions from the first to the second ball of a pair. Each ball remained on the screen for 1.5 seconds with a 1.5-second break in between. After a 4.5-second reflection period, participants were shown a question or statement regarding the outcome change. In each run of the task, participants were presented with all possible pairs of 6 coloured balls. Each pair was sampled twice in a randomized order, with each ball presented once as the first and once as the second stimulus of the pair. A second presentation of the ball pair was separated from its first presentation by at least one trial in between. Additionally, the last ball of the pair in the previous trial never matched the first ball of the pair in the next trial. The trial arrangement created for each run varied between different runs and across participants. Participants completed two and three runs of the task on the first and second training days. This resulted in a total of 60 trials on the first day and 90 trials on the second day, with each run consisting of 30 trials. For 28 trials of every run, each type of question and true or false statement was presented equal amount of time in a randomized order. However, for the last 2 trials, a question and a statement type were randomly selected from the available options. For both Comparison Tasks, participants were given 10 seconds to respond on the first training day, which was reduced to 6 seconds on the second training day. Following each run, participants received a general feedback on the accuracy of their performance, along with an overall accuracy score at the end of each task. If their overall performance in the task exceeded 80%, they received a bonus of 2.50€. All participants completed the Comparison Tasks in the same order, matching the order in which they later performed the tasks in the fMRI session.

**fMRI tasks**

During the scanning sessions, participants performed the same Comparison Tasks as on the previous training days. In the first scanning session they compared pairs of action combinations (Comparison Task 1), whereas in the subsequent session they were presented with pairs of coloured balls instead (Comparison Task 2). As before, participants were instructed to judge the change in the two underlying outcomes of the presented stimulus pair and to respond to the subsequently displayed questions or a statement (Fig. 1H; see Methods). The tasks were based on the assumption that the comparison of stimuli in terms of their outcomes would rely heavily on the mentally created abstract action-outcome space, covering different directions and distances within that space. In this context, the first stimulus of the pair served as the starting point for the direction of the current trial, while the second stimulus marked the end point of that direction. To streamline the process, the response time was reduced to 4 seconds, which was long enough for participants to read the displayed text, but short enough to encourage them to make mental comparisons during the reflexion period before the question appeared. They were asked to select the answer by pressing the left, middle or right button on a button box. The Comparison Task from the first scanning session was divided into 6 runs of 24 trials each, giving a total of 144 trials. The Comparison Task from the second scanning session consisted of 5 runs of 30 trials each. Participants completed a total of 150 trials. The randomization structure for the second Comparison Task corresponded to the training days (see Methods).

**Randomization structure for the first Comparison Task.** The organization of the 2D abstract action-outcome space (Fig. 1E) allowed the controlled sampling of 12 directions with an angular difference of ~30° between adjacent directions. The actual sampled mean directions are 0°, 28.35°, 61.65°, 90°, 118.35°, 151.65°, 180°, 208.35°, 241.65°, 270°,

298.35°, 331.65°, calculated in relation to the x-axis as 0°. The 8 non-cardinal directions could deviate from the ideal direction by either ±3.43° or ±3.7°. Every direction was sampled 12 times from 12 different starting positions in the space, randomly selected from the set of possible starting positions for the current direction. Each direction occurred twice per run: once in one half and once in the second half of the run, in non-consecutive trials. The order of direction sampling was randomized across runs to ensure variability. We deliberately avoided long blocks of trials with directions that were considered either similar or dissimilar to each other (see Methods, Representational similarity analysis: six-fold symmetry), aiming for balanced trial structure. To do this, we prevented the same direction type (similar or dissimilar) from following each other for more than a maximum of 3 consecutive trials. Our design aimed for a balance between repetition and alternation among direction types, while controlling for the equal occurrence of all possible transitions between these two types (see Supplementary Fig. 11A, B). The generated trial structure of the run was different across participants and runs (see Supplementary Fig. 11C). Assuming that the shortest distance between locations in the action-outcome space is the unit 1, the space was sampled using 3 different distances for each direction. The distances covered 4, 3 and 2 units for the cardinal directions (0°, 90°, 180° and 270°) (M = 2.8 units for each direction, calculated based on the frequency of each distance covered by that direction. See Supplementary Fig. 3 for details), while 4.5, 3.6 and 2.2 units were covered for all other directions (M = 3 units for each direction; See Supplementary Fig. 3 for details). Given the limitations of the action-outcome space architecture, and with the aim of maintaining variation, the longer distances were of 2 different lengths, while the shorter distances were all of the same length. Within each run, the distances of each non-cardinal direction could be balanced by being presented once as the shortest and once as the longer distance in a randomized order. Due to design constraints, the number of shortest distances for each cardinal direction was one less than the number of longer distances, resulting in a slightly unbalanced presentation within runs. Finally, in each run, each type of question and statement (true and false) was randomly assigned an equal number of times to the similar and dissimilar directions.

## Statistics

The statistical significance of the effects was determined using t-tests based on a non-parametric permutation-based approach. The observed t-values were compared against a surrogate distribution obtained from 10000 random sign-flips. The non-parametric tests were performed against 0 when a single condition was considered, or to assess within-participant differences between conditions. One-sided or two-sided tests are used, depending on whether or not we had a pre-existing hypothesis about the direction of the effect (see details in the description of the analysis). All the reported p-values, except for the control analyses, are adjusted for multiple comparisons using Bonferroni correction. The adjustments take into account the number of masks and separate comparisons in left and right hemispheres.

## Behavioural data analysis

**Analysis: VR Tasks.** Participants' overall performance in the Goal-directed Action Task at the end of VR training was high, as reflected by the percentage of correctly chosen action combinations (Day 2: Mean (M) = 97.3%, Standard deviation (SD) = 3.99%; Supplementary Fig. 1B). To ensure that the outcome dimensions were orthogonal and did not influence behavioural performance, we examined catching performance from the Guided Exploration Task on the third day of training, when participants were fully adapted to the experiment. The combinations of actions, and thus the combinations of outcomes, were equally sampled in the task. This allowed an unbiased test of whether the five probabilities of ball visibility affected one of the three highest catching probabilities (see Methods). We found no statistically

significant effect of ball visibility on catching performance (for catching probability of 0.9: Friedman's $\chi^2(4) = 6.70$, $p = 0.153$; for catching probability of 0.7: Friedman's $\chi^2(4) = 3.31$, $p = 0.508$; for catching probability of 0.5: Friedman's $\chi^2(4) = 4.67$, $p = 0.32$).

**Analysis: Rating Tasks.** We investigated whether participants' behavioural responses in the two Rating Tasks could capture the existence of a two-dimensional relational representation of action-outcome associations. For the first Rating Task, participants' estimates of similarity between pairs of action combinations were normalised to fall between 0 and 1 and then averaged across the two repetitions of each pair. The resulting similarity scores were then analyzed by calculating the Spearman's correlation with the Euclidean distances between the respective action combinations in the action-outcome space (Fig. 2A). As a next step, we reconstructed the participants' mental map from the pairwise similarity scores by applying multidimensional scaling (MDS) and obtaining coordinate positions that represent the structure of these similarity scores[107]. This analysis was performed using MATLAB 2019b. Metric stress served as the cost function during the MDS process, and a random initial configuration of points was utilised. The resulting coordinates were mapped using Procrustes analysis to the original positions of action combinations in the action-outcome space (Fig. 2B). The Procrustes distance, a measure of goodness of fit, was assessed as the normalised sum of squared errors between the reconstructed and actual positions. To evaluate the significance of the mapping, we fitted the MDS coordinates using the same Procrustes analysis to the sets of coordinates with shuffled action combination – position assignment. This resulted in a distribution of Procrustes distances from 40320 possible permutations, which we thresholded at 5th percentile (a threshold, we termed critical distance) corresponding to the threshold for statistical significance at α = 0.05. At the group level, we assessed whether the mapping between the reconstructed and actual positions outperformed the mapping that constituted the predefined threshold of each participant's distribution from the permutation analysis (see Fig. 2B and Supplementary Fig. 12B).

We performed similar steps for the second Rating Task, correlating the participants' estimated similarity scores between the pairs of coloured balls with the Euclidean distances between these pairs in the action-outcome space using Spearman's correlation (Fig. 2C). Following the same procedure as described above, we applied MDS to the pairwise similarity scores and mapped the resulting coordinates to the original positions of the coloured balls in the action-outcome space using Procrustes analysis (Fig. 2D). To assess the significance of the mapping, we performed permutation tests using shuffled balls - position assignment and tested whether participants' fit of two maps was better than the critical distance. The critical distance was obtained from 720 possible permutations (see Fig. 2D and Supplementary Fig. 12C).

## MRI data analysis

**MRI data acquisition and preprocessing.** MRI data were acquired using a 32-channel head coil on a 3 Tesla Siemens Magnetom SkyraFit system (Siemens, Erlangen, Germany). Axial-oriented fMRI scans were obtained through T2∗-weighted whole-brain gradient-echo echo planar imaging (GE-EPI) with multiband acceleration, sensitive to blood-oxygen-level-dependent (BOLD) contrast (Feinberg et al., 2010; Moeller et al., 2010). The parameters for the fMRI sequence were: TR = 1500 ms, TE = 22 ms, voxel size = 2.5 mm isotropic, FOV = 204 mm, flip angle = 80°, bandwidth = 1794 Hz/Px, 63 interleaved slices, distance factor = 10 %, phase encoding direction = A-P. Between the task runs, field maps were acquired using the opposite phase-encoded EPIs with the following parameters: TR = 8000 ms; TE = 50 ms; voxel size = 2.5 mm isotropic; field of view = 204 mm; flip angle = 90°; partial fourier = 0.75; bandwidth = 1794 Hz/Px; multi-band acceleration factor = 1; 69 slices interleaved; slice thickness = 2.5 mm; distance factor = 0

%. This was done to correct for magnetic field inhomogeneities. At the end of the second scanning session, a T1-weighted MP2RAGE anatomical scan was obtained (Parameters: TR = 2300 ms; TE = 2.98 ms; voxel size = 1 mm isotropic; field of view = 256 mm; flip angle = 9°; bandwidth = 240 Hz/Px; slice thickness = 1 mm; distance factor = 50 %). Task stimuli were presented to participants on a screen, which they viewed through a mirror fixed to the head coil.

**Anatomical data preprocessing.** Results included in this manuscript come from preprocessing performed using *fMRIPrep* 21.0.2 (Esteban, Markiewicz, et al. (2018); Esteban, Blair, et al. (2018); RRID:SCR_016216), which is based on *Nipype* 1.6.1 (K. Gorgolewski et al. (2011); K. J. Gorgolewski et al. (2018); RRID:SCR_002502). A total of 1 T1-weighted (T1w) images were found within the input BIDS dataset. The T1-weighted (T1w) image was corrected for intensity non-uniformity (INU) with N4BiasFieldCorrection (Tustison et al. 2010), distributed with ANTs 2.3.3 (Avants et al. 2008, RRID:SCR_004757), and used as T1w-reference throughout the workflow. The T1w-reference was then skull-stripped with a *Nipype* implementation of the antsBrainExtraction.sh workflow (from ANTs), using OASIS30ANTs as target template. Brain tissue segmentation of cerebrospinal fluid (CSF), white-matter (WM) and gray-matter (GM) was performed on the brain-extracted T1w using fast (FSL 6.0.5.1:57b01774, RRID:SCR_002823, Zhang, Brady, and Smith 2001). Brain surfaces were reconstructed using recon-all (FreeSurfer 6.0.1, RRID:SCR_001847, Dale, Fischl, and Sereno 1999), and the brain mask estimated previously was refined with a custom variation of the method to reconcile ANTs-derived and FreeSurfer-derived segmentations of the cortical gray-matter of Mindboggle (RRID:SCR_002438, Klein et al. 2017). Volume-based spatial normalisation to two standard spaces (MNI152NLin2009cAsym, MNI152NLin6Asym) was performed through nonlinear registration with antsRegistration (ANTs 2.3.3), using brain-extracted versions of both T1w reference and the T1w template. The following templates were selected for spatial normalisation: *ICBM 152 Nonlinear Asymmetrical template version 2009c* [Fonov et al. (2009), RRID:SCR_008796; TemplateFlow ID: MNI152NLin2009cAsym], *FSL's MNI ICBM 152 nonlinear 6th Generation Asymmetric Average Brain Stereotaxic Registration Model* [Evans et al. (2012), RRID:SCR_002823; TemplateFlow ID: MNI152NLin6Asym].

**Functional data preprocessing.** For each of the 11 BOLD runs found per participant (across all tasks and sessions), the following preprocessing was performed. First, a reference volume and its skull-stripped version were generated using a custom methodology of *fMRIPrep*. Head-motion parameters with respect to the BOLD reference (transformation matrices, and six corresponding rotation and translation parameters) are estimated before any spatiotemporal filtering using mcflirt (FSL 6.0.5.1:57b01774, Jenkinson et al. 2002). A $B_0$-nonuniformity map (or *fieldmap*) was estimated based on two (or more) echo-planar imaging (EPI) references with topup (Andersson, Skare, and Ashburner (2003); FSL 6.0.5.1:57b01774). The estimated *fieldmap* was then aligned with rigid-registration to the target EPI (echo-planar imaging) reference run. The field coefficients were mapped on to the reference EPI using the transform. BOLD runs were slice-time corrected to 0.704 s (0.5 of slice acquisition range 0s-1.41 s) using 3dTshift from AFNI (Cox and Hyde 1997, RRID:SCR_005927). The BOLD reference was then co-registered to the T1w reference using bbregister (FreeSurfer) which implements boundary-based registration (Greve and Fischl 2009). Co-registration was configured with six degrees of freedom. Several confounding time-series were calculated based on the *preprocessed BOLD*: framewise displacement (FD), DVARS and three region-wise global signals. FD was computed using two formulations following Power et al. (absolute sum of relative motions, Power et al. (2014)) and

Jenkinson et al. (relative root mean square displacement between affines, Jenkinson et al. (2002)). FD and DVARS were calculated for each functional run, both using their implementations in *Nipype* (following the definitions by Power et al. 2014). The three global signals are extracted within the CSF, the WM, and the whole-brain masks. Additionally, a set of physiological regressors were extracted to allow for component-based noise correction (*CompCor*, Behzadi et al. 2007). Principal components are estimated after high-pass filtering the *preprocessed BOLD* time-series (using a discrete cosine filter with 128 s cut-off) for the two *CompCor* variants: temporal (tCompCor) and anatomical (aCompCor). tCompCor components are then calculated from the top 2% variable voxels within the brain mask. For aCompCor, three probabilistic masks (CSF, WM and combined CSF + WM) are generated in anatomical space. The implementation differs from that of Behzadi et al. in that instead of eroding the masks by 2 pixels on BOLD space, the aCompCor masks are subtracted a mask of pixels that likely contain a volume fraction of GM. This mask is obtained by dilating a GM mask extracted from the FreeSurfer's *aseg* segmentation, and it ensures components are not extracted from voxels containing a minimal fraction of GM. Finally, these masks are resampled into BOLD space and binarized by thresholding at 0.99 (as in the original implementation). Components are also calculated separately within the WM and CSF masks. For each CompCor decomposition, the *k* components with the largest singular values are retained, such that the retained components' time series are sufficient to explain 50 percent of variance across the nuisance mask (CSF, WM, combined, or temporal). The remaining components are dropped from consideration. The head-motion estimates calculated in the correction step were also placed within the corresponding confounds file. The confound time series derived from head motion estimates and global signals were expanded with the inclusion of temporal derivatives and quadratic terms for each (Satterthwaite et al. 2013). Frames that exceeded a threshold of 0.5 mm FD or 1.5 standardised DVARS were annotated as motion outliers. The BOLD time-series were resampled into standard space, generating a *preprocessed BOLD run in MNI152NLin2009cAsym space*. First, a reference volume and its skull-stripped version were generated using a custom methodology of *fMRIPrep*. The BOLD time-series were resampled onto the following surfaces (FreeSurfer reconstruction nomenclature): *fsnative, fsaverage*. Automatic removal of motion artifacts using independent component analysis (ICA-AROMA, Pruim et al. 2015) was performed on the *preprocessed BOLD on MNI space* time-series after removal of non-steady state volumes and spatial smoothing with an isotropic, Gaussian kernel of 6 mm full-width half-maximum (FWHM). Corresponding "non-aggresively" denoised runs were produced after such smoothing. Additionally, the "aggressive" noise-regressors were collected and placed in the corresponding confounds file. All resamplings can be performed with *a single interpolation step* by composing all the pertinent transformations (i.e. head-motion transform matrices, susceptibility distortion correction when available, and co-registrations to anatomical and output spaces). Gridded (volumetric) resamplings were performed using antsApplyTransforms (ANTs), configured with Lanczos interpolation to minimize the smoothing effects of other kernels (Lanczos 1964). Non-gridded (surface) resamplings were performed using mri_vol2surf (FreeSurfer).

**Defining ROIs.** We focused on analyzing hexadirectional signals in the entorhinal cortex (EC) and distance representations within the both, EC and hippocampus (HPC). Previous work has reported an existence of grid-like representations in the EC[27,34,35,49,102] and other brain regions[26,31]. Further studies demonstrate that the HPC, which

has been associated with inferential reasoning and generalization, plays a key role in construction and generalization of relational knowledge[28,29,32,33,50,108–111]. Therefore, we targeted these regions for planned analyses on integrated and relational coding of the action-outcome associations.

We chose the premotor cortex (PMC) and the supplementary motor area (SMA) to investigate whether participants represent individual motor actions while reasoning about their outcomes, thus having a parallel relevant representation of the motor plans in addition to the integrated map-like representation of the action-outcome associations. We did not focus on the primary motor cortex, as participants did not have to perform the actions during the fMRI session. In addition, previous literature mainly points to the involvement of PMC and SMA in action planning, such as the sensorimotor planning of hand movements directed toward objects[112] or motor imagery of familiar actions[113,114]. Moreover, insights from single-cell recordings in monkey premotor neurons indicate that the population coding within this area contains information about the trajectory of upcoming movements[115,116]. Therefore, we aimed to identify action-related motor information present in these specific brain regions.

The majority of the Region of Interest (ROI) masks were taken from the Julich Brain Atlas, with the exception of the SMA and lingual gyrus (LG) masks, which were taken from the Harvard-Oxford Brain Atlas. All masks were resampled to match the 2.5 mm resolution of the functional images. Given that the EC is an area where we might anticipate the dropout of numerous voxels, for each participant, we further intersected the EC mask with the average of the brain masks obtained for each run during preprocessing. The masks were thresholded at two levels of probability for voxels falling within these masks, using 25% increments. The hippocampal-entorhinal masks were thresholded at 75% probability to take a more conservative approach and to account for potential overlap between masks. Similarly, the PMC was also thresholded at 75% for conservative reasons. The SMA mask was set to a 50% probability threshold due to the limited number of voxels falling within this region when thresholded at a higher probability. The LG, thresholded at 50% probability, was consistently selected as the control ROI for all analyses, as we did not expect any of the effects to occur in this region. Other ROIs were used as either main or control ROIs, depending on the type of analysis. All ROI analyses are reported with a significance threshold of $p < 0.05$. For whole brain data, a false discovery rate (FDR) correction was applied at a voxel-level threshold of $p < 0.01$.

**General linear models (GLMs).** The GLMs of the functional images were calculated using a python module "Nilearn" (https://nilearn.github.io/stable/index.html) for the MRI data analysis. Given the sensitivity of the BOLD signal to motion and physiological noise, all GLMs included framewise displacement, 24 head motion parameters (three translations and three rotations, their first and second-order derivatives) and 12 parameters for WM, CSF and global signal change (their first and second-order derivatives). These parameters were obtained during preprocessing. As part of the additional parameters for denoising, 9 cosine regressors were modelled by Nilearn. In total, each GLM included 46 nuisance regressors. All regressors, including analysis-dependent task-related regressors, were convolved with a canonical HRF and were applied to the fMRI data. Additionally, a temporal high-pass filtering at a frequency of 0.01 Hz was done to eliminate low-frequency fluctuations. Prior to entering the GLMs, the data were spatially smoothed. For the representational similarity analysis (RSA, see below), a minimal smoothing was applied using a Gaussian kernel with a FWHM of 2.5 mm (1 times the voxel size). For all other analyses, the data were smoothed using a Gaussian kernel with a FWHM of 7.5 mm (3 times the voxel size).

**Representational similarity analysis: six-fold symmetry.** To investigate whether the comparison of two action combinations associated with different positions in the action-outcome space was supported by the grid-like representation in the EC, we applied Representational Similarity Analysis[48] (RSA) to the neuronal data from the first Comparison Task. The analysis was conducted using the Python Representational Similarity Analysis Toolbox (rsatoolbox, version 0.1.2, https://rsatoolbox.readthedocs.io/en/stable/). We calculated a GLM for the functional images from the first fMRI session of each participant and each run separately. The task-related regressors of no interest included 4 regressors, modelling the presentation of the first and second stimuli, the presentation of the question and participant's response. The 12 regressors of interest were created based on the 12 sampled directions in the action-outcome space (see Methods, fMRI tasks), modelling the onset of the reflection phase of the corresponding trials. In total, each GLM included 62 regressors.

As a next step, we tested whether the structure of the action-outcome space is supported by the hexadirectional modulation of activity patterns in the EC. The six-fold modulation of the amplitude of the BOLD signal had been observed in humans during spatial navigation[47]. However, we did not use the univariate quadrature filter procedure used by Doeller et al.[47], which estimates the preferred putative grid orientation of each voxel on an independent dataset. Instead, we relied on the multivariate pattern similarity modelling approach used by Bellmund et al.[49] and Viganò et al.[35], ideally suited for tasks with limited sampling of the range of direction. The core assumption of this representational similarity approach is the modulation of EC pattern similarity as a function of the angular difference between the sampled directions for each pair of trials (Fig. 3A). Based on this assumption, we divided our sampled directions into two conditions, corresponding to the remainder of modulus $60^0$ of the angle for each direction (Fig. 3B). Specifically, we expected the directions with a remainder of $0^0$ to be more similar to each other than to the other directions. For each run of every participant, we applied the EC ROI mask to the maps of the estimated parameters of interest. The extracted parameter estimates were averaged across runs for each of the 12 distinct directions, and z-score normalisation was subsequently applied to these averaged estimates. After that, we performed RSA on the normalized average parameter values. The pattern similarity between all pairs of directions was calculated using Mahalanobis distance as the similarity measure. This calculation resulted in a neural dissimilarity matrix (DSM) representing the dissimilarities between the directions (Fig. 3C). The DSM was then correlated using Spearman's correlation with the model matrix, which represented the expected dissimilarities of multivariate neural activity patterns across directions. Considering that human grid-like representation, captured using fMRI, is characterized by a difference in neural signals for directions aligned or misaligned with the main axes of the putative grid[27,31,38,47,102,117] and that multivoxel patterns also reveal differences between aligned and misaligned directions[35,49], we hypothesised a positive correlation of our model matrix with the data. To test for the group-level effects, we compared the resulting correlation values against 0 using a one-sided, one-sample t-test (Fig. 3E). Following the studies on conceptual maps that report a lateralization of grid-like representation[34,35,49], we further replicated the analysis for left and right EC separately (Supplementary Fig. 5A). To ensure that the estimated pattern dissimilarities were unbiased, we performed cross-validated RSA using the Crossnobis distance (the cross-validated squared Mahalanobis distance, implemented in rsatoolbox, version 0.1.2) as the dissimilarity measure. Cross-validation was performed across the six runs for each participant. By calculating dissimilarities across independent runs, cross-validation mitigates the effects of noise and overfitting, leading to more reliable estimates of neural representation dissimilarity. The main effect as well as the results from every control analysis remained statistically unchanged (see Supplementary Fig. 4 and see below for

details of the control analyses). To further validate our findings, we conducted a whole-brain searchlight analysis using spheres of radius = 5 voxels, which revealed a cluster in the bilateral hippocampal-entorhinal complex (Supplementary Fig. 5C). Importantly, the sampling of the action-outcome space was carefully balanced to ensure that the mean distance between the start and end positions of each direction was equal across trials and not correlated with direction (see Methods, fMRI tasks).

It is noteworthy that the predicted six-fold modulation of pattern similarity would remain the same when applying a linear transformations of the log odds[118] to the probability values of catching the ball and its visibility. Future research could directly compare the different representations of probabilities to better understand how the brain encodes them within the cognitive map.

**Representational similarity analysis: control analyses.** In order to evaluate whether the modulation of pattern similarity in EC is specific to a sixfold periodicity of grid-like representation[47], we performed a series of control analyses testing a two-, three-, four- and five-fold periodicity of pattern similarity (Fig. 3F). The control periodicities corresponded to a 180°, 120°, 90° and 72° modulation of pattern similarity values. For example, for the four-fold similarity pattern, we divided the sampled directions into conditions based on the remainder of modulus 90° of the angle for each direction. We compared directions with a remainder of 0° against the directions with remainders 30° and 60° (see Supplementary Fig. 13). To address the limitations in sampling constraints of the action-outcome space, where directions originating from the same half of the space often share starting positions, we implemented additional control RDM. These analyses aimed to test whether the six-fold modulation of pattern similarity within the EC was influenced by retrieving action combinations corresponding to the identical starting positions for different directions. To create the control RDM, 12 different vectors were generated, each containing 25 positions. These vectors represented all possible starting positions within the action-outcome space and corresponded to the 12 sampled directions. We then determined how many times each direction would start from each of the 25 possible positions, and assigned these values to the corresponding positions within the individual vector. As a result, each vector denoted a unique code of starting positions for a particular direction, allowing us to compute the pairwise correlations between these vectors and construct a 12x12 control model assessing the similarity between directions. To rule out the possibility that the six-fold modulation of pattern similarity could alternatively have been driven by similar action combinations corresponding to the overlapping ending positions of directions, we created another control 12x12 model matrix. This model was constructed using the same approach as the one used to determine the repetition of the starting positions, but instead included information about the positions in space where the directions would end. In addition, certain directions that shared a combination of both starting and ending positions were more likely to be associated with the same condition in the model capturing the six-fold modulation of pattern similarity. Consider, for example, the 30° and 210° directions. These directions have many positions in common where the starting position of one direction coincides with the ending position of the other, and vice versa. To account for this, we created another model matrix that included information about both the starting and ending positions of the directions. None of the three models showed a significant correlation with the pattern similarity across different directions (see Results). Finally, the control ROIs including HPC, PMC, SMA and LG did not exhibit a six-fold modulation of pattern similarity (Supplementary Fig. 5B).

**Distance-based adaptation to action combinations.** We focused on another distinctive aspect of relational representations within the hippocampal-entorhinal system, namely the presence of a distance representation associated with cognitive maps. This concept implies that when comparing two action combinations, the similarity of their neural representations[119] depends on the proximity of the occupied positions within the abstract action-outcome space. Given the involvement of the hippocampal-entorhinal system in the encoding of Euclidean distances to the navigational goals[120] and following the studies showing the modulation of hippocampal BOLD adaptation response by the distance between the objects[29,32,33,50], we hypothesised that the magnitude of hippocampal-entorhinal activity would adapt as a function of the distance between the action combinations in the action-outcome space. Similar to the previous fMRI analysis, we applied an individual GLM on each participant's functional images from each run of the first fMRI session. The 4 task-related regressors of no interest modelled the presentation of the first and second stimuli of the pairs, the question and participant's response. To test for the adaptation effect, we constructed a runwise parametric regressor, consisting of the demeaned Euclidean distances between pairs of action combinations from each trial. Expected modulation was aligned to the presentation time point and duration of the second stimulus of each pair. In total, each GLM included 51 regressors. We then applied the hippocampal and entorhinal ROI masks to the estimated parameters from each run of every participant. For each mask, we calculated the mean of the extracted parameter estimates across runs, resulting in one value per participant. Building upon prior evidence[29,32,33,50], we had a well-defined hypothesis regarding the direction of BOLD modulation by the distance in the action-outcome space. Consequently, we compared the mean parameter estimates to 0 using a one-sided, one-sample t-test to examine group-level effects (Fig. 4D). Since a more pronounced grid-like representation was observed in the left EC (Supplementary Fig. 5A), we additionally examined the adaptation effect separately for the left and right HPC (Supplementary Fig. 6A). To confirm the effects, we performed a whole-brain analysis. Whole-brain maps were corrected using a false discovery rate (FDR) correction method and reported at a voxel-level threshold of $p < 0.01$, indicating a significant adaptation effect in the bilateral HPC (Fig. 4E).

**Distance-based adaptation to action combinations: control analyses.** We again used the PMC and SMA as our control regions, along with the LG, as we did not expect to see a statistically significant adaptation effect to occur in these regions. Indeed, none of the regions showed any adaptation of the BOLD signal to the distance between stimuli in the abstract action-outcome space (Supplementary Fig. 6B). To explore the possibility that hippocampal adaptation occurs independently for each dimension, rather than in relation to distances within a conjunctive space, we conducted separate analyses for each dimension. Following the steps of the main analysis, we constructed a regressor individually for each dimension, modelling the one-dimensional distance between the stimuli in the abstract action-outcome space. However, no statistically significant effects were observed for either dimension (Dimension 1, probability of catching: $t(45) = 0.09$, $p = 0.471$, 95% CI= [−0.017, 0.018]; Dimension 2, probability of visibility: $t(45) = 1.27$, $p = 0.103$, 95% CI= [−0.005, 0.024]; one-sided t-test). Due to the constraints of the design, the distribution of the longer and shorter distances between the action combinations of the pairs was uneven in some of the runs. This was the result of our decision to assign slightly more long distances than short distances to the cardinal directions in the action-outcome space, in order to match their mean distances to the mean distances of the non-cardinal directions (see Methods, fMRI tasks). To account for the unequal number of short- and long-distance trials, we subsampled and equalized the two types of trials across runs and participants. We iterated the subsampling process 10 times to include the omitted trials in the analysis, and ran a separate GLM for each iteration, repeating the main

analysis on the output of each GLM. The resulting mean parameter estimates were averaged across all iterations for each participant and compared to 0 using a one-sided, one-sample t-test to confirm the main effect (HPC: t(45) = 2.73, $p$ = 0.004, Cohen's d = 0.40, 95% CI= [0.001, 0.012]; one-sided t-test).

**Distance-based adaptation to landmark outcomes.** To get one step closer to understanding how the map-like structure created by participants is generalized, we investigated whether the retrieval and comparison of six coloured balls relies on the action-outcome space, despite not directly requiring the comparison of action combinations. Since the coloured balls represent the landmark outcomes of action combinations (see Methods, Action-outcome space), we hypothesised that these balls should be positioned within the action-outcome space by marking six corresponding positions on it. To test this, we performed a BOLD adaptation analysis using data from the second Comparison Task, structured similarly to the first task. In this task, participants compared the pairs of coloured balls to each other according to their associated probabilities of catching and visibility. We examined the adaptation of the magnitude of BOLD signal as a function of the distance separating pairs of coloured balls in the action-outcome space. We first performed individual GLM analyses on each participant's functional images obtained during each run of the second fMRI session. We included 15 additional task-related regressors of no interest to account for all events occurring during the picture viewing task. The 2 of these regressors modelled the question and participant's response, and a further 12 were dedicated to modelling each stimulus type presented in the pair. Specifically, 6 regressors captured the 6 colours of the balls presented as the first stimulus, while the other 6 accounted for the colours of the balls presented as the second stimulus. This approach allowed us to eliminate any potential bias in BOLD activity attributed to stimulus colour, thus ensuring that the main effect of distance-dependent modulation of the signal was not driven by specific differences induced by colour. To examine the adaptation effect, we incorporated a parametric regressor aligned with the presentation of the second ball of each ball pair, representing the distance between the two balls within the pair. The regressor was demeaned prior to inclusion in the GLM analysis. Each GLM consisted of 61 regressors in total. We followed a similar approach to the previous BOLD adaptation analysis and extracted the estimated parameters using the hippocampal and entorhinal ROI masks. Based on the clear hypothesis about the direction of the BOLD modulation (see previous section), the mean parameter estimates were contrasted against 0 using a one-sided, one-sample t-test to examine group-level effects (Fig. 4F). In addition, the adaptation effect was tested separately for the left and right parts of each mask (Supplementary Fig. 6C). The control ROIs including PMC, SMA, and LG did not show the statistically significant adaptation effect (Supplementary Fig. 6D). An FDR-corrected whole-brain map confirmed the adaptation effect in the HPC (voxel-level threshold of $p$ < 0.01) (Fig. 4G).

**Action similarity-based adaptation.** Our prior analyses successfully identified relation-based map-like representations in the hippocampal-entorhinal system during the comparison of action combinations. This encouraged us to detect additional brain regions supporting the process of action comparison. During the first Comparison Task, the arrow cues on each joystick corresponded to motor actions that participants had been trained to perform in immersive VR, possibly prompting them to use motor representations in order to infer their outcomes. Using the data from the first Comparison Task, we performed the BOLD adaptation analysis on the PMC and SMA, our motor system ROIs. We categorized the trials into distinct conditions based on the shared actions between the two combinations in a pair, creating three trial types. The first type comprised combinations with no overlap across actions (considered dissimilar),

while the second type included trials where only one action was shared (considered similar). Lastly, both actions could overlap across combinations, but in reverse: the action of the first joystick for the first combination was equal to the action of the second joystick for the second combination, and vice versa (considered very similar). To examine the relationship between BOLD signal and action similarity between pairs of action combinations, we created a parametric regressor for each run. We assigned three numerical values (0, 1, and 2) to represent the BOLD response to the three trial types. The parametric regressor was demeaned and matched to the presentation time point and duration of the second stimulus within each pair. Each GLM additionally modelled the 4 regressors of no interest from the task (see Methods, Distance-based adaptation to action combinations), resulting in 51 regressors in total. GLMs were applied to each functional run of every participant from the first fMRI session. Similar to the previous analyses, the PMC and SMA masks were applied to the estimated parameters of interest. As a next step, we examined the resulting effects across all participants. Importantly, we did not formulate a specific hypothesis about the direction of BOLD modulation by similarity between action combinations, as there is no clear consensus in the literature about the repetition effects observed in motor regions[57,58]. To account for the divergent findings, we contrasted the mean estimates for both ROI masks against 0 using a two-sided, one-sample t-test (Fig. 5C). The effect in the SMA was confirmed by the FDR-corrected whole-brain map (voxel-level threshold of $p$ < 0.01) (Fig. 5D).

**Action similarity-based adaptation: control analyses.** The abstract action-outcome space is defined by the relations between action-outcome associations. Consequently, despite the random assignment of outcomes to actions across participants, there is a small inherent correlation between the similarity of action combinations in terms of overlapping actions and the similarity of action combinations in 2D abstract space (mean= 0.091, SD = 0.045). To ensure the independence of these analyses, we ran a single GLM with two parametric regressors separately modelling the modulation of the BOLD signal by motor and 2D abstract information. Within all regions of interest, the results of the original analyses were replicated, suggesting independence of two types of representations (Motor adaptation SMA: t(45)= −2.77; $p$ = 0.005, Cohen's d = −0.40, 95% CI= [−0.028, −0.004]; Motor adaptation HPC: t(45) = 1.11, $p$ = 0.261, 95% CI= [−0.003, 0.013]; two-sided t-test; Abstract distance-based adaptation HPC: t(45) = 2.36, $p$ = 0.01, Cohen's d = 0.34, 95% CI= [0.0008, 0.011]; Abstract distance-based adaptation SMA: t(45) = 0.76, $p$ = 0.217, 95% CI= [−0.003, 0.007]; one-sided t-test). The HPC and EC were chosen as control regions for the action similarity-based BOLD adaptation analysis, assuming that they represent abstract space rather than information about individual actions. Furthermore, the LG was again used as a neutral control region. Indeed, none of the control regions showed the statistically significant repetition effect associated with motor actions (Supplementary Fig. 7).

To introduce variation in the sampling of space (see Methods, fMRI tasks), we created a trial distribution that resulted in some runs having an uneven number of trials with shared and non-shared actions between pairs of action combinations. To ensure that the main effect was not driven by the biases in the distribution, we subsampled the trials. We first excluded a small number of cases where both actions were reversed between pairs of combinations. We then equalized the number of trials with similar and dissimilar action combinations across runs and participants, iterated the subsampling steps 10 times, and performed a separate GLM on every iteration. The parameter estimates from all iterations were averaged for each participant and subjected to a two-sided, one-sample t-test against 0 to confirm the main effect (SMA: t(45)= −3.47, $p$ < 0.001, Cohen's d = −0.51, 95% CI= [−0.037, −0.01]; two-sided t-test).

**Generalized psychophysiological interaction (gPPI) analyses.** As a final step, we used a gPPI analysis[60] implemented in the CONN toolbox (RRID:SCR_009550)[121] to investigate whether the two parallel representations, (i) a hippocampal-entorhinal cognitive map of action-outcome associations and (ii) the individual action-related information within the SMA, interact in the process of comparing action combinations. PPI is a type of functional connectivity analysis identifying the task-based change in correlation of BOLD time-series between the seed region and all other voxels or ROIs. The CONN Toolbox offers a generalized approach to PPI, known as gPPI, forming a separate psychological regressor for each condition within a single model. This approach has been shown to be more powerful compared to standard PPI analysis, improving the fit of the regression model to the fMRI data[60,122]. Building on our previous analysis showing that the HPC adapts its response according to the distances between action combinations within the action-outcome space, we aimed to investigate whether its connectivity with the SMA also shows a similar modulation in response to the task conditions of the first Comparison Task. As we observed the adaptation effect predominantly in the left hippocampal ROI, we used a lateralized mask for our analysis. The task trials were divided into two conditions: condition 1 comprised trials with long distances, while condition 2 consisted of trials with short distances between pairs of action combinations. The preprocessed data was imported into the Toolbox together with the hippocampal mask. We included 46 nuisance regressors derived from fMRIprep (see Methods, General linear models (GLMs)), along with the two main task-related conditions, in the denoising model to address any potential sources of confounding effects. Furthermore, temporal high-pass filtering was applied to remove signal fluctuations below 0.01 Hz. The first-level model included a total of five regressors (Fig. 6A), where (i) two of these regressors represented the task conditions, serving as the two main psychological factors. (ii) Another regressor captured the BOLD time series data from the seed ROI, representing the main physiological factor. (iii) The remaining two regressors modelled the PPI term as the product of (i) and (ii), capturing the interaction term within the analysis. To make a statistical inference on the difference in functional connectivity between the long and short distance conditions at the group level, we used the parameter estimates derived from the first level analysis. Specifically, for each participant, we contrasted the whole-brain map of the interaction term between condition 1 and condition 2 within the toolbox. We then used the SMA mask to extract the parameter estimates from the individual contrast images, aiming to assess the connectivity with the left hippocampal ROI. As the PPI method lacks directional information, we repeated the analysis, this time using the SMA as the seed ROI and obtained the contrasted parameter estimates from the left hippocampal region. Combining the results from both ROIs as seeds, we averaged the contrasts from two analyses for each participant and subjected the average estimates to a two-sided, one-sample t-test against 0 (Fig. 6C). To test for the connectivity effect for both ROIs separately, we contrasted the results of each analysis against 0, replicating the main result (Supplementary Fig. 9A, B). Notably, the LG, our control region, showed no statistically significant task-related effect on the connectivity with either ROI (Supplementary Fig. 9C, D). The whole brain maps of the parameter estimates of each ROI (uncorrected for multiple comparisons; statistical significance threshold defined at single voxel level ($z_\alpha = 1.96$)), confirm the task-based modulation of connectivity between HPC and SMA (Supplementary Fig. 9E, F).

**Reporting summary**
Further information on research design is available in the Nature Portfolio Reporting Summary linked to this article.

## Data availability
Raw data are protected and are not available due to data privacy. Preprocessed data will be made available upon request to the corresponding author. The processed fMRI and behavioural data to reproduce the statistical analyses reported in this paper are available on the Open Science Framework https://doi.org/10.17605/OSF.IO/UZ83D[123]. Source data are provided as a Source Data file. Source data are provided with this paper.

## Code availability
Analysis code is available on the Open Science Framework https://doi.org/10.17605/OSF.IO/UZ83D[123].

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

## Acknowledgements

We would like to thank Kerstin Schumer, Danielle Schewe, Max Schulz, Annika Schmitt, Anke Kummer, Simone Wipper, Sylvie Neubert and Mandy Jochemko for help with data collection. We would also like to thank Toralf Mildner, Joeran Lepsien and colleagues of the Doellerlab for providing and discussing the MRI sequence. The key behavioural analyses of this study were developed following previous work of Jacob L.S. Bellmund on cognitive representation of space, whom we would like to thank for helpful input. We would also like to thank all colleagues of the Doellerlab for fruitful discussions of the study. CFD's research is supported by the Max Planck Society, the European Research Council (ERC-CoG GEOCOG 724836), the Kavli Foundation, the Jebsen Foundation, Helse Midt Norge and The Research Council of Norway (223262/F50, 197467/F50). PH was supported by a Reimar-Lüst Prize from the Humboldt Foundation and Thyssen Foundation.

## Author contributions

I.B. and C.F.D. conceived the initial Idea. I.B. and C.F.D. designed the experiment and developed the tasks with the inputs from D.R. and P.H. I.B. made the VR environment. I.B. acquired the data. I.B., S.V., D.R., P.H. and C.F.D. planned the analyses. I.B. performed the analyses and all authors discussed the results. I.B. wrote the initial draft and all authors finalized the manuscript.

## Funding

## Competing interests

The authors declare no competing interests.
