## [Transparent Peer Review file · Nature Communications]

Hippocampal-entorhinal cognitive maps and cortical motor system represent action plans and their outcomes

Corresponding Author: Ms Irina Barnaveli

Version 0:

Reviewer comments:

Reviewer #1

(Remarks to the Author)

This is a real tour de force study that examined the encoding of action outcomes, testing the hypothesis that subjects learn arbitrary action-outcome associations by building a cognitive map represented in the hippocampal-entorhinal cortical system. We were very impressed with the careful attention paid to all experimental details, the cleverness and uniqueness of the study design, the wide range of analytic controls used, and the overall sophistication of both the behavioral and neuroimaging analyses. The paper is well-written and relatively easy to follow, which is no small feat given the complexity of the experimental design and analytic approaches. We feel that this will be a very important paper for the cognitive neuroscience research community, as well as researchers interested in motor learning and control.

Below, we highlight some important considerations for the authors to address in revising their paper. We believe that addressing these points will further strengthen this already impressive study.

1. The authors evaluate the similarity of action-outcome pairs based on probabilities of catching/visibility, but evidence from other domains (e.g., Zhang et al.) suggests the brain instead encodes something akin to log-odds. The experiment as designed makes it difficult to disentangle these two, since the angular difference between these two parametrizations are generally similar over the experienced probabilities. Still, there are several action-outcomes pairs with identical angles in probability space, but with differing angles in log-odds space that might be used to disentangle the two possibilities. For example, taking all action-outcome pairs with an angle of 45-deg in probability space, these same pairs range from 32-deg to 60-deg in log-odds space. A straightforward prediction would be that, if the entorhinal-hippocampal system is representing log-odds rather than probability, then the neural activity of those pairs with identical directions in probability space, but which are closer to 60-deg in log-odds space, should be more similar than those which are closer to 30-deg. If this is true, then an analysis using the log-odds would likely be more powerful (and would clarify the exact nature of the neural representation). We suggest analyzing these pairs to potentially clarify the exact nature of the neural representation and possibly increase the power of the analysis.

2. The finding of BOLD adaptation to distance in action-outcome space in orbitofrontal cortex (OFC) is consistent with literature supporting the encoding of (value-weighted) cognitive maps in OFC (e.g., Wilson et al.) and the known direct coupling of OFC-HPC. We recommend discussing this finding, as it's consistent with the known role of HPC-OFC connectivity in the use of these maps in goal-directed behavior.

3. The motivation for analyzing different periods of the task for hexagonal coding (reflection period) versus proximity coding (stimulus presentation periods) was not clear and not well setup during the Results section (which is naturally read first than the Methods). Additionally, please clarify why these different periods of the task necessitated an RSA-approach (hexagonal coding) versus an adaptation-related approach (proximity coding). For example, why wouldn't the proximity coding (or the SMA encoding of individual actions effects) also be present using RSA-related methods, and did the authors test this? Clarification on these points would be appreciated.

4. The use of normality tests to decide whether to perform a t-test or some non-parametric equivalent is known to result in inflated error rates (e.g. Shamsudheen and Hennig). The mixture of t-tests and Wilcoxon's in particular makes it difficult to compare the different tests (e.g. line 317; a positive t-test vs. a negative Wilcoxon), since they test different hypotheses (i.e. means vs. stochastic equality -- not medians, since many of the variables being compared are bounded, and so cannot be

location shifts of each other). It might be better just to use the Wilcoxon as a default, if assuming normality is unrealistic.

5. Please add methodological details concerning the implementation of the 2-, 3-, 4- and 5-fold control analyses to the methods section. We could not find specific details regarding these analyses.

6. The authors should consider toning down the strong action planning/motor control angle to the study (or at least better clarifying what they show). Indeed, while it is true that the action-outcome space was defined by motor action learning in the VR setup, at the actual time of fMRI testing, subjects are merely making perceptual judgements (and not performing the actions). As such, this indicates that the form of encoding – during the fMRI task -- is much more explicit in nature, and indeed the behavior seems to support this. This slightly undermines some of the claims being made in the discussion about the novelty of this work with respect to juxtaposing it with the idea that action-outcome associations are predominantly procedural/implicit in nature. Specifically, there is already quite a bit of work in the motor domain showing that motor processes/learning have a highly explicit component (e.g., Taylor et al., 2014, Areshenkoff et al., 2022 *elife*, Krakauer et al., 2019; etc.)

References:

- Shamsudheen, I., & Hennig, C. (2023). Should we test the model assumptions before running a model-based test?. *Journal of Data Science, Statistics, and Visualisation*, 3(3).
- Wilson, R. C., Takahashi, Y. K., Schoenbaum, G., & Niv, Y. (2014). Orbitofrontal cortex as a cognitive map of task space. *Neuron*, 81(2), 267-279.
- Zhang, H., & Maloney, L. T. (2012). Ubiquitous log odds: a common representation of probability and frequency distortion in perception, action, and cognition. *Frontiers in neuroscience*, 6, 1.OFC
- Areshenkoff, Corson, Daniel J. Gale, Dominic Standage, Joseph Y. Nashed, J. Randall Flanagan, and Jason P. Gallivan. 2022. "Neural Excursions from Manifold Structure Explain Patterns of Learning during Human Sensorimotor Adaptation." *eLife* 11 (April). <https://doi.org/10.7554/eLife.74591>.
- Krakauer, John W., Alkis M. Hadjiosif, Jing Xu, Aaron L. Wong, and Adrian M. Haith. 2019. "Motor Learning." *Comprehensive Physiology* 9 (2): 613–63.
- Taylor, Jordan A., John W. Krakauer, and Richard B. Ivry. 2014. "Explicit and Implicit Contributions to Learning in a Sensorimotor Adaptation Task." *The Journal of Neuroscience: The Official Journal of the Society for Neuroscience* 34 (8): 3023–32.

(Remarks on code availability)

I could not actually find any reference to the code (presumably on github) anywhere in the paper.

Reviewer #2

(Remarks to the Author)

This well-written paper from Barnaveli et al. uses fMRI and VR to show that when multiple action-outcome mappings need to be related to one another, instead of only being represented as distinct, one-to-one mappings between action and outcome, or some weighted mixture, 'motor primitives' are instead organized in a relational 'action-outcome' cognitive map in the EC–HC system. Specifically, they find a grid-like representation in EC for vectors between the two action-outcome/action-outcome combinations required to be evaluated and compared in their task. They then show that BOLD adaptation in HC and EC (as well as OFC and other regions) is proportional to the distance between outcomes in the 2D space in both experimental tasks. By contrast, the SMA shows BOLD enhancement dependent on the repetition of a specific action, potentially suggesting the representation of specific actions (though see below). The HC and SMA exhibit increased functional connectivity in this task when actions evaluated are more distinct in this action-outcome space. The analyses in this paper are generally rigorous and well-controlled. The paper makes a significant contribution by extending understanding of action-outcome mapping and evaluation, and, in particular, elucidates an unexpected role of the HC–EC system in flexible action planning or relational evaluation. Below are some important questions and concerns regarding analyses and their interpretation that should be addressed to support publication:

1. An important question concerns how generalizable these findings are. Does the inclusion of outcomes (specific colored balls) or that task behavior explicitly required evaluating relations between action-outcome mappings on both dimensions for comparisons change the way the brain solves the task, making it more likely to incorporate the HC-EC system? In other words would the HC-EC system map this space if participants simply had to plan the actions to obtain the desired outcomes without any task requirement to relate them to one another?

2. I am unsure about the interpretation of findings concerning the role of the HC-EC system in this task and its interaction with the SMA. It would seem that the findings are likewise consistent with the view that it is the SMA that first simulates a forward model of the actions, permitting the HC-EC system to localize the outcome in the 2D map, and then to compare them using a vector comparison. Or it could be that the cognitive map has been formed in the HC-EC system, and the mappings are retrieved and simulated from this map, as suggested. Unless data can speak to this distinction, the alternative interpretation should at least be discussed.

3. The task design included specific colored ball outcomes, referred to as landmarks, that had to be evaluated in relation to one another in one task. Were there landmark effects, in terms of better performance or attraction, neurally or behaviorally? For instance, were landmark locations more decodable? Or is there evidence of distortion in distances towards the anchors?

4. For the RSA approach, there does not appear to be any cross-validation. Were on and off grid (similar and dissimilar) angles randomly distributed within each run? This is important to demonstrate due to potential differences in auto-correlation if the distributions are not matched.
5. It is stated that the mean distances across angles was similar but I did not find an accompanying figure or table. This is an important control. What is the distribution of distances across angles?
6. I could not find a whole-brain map of the grid-like RSA results. Were there grid-like effects in mPFC and PCC, as found in several previous studies (Doeller et al., 2010; Constantinescu et al., 2016; Bao et al., 2019; Park et al., 2022; Veselic et al., bioRxiv, 2024)? The medial wall isn't visible in Fig. S4C.
7. In the distance adaptation analysis, in addition to HC there is prominent adaptation in lateral OFC. This is interesting in light of the OFC literature when considering these pairings are associated with distinct predictions about outcomes (visibility and catch p) (see e.g. O'Reilly et al., PLoS Biology, 2013).
8. How can we be certain SMA is representing 'action plans' per se without using reverse inference? Outcomes and visual stimuli are highly correlated with the specific action plan. Can these be decoupled in the design? For example, would this be possible to dissociate by restricting analysis to trials where the action was the same but the joystick identity and associated outcome differed (i.e. when they switched joysticks)?
9. For the action similarity-based adaptation analysis: Is there evidence for adaptation when two of the same actions are presented for the same option? E.g. a left, left option? Similarly, this brings up the possibility of more than 3 trial types (e.g., if left is used twice in the first option and once in the second option), or were those trials from this analysis?
10. I did not follow the logic for why the correlation between HC and SMA would be predicted to increase with the distance between action-outcome points. The motivation needs to be better articulated.
11. The adaptation results in Fig 4 and supplementary tables of significant clusters use FDR correction. While this is a bit unconventional, the t-stats look respectable. The authors should upload their contrast images (I assume this will be done at some stage).
12. Are the example fits for the MDS and Procrustes analysis from day one or day two? It would be interesting to see what these fits look like for a good, bad, and median participant from day 2 to get a better sense of how accurately these spaces are represented.

(Remarks on code availability)

Reviewer #3

(Remarks to the Author)

I enjoyed reading this manuscript arguing for a form of abstract coding of actions in two parts of the medial temporal lobe. The results are both novel and compelling and they will be of broad interest to cognitive scientists, neuroscientists, and motor scientists not least because they challenge the prevailing belief that such medial temporal lobe structures are not concerned with action representation (although as the authors correctly note, the importance of the medial temporal lobe is indicated by often overlooked studies such as Brasted et al., *Brain*, 2003). The analyses that have been used are ones that are relatively novel but they have now been used in other contexts and there is broad agreement on how they should be performed.

1. The right hand panels of both parts A and B of figure 2 each contain keys with elements called "critical distance". I am not sure what this label means. It looks as if there are lines that correspond to these labels in right hand panel A but not in right hand panel B? Is that right? Are they, however, overlapping with one another in panel A? Is there perhaps a line in panel B but which overlaps with the continuous dark blue/purple line? I wonder if a revised manuscript might clarify what is being shown in the two right hand panels and also explain why it is being shown and how it relates to the interpretation of the results?

2. Figure 3A. I think that it might help some readers to make clear in the legend for figure 3a that what is being shown is the trajectory in the abstract action representation space between one action and the second action when they are presented sequentially one after the other in the action comparison task. I realise that this is in the main text but it would be easy to come away from the figure 3A and its legend thinking that it is something about the trajectory of a given action on each trial that is being analysed.

3. Line 272, figures 3 and supplementary 4. Is there any evidence of hexadirectional grid-like encoding in anterior medial prefrontal cortex? While the focus on the entorhinal cortex in the current report is clearly warranted, there is evidence from some of the authors' own previous work (Doeller et al., *Nature*, 2010) as well as the work of others (Constantinescu et al., 2016 *Science*, Bongioanni et al., *Nature*, 2021) that such patterns of activity are sometimes almost more easily identifiable in anterior medial prefrontal cortex. It is difficult to assess this from the supplementary figure because of the placement of the

sections.

4. Line 450-470, figure 6. I think that the abstract motor space in which the PPI is conducted is the same one that is explored in figure 4 as opposed to the ones explored in figures 3 or 5. If that is the case then maybe that could be made clearer somewhere in lines 450-470. On a related note, I wondered if although connectivity between hippocampus and motor association reflects the abstract motor space, does connectivity between different motor association regions reflect the action similarity space explored in figure 5? In other words, while PPIs between hippocampus and motor association areas in premotor and parietal cortex might reflect the abstract motor space, do the motor association areas themselves show evidence, in a PPI, of interacting with one another in the same abstract space (illustrated in figure 4) or the motor similarity space (illustrated in figure 5)?

5. Notwithstanding the novelty of the current report, some mention of the pioneering work of Passingham, Petrides, and Murray and Wise on conditional motor mappings / arbitrary sensorimotor mappings, as opposed to reaching and grasping seems necessary.

(Remarks on code availability)

Reviewer #4

(Remarks to the Author)

(Remarks on code availability)

Version 1:

Reviewer comments:

Reviewer #1

(Remarks to the Author)

The authors have done a thorough job of responding to my comments with their revision.

(Remarks on code availability)

Reviewer #2

(Remarks to the Author)

The authors have thoroughly addressed my questions and comments with appropriate new analyses and revisions to their manuscript. I hope they feel this has strengthened their paper. I have no further comments and wish to congratulate them.

(Remarks on code availability)

Reviewer #3

(Remarks to the Author)

The authors have addressed all of the concerns that I raised very carefully and convincingly.

(Remarks on code availability)

Reviewer #4

(Remarks to the Author)

(Remarks on code availability)

Reviewer #5

(Remarks to the Author)

(Remarks on code availability)

We thank the reviewers for their general enthusiasm for our study and for their insightful comments. Our responses are written in blue and the corrections in the manuscript are displayed in bold font.

REVIEWER COMMENTS

Reviewer #1 (Remarks to the Author):

This is a real tour de force study that examined the encoding of action outcomes, testing the hypothesis that subjects learn arbitrary action-outcome associations by building a cognitive map represented in the hippocampal-entorhinal cortical system. We were very impressed with the careful attention paid to all experimental details, the cleverness and uniqueness of the study design, the wide range of analytic controls used, and the overall sophistication of both the behavioral and neuroimaging analyses. The paper is well-written and relatively easy to follow, which is no small feat given the complexity of the experimental design and analytic approaches. We feel that this will be a very important paper for the cognitive neuroscience research community, as well as researchers interested in motor learning and control.

Below, we highlight some important considerations for the authors to address in revising their paper. We believe that addressing these points will further strengthen this already impressive study.

We thank the Reviewer for the interest in our manuscript and for the very positive feedback! Below please find our point-by-point responses to the comments raised by the Reviewer.

1. The authors evaluate the similarity of action-outcome pairs based on probabilities of catching/visibility, but evidence from other domains (e.g., Zhang et al.) suggests the brain instead encodes something akin to log-odds. The experiment as designed makes it difficult to disentangle these two, since the angular difference between these two parametrizations are generally similar over the experienced probabilities. Still, there are several action-outcomes pairs with identical angles in probability space, but with differing angles in log-odds space that might be used to disentangle the two possibilities. For example, taking all action-outcome pairs with an angle of 45-deg in probability space, these same pairs range from 32-deg to 60-deg in log-odds space. A straightforward prediction would be that, if the entorhinal-hippocampal system is representing log-odds rather than probability, then the neural activity of those pairs with identical directions in probability space, but which are closer to 60- deg in log-odds space, should be more similar than those which are closer to 30-deg. If this is true, then an analysis using the log-odds would likely be more powerful (and would clarify the exact nature of the neural representation). We suggest analyzing these pairs to potentially clarify the exact nature of the neural representation and possibly increase the power of the analysis.

We thank the reviewer for the insightful comment regarding the potential use of log-odds in evaluating the similarity of action-outcome pairs. The reviewer suggests using the multiple action-outcome pairs with identical directions in the action-outcome space and analyse them in the log-odds space, since the same action-outcome pairs might have different directions in the log-odds space. More precisely, the analysis should focus on the action-outcome pairs underlying the identical direction in the action-outcome space, but that would range from 30 to 60 degrees in log-odds space, allowing us to test for hexadirectional (6-fold) symmetry in the log-odds space. We appreciate the suggestion to explore this alternative representation. However, there are several limitations in applying this approach to our current analysis. We will address these limitations below.

We first created a log-odds space using the formula from Zhang & Maloney (2012), transforming the probabilities of action-outcome space to log-odds values. The resulting probabilities were the following: -2.197, -0.847, 0, 0.847, 2.197. As the reviewer suggests, the 45-degree directions in the action-outcome space would be ideal candidate directions to test for a hexadirectional symmetry in the log-odds space, where they would range from 32 to 60 degrees. Unfortunately, our fMRI task does not include action-outcome pairs with 45-degree directions in the action-outcome probability space. Thus, we explored alternative directions. The cardinal directions (0, 90, 180, 270 degrees) remain unchanged between our probability space and log-odds spaces, making them unsuitable for distinguishing between the two representations. All other directions, sampled in the action-outcome space, have a small and thereby insufficient range in degrees in the log-odds space, making the hexadirectional analysis unfeasible. For instance, the directions that are 30 degrees and their 'mirror' directions (150, 210, 330 degrees) in our action-outcome space would range from approximately 21 to 38 degrees in the log-odds space, indicating that they would still be closer to 30 degrees in the log-odds space. Similarly, the directions at 60 degrees and their mirror directions (120, 240, 300 degrees) would range from approximately 51 to 68 degrees in log-odds space, still being closer to 60 degrees in the log-odds space.

In summary, to effectively apply a hexadirectional analysis to directions that are identical in the action-outcome space but different in the log-odds space, the deviations in the log-odds space from the original degree must be more than 15 degrees. However, our directions could have a maximum deviation of 9 degrees, showing low variability. While the suggestion to analyse the directions using log-odds is intriguing, the current design of our study does not allow for a clear distinction between the two spaces. We thank the reviewer for raising this point and we will consider this possibility for future research to potentially improve the power and clarity of our analyses. We now address this topic in the Methods section as follows (see page 37 of the revised manuscript):

“It is noteworthy that the predicted six-fold modulation of pattern similarity would remain the same when applying a linear transformations of the log odds (Zhang et al., 2012) to the probability values of catching the ball and its visibility. Future research could directly compare the different representations of probabilities to better understand how the brain encodes them within the cognitive map.”

2. The finding of BOLD adaptation to distance in action-outcome space in orbitofrontal cortex (OFC) is consistent with literature supporting the encoding of (value-weighted) cognitive maps in OFC (e.g., Wilson et al.) and the known direct coupling of OFC-HPC. We recommend discussing this finding, as it's consistent with the known role of HPC-OFC connectivity in the use of these maps in goal-directed behavior.

We thank the reviewer for the comment. We agree with them that our results are consistent with the critical involvement of both the HPC and OFC in the construction/use of these map-like representations. This was also pointed out by reviewers 2 and 3. We now address this finding in the Discussion section as follows (see pages 19 and 20 of the revised manuscript):

“Evidence for a cognitive map representation in mPFC

In addition to the hippocampal-entorhinal system, our analyses also revealed the involvement of the medial prefrontal cortex (mPFC) and the lateral orbitofrontal cortex (lOFC) in the representation of the abstract action-outcome space (see Supplementary Figure 5C and

Supplementary Tables 1 and 2). Neural representations consistent with a cognitive maps have been found in the mPFC in humans (Doeller et al., Nature, 2010; Constantinescu et al., 2016) and non-human primates (Bongioanni et al., Nature, 2021). IOFC has been shown to be specifically involved in the construction of such a map (e.g., Costa et al., 2023, rodent study), perhaps reflecting the direct projections from the hippocampus to these prefrontal regions (Reznik et al., 2024; Eichenbaum, 2017; Preston & Eichenbaum, 2013). Other work has suggested that the OFC, particularly its lateral part, represents the expected outcomes based on environmental statistics (O'Reilly et al., 2013) in many tasks (Rudebeck & Murray, 2014) and is involved in disambiguating the unobservable states in a cognitive map of task space (Wilson et al., 2014). The mPFC and OFC, together with the hippocampal-entorhinal system, could therefore participate in acquiring and structuring associative representations (Stalnaker et al., 2015). These representations may play an important role in guiding behaviour when multiple action alternatives are possible.”

3. The motivation for analyzing different periods of the task for hexagonal coding (reflection period) versus proximity coding (stimulus presentation periods) was not clear and not well setup during the Results section (which is naturally read first than the Methods). Additionally, please clarify why these different periods of the task necessitated an RSA-approach (hexagonal coding) versus an adaptation- related approach (proximity coding).). For example, why wouldn't the proximity coding (or the SMA encoding of individual actions effects) also be present using RSA-related methods, and did the authors test this? Clarification on these points would be appreciated.

The reviewer is asking to clarify the rationale behind two points: (1) why did we select two different analytical approaches for our research questions? And (2) why these approaches necessitated analyzing different periods of the task? We thank the reviewer and address these points in turn.

1. We used RSA for detecting the hexadirectional signal and adaptation for the distance-modulated activity because these are different questions, dealing with two distinct readouts from the underlying cognitive map. Our main question addressed whether participants developed relational representations in the hippocampal-entorhinal system, usually interpreted as signatures of 'cognitive mapping' in the spatial memory literature. To answer this question, we tested for the existence of specific neural representations associated with such maps, a hexadirectional (6-fold) modulation of the activity in entorhinal cortex and a distance-based modulation of the activity in the hippocampus. To investigate the grid-like hexadirectional signal, we relied on the RSA approach, which assumes that EC activity patterns would be more similar for pairs of directions that are multiple of 60°, compared to those that are multiple of 30°, in line with the 60° rotational symmetry typical of grid-cells. This approach has been proven to be powerful and flexible, as it does not require the estimation of putative "grid orientations" and it is known to be suited for tasks with limited sampling of the range of directions (Bellmund et al., 2016; Bao et al., 2019; Viganò et al., 2021, 2023). fMRI adaptation, in contrast, is less suited for hexadirectional analyses (cf. Doeller et al, 2010).

On the other hand, for testing distance-like representations, we opted for an adaptation approach because the evidence for distance representations using RSA is less established. For instance, in their study, Morgan et al., 2011 reported distance-modulated activity in the hippocampus using adaptation and also showed that RSA was not sensitive to such information. Other studies have indeed relied on adaptation (repetition suppression) for investigating distance-like representations (e.g., Theves et al. 2020; Viganò & Piazza 2020; Viganò et al. 2021). The repetition suppression approach has also been successfully used to discriminate between similar and dissimilar motor

actions (e.g., Chouinard & Goodale, 2009; Persichetti et al., 2020), motivating us to apply this analysis to access the individual action-outcome representations in the SMA. Thus, we decided to optimize our tasks for the adaptation approach, allowing us to investigate both distance-based and motor similarity-based modulation of BOLD activity.

2. Our analyses required two distinct, albeit adjacent, periods of the task due to the different nature of the information we were trying to capture with each method. Grid-RSA was performed during the reflection period when no visual stimuli were present on the screen because we were interested in abstract directional information rather than stimulus-bound representations: grid-like activity is thought to represent abstract relational information capturing the structure of the spatial or conceptual environment (e.g., Behrens et al. 2018). Moreover, previous studies have reported grid-like activity in entorhinal cortex (and sometimes beyond) during imagination periods, when people were prompted with longer time frames of relational reasoning between stimuli held in mind (Constantinescu et al., 2016; Bellmund et al., 2016; Bao et al., 2019; Viganò et al., 2021, 2023). Following this, we specifically instructed participants to start comparing the two stimuli of the trial in terms of their outcomes *only during* a reflection period and thus we expected that any directional information should be available principally in this time frame.

Conversely, the BOLD adaptation approach was applied during the presentation of the second stimulus of each pair, because the focus was on the similarities (corresponding to the different levels of adaptation as illustrated in the previous point) *between the stimuli themselves*, here conceived as specific positions in action-outcome space, as opposed to the underlying structure of the space itself. The type of representation we aimed for here, following previous studies (e.g. Morgan et al. 2011; Theves et al. 2020; Viganò et al. 2021), was that of the distance between the presented action combinations or coloured balls. Thus, we directly modelled the time window of stimulus presentation, as done in the abovementioned studies.

In addition to the methods section, we now clarify the decision to analyse different task periods in the Results section of the manuscript, changing the legends of Figure 1, Figure 3 and Figure 4 in the following way (the changed text is marked in bold):

*Figure 1: Experimental design and cognitive tasks. ...In the two Comparison Tasks (H), participants were sequentially presented with pairs of action combinations (Comparison Task 1) or coloured balls (Comparison Task 2). **On each trial, participants were instructed to consider the associated outcomes of the stimuli of the pair during the reflection period. They then answered the subsequent question or statement on how the outcomes changed from one stimulus to the other.** The two Comparison Tasks were performed in the scanner on the third day. | Behavioural performance in the scanner for the two Comparison Tasks. Each dot represents data from an individual participant.*

*Figure 3: Grid-like representation of the abstract action-outcome space. A,B,C Logic of analysis. **The marked positions in the abstract action-outcome space correspond to different combinations of actions presented sequentially as pairs in the first Comparison Task. Each arrow corresponds to a direction in this abstract space, relating the first to the second action combination of a pair.** The action- outcome space was sampled in 30^0 steps using 12 different directions in the first Comparison Task (A). **To capture the directional information, we modeled the reflection phase of the task, where participants assessed the change in outcomes from***

one action combination to the other. Assuming grid-like activity in the entorhinal cortex (EC), we anticipated greater similarity between fMRI patterns for directions differing by multiples of 60° (with a remainder of 0° when dividing their angular difference by 60°) compared to directions whose angular difference results in a remainder of 30° (see Methods).

Figure 4: Distance representations of the abstract action-outcome space. **A,B,C** Logic of analysis. The BOLD response in the hippocampal-entorhinal system was expected to show different levels of adaptation depending on the distance between the pair of stimuli in the action-outcome space. In particular, we predicted stronger adaptation for such trials where the two action combinations (B) or two coloured balls (C) of a pair are located closer to each other in the abstract 2D space, compared to when they are located farther apart. **We modeled the presentation of the second stimulus in each pair, conceiving them as positions within the action-outcome space.**

4. The use of normality tests to decide whether to perform a t-test or some non-parametric equivalent is known to result in inflated error rates (e.g. Shamsudheen and Hennig). The mixture of t-tests and Wilcoxon tests in particular makes it difficult to compare the different tests (e.g. line 317; a positive t-test vs. a negative Wilcoxon), since they test different hypotheses (i.e. means vs. stochastic equality -- not medians, since many of the variables being compared are bounded, and so cannot be location shifts of each other). It might be better just to use the Wilcoxon as a default, if assuming normality is unrealistic.

We agree with the reviewer that using different tests makes it difficult to compare results across different analyses. Since we already know that most of our data is normally distributed, permutation tests can be more powerful than Wilcoxon tests as they do not make assumptions about the distribution of the data. For statistical rigor, we decided to perform group-level statistics using t-tests based on a non-parametric permutation-based approach to assess significance. For this, we compared the observed t-values against a surrogate distribution obtained from 10,000 random sign-flips. The non-parametric tests were performed against 0 when a single condition was considered, or to assess within-participant differences between conditions. The results remained statistically unchanged and are now reported in the revised version of the manuscript. Below, we provide the results of the main analyses using the Wilcoxon test to demonstrate to the reviewer that all effects remain statistically significant in this case as well.

Referring to Figure 2:

The estimated pairwise similarity scores of action combinations and coloured balls from the two Rating Tasks were correlated with the distances between their positions in the action-outcome space.

Day 1

Task 1: $z(45) = 5.73$, $p < 10^{-4}$; two-tailed Wilcoxon signed rank test

Task 2: $z(45) = 5.719$, $p < 10^{-4}$; two-tailed Wilcoxon signed rank test

Day 2

Task 1: $z(45) = 5.839$, $p < 10^{-4}$; two-tailed Wilcoxon signed rank test

Task 2: $z(45) = 5.905$, $p < 10^{-4}$; two-tailed Wilcoxon signed rank test

Referring to Figure 3:

The significant 6-fold periodicity of pattern similarity in the bilateral EC ($z(45)= 2.086$, $p= 0.036$; one-tailed Wilcoxon signed rank test).

Referring to Figure 4:

The significant fMRI BOLD adaptation in the HPC ($z(45)= 2.657$, $p= 0.015$; one-tailed Wilcoxon signed rank test), but not in the EC ($z(45)= 2.081$, $p= 0.074$; one-tailed Wilcoxon signed rank test) for the first Comparison Task.

The significant fMRI BOLD adaptation in the HPC and the EC for the second Comparison Task (HPC: $z(45)= 2.936$, $p= 0.006$; EC: $z(45)= 2.334$, $p= 0.038$; one-tailed Wilcoxon signed rank tests).

Referring to Figure 5:

A significant *increase* in activation within the SMA, but not in the PMC, in the trials with shared actions across two action combinations (SMA: $z(45) = 2.845$, $p=0.008$; PMC: $z(45)= 1.826$, $p= 0.067$; two-tailed Wilcoxon signed rank tests).

Referring to Figure 6:

Functional connectivity between the left HPC and SMA increased as a function of distance between action combinations in the action-outcome space ($z(45) = 2.354$, $p= 0.017$; two-tailed Wilcoxon signed rank test).

Revised methods:

*The statistical significance of the effects was determined using **t-tests based on a non-parametric permutation-based approach. The observed t-values were compared against a surrogate distribution obtained from 10000 random sign-flips. The non-parametric tests were performed against 0 when a single condition was considered, or to assess within-participant differences between conditions.***

5. Please add methodological details concerning the implementation of the 2-, 3-, 4- and 5-fold control analyses to the methods section. We could not find specific details regarding these analyses.

We apologize to the reviewer for not clarifying the details of the control periodicities and have added these to the Methods section of the manuscript and as Supplementary Figure 13.

Revised methods:

*In order to evaluate whether the modulation of pattern similarity in EC is specific to a sixfold periodicity of grid-like representation⁴⁵, we performed a series of control analyses testing a two-, three-, four- and five-fold periodicity of pattern similarity (Figure 3F). **The control periodicities corresponded to a 180°, 120°, 90° and 72° modulation of pattern similarity values. For example, for the four-fold similarity pattern, we divided the sampled directions into conditions based on the remainder of modulus 90° of the angle for each direction. We compared directions with a remainder of 0° against the directions with remainders 30° and 60° (see Supplementary Figure 13).***

Model RDMs

Periodicities

Supplementary Figure 13. Different models of pattern similarity values. To test for the representation of directions in the abstract action-outcome space, we created eight different models of pattern similarity between pairs of trials sampling different directions in the first Comparison Task. Each entry on the modeled representational dissimilarity matrices (RDMs) corresponds to a direction in the abstract space, indicated by a small black arrow, relating the first to the second action combination of a pair (see Figure 3A and Methods). We hypothesized that pattern similarity would follow a 6-fold periodicity in the entorhinal cortex, while the other models served as control periodicities for the analysis. The modeled periodicities correspond to a 180°, 120°, 90°, 72°, 60°, 51.43° and 45° modulation of pattern similarity values (see Figure 3 for results).

6. The authors should consider toning down the strong action planning/motor control angle to the study (or at least better clarifying what they show). Indeed, while it is true that the action-outcome space was defined by motor action learning in the VR setup, at the actual time of fMRI testing, subjects are merely making perceptual judgements (and not performing the actions). As such, this indicates that the form of encoding – during the fMRI task -- is much more explicit in nature, and indeed the behavior seems to support this. This slightly undermines some of the claims being made in the discussion about the novelty of this work with respect to juxtaposing it with the idea that action-outcome associations are predominantly procedural/implicit in nature. Specifically, there is already quite a bit of work in the motor domain showing that motor processes/learning have a highly explicit component (e.g., Taylor et al., 2014, Areshenkoff et al., 2022 elife, Krakauer et al., 2019; etc.)

We thank the reviewer for the important comments regarding the action planning and motor control angle of our study and the need for further clarification. The reviewer's core arguments are that (1) because our readout task involved perceptual judgement, the initial encoding of action-outcome associations must have been explicit, and (2) therefore, the work is not particularly different or novel from previous studies of mental representation. We respectfully question argument (1) – it is common to use perceptual judgements as a source of information about how actions are represented (for example, the effects of actions on temporal experience have only been studied through their secondary effects on perceptual judgement (Hagura et al., 2017; Stetson et al., 2006). Regarding argument (2), the point raised by the reviewer is that our main measures do not involve actions and motor performance, so our results only provide information about action *representations* and not about motor *control*. We agree with this point, and now say so clearly in our Discussion (page 20). However, our investigation of space-like representation for action-outcome *relations* through neuroimaging does seem to be novel. The action-outcome learning literature classically focusses on the entirely arbitrary relation or mapping between each individual action and its corresponding outcome (Passingham 1988, Wise & Murray, 1999). This is also true for the studies the reviewer

cites on explicit components of action representation. In this literature, each individual mapping is generally assumed to be independent: multiple mappings are simply stored in a look-up table, whose structuring principles, if any, remain unknown. For example, the literature on human sense of agency insists on the mental representation of multiple, parallel, arbitrary relations between different actions and their individual outcomes (for example, Barlas & Obhi, 2013). The novelty of our study lies in showing a latent organizing principle for *multiple* action-outcome memories, and showing that this organizing principle resembles a cognitive map. We suggest that this organizing principle could underlie either explicit or implicit uses of the knowledge.

“Classically, cognitive motor control studies focused on procedural aspects of representing an individual action, recalling the concepts of the procedural memory literature. However, more recent studies recognized an additional major contribution of explicit learning processes in action representation and control (Taylor et al., 2014; Krakauer et al., 2019; Areshenkoff et al., 2022), but did not consider how many such representations might be organized into an overall structure. Here, we have investigated this process of *relating* multiple actions (and their linked outcomes) to each other, as opposed to simply planning an individual action. We show that relations between multiple action-outcome pairs are supported by map-like representations in the hippocampal-entorhinal system. Our findings suggest that multiple arbitrary action-outcome associations are represented in a similar way to other forms of semantic knowledge (Wise & Murray 1999), requiring a continuous circumstantial learning and updating of these associations.”

Reviewer #1 (Remarks on code availability):

I could not actually find any reference to the code (presumably on github) anywhere in the paper.

We now provide the link to the uploaded code and the fMRI data here:

https://osf.io/uz83d/?view_only=f54b8f6247b24193ac29393bed91c17d

Reviewer #2 (Remarks to the Author):

This well-written paper from Barnaveli et al. uses fMRI and VR to show that when multiple action-outcome mappings need to be related to one another, instead of only being represented as distinct, one-to-one mappings between action and outcome, or some weighted mixture, 'motor primitives' are instead organized in a relational 'action-outcome' cognitive map in the EC–HC system. Specifically, they find a grid-like representation in EC for vectors between the two action-outcome/action-outcome combinations required to be evaluated and compared in their task. They then show that BOLD adaptation in HC and EC (as well as OFC and other regions) is proportional to the distance between outcomes in the 2D space in both experimental tasks. By contrast, the SMA shows BOLD enhancement dependent on the repetition of a specific action, potentially suggesting the representation of specific actions (though see below). The HC and SMA exhibit increased functional connectivity in this task when actions evaluated are more distinct in this action-outcome space. The analyses in this paper are generally rigorous and well-controlled. The paper makes a significant contribution by extending understanding of action-outcome mapping and evaluation, and, in particular, elucidates an unexpected role of the HC–EC system in flexible action planning or relational evaluation. Below are some important questions and concerns regarding analyses and their interpretation that should be addressed to support publication:

We thank the reviewer for the interest in our manuscript and for their careful review of our study! We provide our point-by-point responses below.

1. An important question concerns how generalizable these findings are. Does the inclusion of outcomes (specific colored balls) or that task behavior explicitly required evaluating relations between action-outcome mappings on both dimensions for comparisons change the way the brain solves the task, making it more likely to incorporate the HC-EC system? In other words would the HC-EC system map this space if participants simply had to plan the actions to obtain the desired outcomes without any task requirement to relate them to one another?

We thank the reviewer for pointing out the important distinction between planning the individual action to obtain a particular outcome and relating different actions to each other. Although the hippocampal-entorhinal system has been suggested to be involved in forming associations between actions and their outcomes (Corbit & Balleine, 2000; Kim et al., 2022; Hindy et al., 2016; Mattfeld & Stark, 2015), we propose that this system plays an additional role when the tasks at hand require relating multiple actions to each other. We think that the map-like structures in the hippocampal-entorhinal system accommodate the action-outcome associations, particularly in circumstances where making relations between these associations is desired. We do not think that the planning of the individual action to obtain the outcome, without the need to engage in a relational process with other possible actions, should necessarily involve dimensional cognitive maps. However, whether this system might spontaneously or automatically map actions in low-dimensional spaces even in the absence of such relational requirements is an interesting question that would deserve further investigation. We now clarify this distinction in the Discussion section in the following way (see page 20 of the revised manuscript):

“Classically, cognitive motor control studies focused on procedural aspects of representing an individual action, recalling the concepts of the procedural memory literature. However, more recent studies recognized an additional major contribution of explicit learning processes in action representation and control (Taylor et al., 2014; Krakauer et al., 2019;

Areshenkoff et al., 2022), but did not consider how many such representations might be organized into an overall structure. Here, we have investigated this process of *relating* multiple actions (and their linked outcomes) to each other, as opposed to simply planning an individual action. We show that relations between multiple action-outcome pairs are supported by map-like representations in the hippocampal-entorhinal system.”

2. I am unsure about the interpretation of findings concerning the role of the HC-EC system in this task and its interaction with the SMA. It would seem that the findings are likewise consistent with the view that it is the SMA that first simulates a forward model of the actions, permitting the HC-EC system to localize the outcome in the 2D map, and then to compare them using a vector comparison. Or it could be that the cognitive map has been formed in the HC-EC system, and the mappings are retrieved and simulated from this map, as suggested. Unless data can speak to this distinction, the alternative interpretation should at least be discussed.

We agree that our methods cannot clearly determine the direction of the interactions between SMA and HPC-EC. The reviewer points out two possible ways in which the interaction between these regions could unfold – we found these comments helpful and have updated the discussion accordingly. We now argue that instead of one, two alternatives are possible for the interaction between SMA and HPC-EC, and we now show these schematically in the Supplementary Figure 10. We further describe the alternative possibilities in the Discussion section (see page 19 of the revised manuscript).

Supplementary Figure 10: Speculative schematic representation of alternative interaction models between SMA and HPC-EC System. The figure illustrates two possible speculative models of interaction between the SMA and the HPC-EC system. **A** The SMA might simulate individual action-outcome links using a forward model approach, allowing the HPC-EC system to position them within an abstract cognitive map for subsequent computation of relations between multiple action-outcome pairs. This model emphasizes a forward connection from SMA to HPC-EC system. **B** Alternatively, the hippocampal-entorhinal system might integrate information about individual action-outcome links from the SMA during the initial formation of the map (indicated by a curved arrow). Later, the HC-EC can act as an inverse model, by providing a map of pointers that, given a desired outcome, can retrieve the corresponding action representations from the SMA (indicated by straight arrows).

Revised Discussion:

“We speculate that the forward simulation of individual action-outcome links by SMA may allow the hippocampal-entorhinal system to position these links in an abstract cognitive map for subsequent comparison. Alternatively, the hippocampal-entorhinal system could integrate information about individual action-outcome links from SMA during the formation of the map and later serve as a reference system to access these individual action-outcome representations in SMA (see Supplementary Figure 10). Our PPI methods cannot identify the directionality of the interactions between hippocampal-entorhinal system and SMA. Further research could address this question, for example, by measuring the flow of information using methods that combine high spatial resolution with high temporal sensitivity, such as intracranial recordings.”

3. The task design included specific colored ball outcomes, referred to as landmarks, that had to be evaluated in relation to one another in one task. Were there landmark effects, in terms of better performance or attraction, neurally or behaviorally? For instance, were landmark locations more decodable? Or is there evidence of distortion in distances towards the anchors?

We thank the reviewer for the interesting suggestion to test for landmark effects in the task. We did not perform any distortion or decoding analysis as the small size of our action-outcome space or the small number of trials didn't allow us to do so. However, we did test the effect of landmarks (coloured balls) on accuracy in the behavioural data, as suggested by the reviewer. To do this, we focused on the first Comparison Task, in which participants made comparative judgments on pairs of action combinations during the scanning. No coloured balls were mentioned or shown throughout the task. We first extracted trials in which the starting action combination (the first combination of the pair), the ending action combination (the second combination of the pair), or both were associated (that is, were producing) coloured balls in the immersive virtual reality during the training days. For each participant, we stored the number of correct answers from these trials. We then randomly subsampled the same number of trials from the remaining trials and stored the number of correct answers. The subsampling was repeated 10 times and averaged across these iterations for each participant. The results for both types of trials were compared and showed higher accuracy for the trials with the action combinations producing the landmark outcomes (see Figure below). We now include the results as a Supplementary Figure 14, also reported below, speculating that the landmark outcomes of actions facilitate the recall of other associated outcomes, namely, the probability of catching the ball and its visibility. However, given that the means are almost equal (Mean salient = 60.8, Mean non-salient = 59.2; see Figure below) and the overall accuracy of the task was very high,

we approach the interpretation of these results, albeit statistically significant in their difference, with caution.

Supplementary Figure 14. Effects of landmark outcomes on the behavioural performance from the first Comparison Tasks. The behavioural data stems from the first scanning session, where participants performed 144 trials of comparisons between sequentially presented pairs of action combinations. Trials were divided in two categories according to whether any action combination in the pair was associated with landmark outcomes, i.e., coloured balls (salient), or whether neither combination had such an association (non-salient). For each participant, the number of trials with no association with coloured balls was subsampled to match the number of trials with an association ($M=64.6$, $SD=2.78$). The number of correct responses for both trial categories was recorded. The subsampling process was repeated 10 times, and correct responses were averaged across iterations. Accuracy was higher in trials linked to coloured balls ($t(45)= 4.17$, $p<10^{-3}$; two-tailed paired t-test), suggesting that the landmark outcomes of actions might facilitate the recall of other associated outcomes, namely the probability of catching the ball and its visibility. However, the means for salient and non-salient trial types are almost equal (M salient = 60.8, M non-salient = 59.2), and thus the interpretation of these results, albeit statistically significant in their difference, must be approached with caution.

4. For the RSA approach, there does not appear to be any cross-validation. Were on and off grid (similar and dissimilar) angles randomly distributed within each run? This is important to demonstrate due to potential differences in auto-correlation if the distributions are not matched.

We thank the reviewer for the suggestion to implement a cross-validated measure to further support our results. We repeated all our RSA analyses using the cross-validated squared Mahalanobis distance (as implemented in `rsatoolbox`, version 0.1.2). All results remained statistically unchanged and are now mentioned in the manuscript (page 8 and 36) as well as reported in the Supplementary Figure 4.

Addressing the second related point of the comment, we pseudo-randomly distributed the on- and off-grid directions (i.e., similar and dissimilar) within each run, matching both distributions to each other. This distribution of trials was based on several key criteria introduced to obtain good balance across runs:

- a) Two direction types (similar or dissimilar) were presented an equal number of times within each run. Moreover, the direction types were equally distributed within each half of the run, with the same number for both halves of the run.

- b) We considered all possible transitions between the two direction types and balanced the number of the occurrences of these transitions both within and across participants.
- c) We made sure that the trial structure of every run differed from all other runs by at least 10 elements, and further varied across participants.

To illustrate this, we included a Supplementary Figure 11 describing our criteria for random distribution and demonstrating the trial structures of runs for four example participants.

Supplementary Figure 11. Trial structure of the first Comparison Task. **A** The trial structure was carefully designed to be balanced and optimized for the RSA analysis (see Methods, Randomization structure for the first Comparison Task). The distribution of trials with similar and

dissimilar directions across runs (see Methods) for four example participants is shown for illustration purposes. Two direction types (similar or dissimilar) were presented an equal number of times within each run (see panel **B**, Count D and Count S). In addition, the direction types were equally distributed within each half of the run, with the same number for both halves of the run. The vertical green lines represent trials with similar directions, whereas the dark grey lines represent trials with dissimilar directions. The trial structure of every run differed from all other runs by at least 10 elements. Furthermore, the trial structures of runs varied across participants **B,C**. All possible transitions between the two direction types were considered and divided into categories of transitions, and the number of their occurrences was well balanced both within and across participants, with only minor deviations. For individual participants, the number of transitions from dissimilar to similar direction and vice versa (in other words, the number of times the dissimilar direction was followed by a similar direction and the similar direction was followed by a dissimilar direction, abbreviated as D-S and S-D) was almost equal in each run (**B**) and completely equal across all six runs (**C**). Following the same logic, the number of transitions from dissimilar to dissimilar direction (D-D) or from similar to similar direction (S-S) was almost equal in each run (**B**) and completely equal across all six runs (**C**). The same applies to the triple transitions (D-D-D and S-S-S). The black dots in the plot represent individual participants.

Revised methods:

*Our design aimed for a balance between repetition and alternation among direction types, while controlling for the equal occurrence of all possible transitions between these two types (see **Supplementary Figure 11A,B**). The generated trial structure of the run was different across participants and runs (see **Supplementary Figure 11C**).*

Revised results:

*The observed 6-fold periodicity of pattern similarity could not be readily explained by competing alternative models assuming 2- to 8-fold periodicities (all $p > 0.564$; Figure 3F; see Methods). We further found that this effect could not be explained by the relation between the similarity of the directions within the 2D abstract action-outcome space and their starting positions ($t(45)=1.03$; $p=0.155$; one-tailed t -test), ending positions ($t(45)= 0.368$; $p=0.357$; one-tailed t -test), or a model combining both starting and ending positions in that space ($z(45)= 0.522$, $p=0.601$; one-tailed Wilcoxon signed rank test; see Methods for details). Moreover, the effect could not be explained by the mean distance between starting and ending positions of each direction in the action-outcome space, as the mean was controlled to be similar for all directions (see Methods and **Supplementary Figure 3**). **The effect was confirmed using a cross-validated RSA (see Methods and Supplementary Figure 4).***

Revised methods:

*To test for the group-level effects, we compared the resulting correlation values against 0 using a one-tailed one-sample test (Figure 3E). Following the studies on conceptual maps that report a lateralization of grid-like representation^{27,28,47}, we further replicated the analysis for left and right EC separately (Supplementary Figure 5A). **To ensure that the estimated pattern dissimilarities were unbiased, we performed cross-validated RSA using the Crossnobis distance (the cross-validated squared Mahalanobis distance, implemented in rsatoolbox, version 0.1.2) as the dissimilarity measure. Cross-validation was performed across the six runs for each subject. By calculating dissimilarities across independent runs, cross-validation mitigates the effects of noise and overfitting, leading to more reliable estimates of neural representation***

dissimilarity. The main effect as well as the results from every control analysis remained statistically unchanged (see Supplementary Figure 4 and see below for details of the control analyses). To further validate our findings, we conducted a whole-brain searchlight analysis using spheres of radius = 5 voxels, which revealed a cluster in the bilateral hippocampal-entorhinal complex (Supplementary Figure 5C). Importantly, the sampling of the action-outcome space was carefully balanced to ensure that the mean distance between the start and end positions of each direction was equal across trials and not correlated with direction (see Methods, fMRI tasks).

Supplementary Figure 4. Cross-validated representational similarity analysis (RSA) for the grid-like representation of the abstract action-outcome space. **A** Entorhinal cortex (EC) exhibited a significant 6-fold periodicity of pattern similarity ($t(45) = 2.04$, $p = 0.023$; one-tailed t-test). **B** There was no statistically significant effect in control periodicities (2-fold: $t(45) = 0.56$, $p = 0.291$; 3-fold: $t(45) = -0.96$, $p = 0.83$; 4-fold: $t(45) = -0.53$, $p = 0.696$; 5-fold: $t(45) = -0.93$, $p = 0.826$; one-tailed t-test). Only four of these are shown in the figure, as the 7-fold and 8-fold periodicities are equivalent to the 5-fold and 4-fold periodicities due to the size of the action-outcome space. In addition, this effect could not be explained by the relation between the similarity of the directions within the 2D abstract action-outcome space and their starting positions ($t(45) = 0.82$; $p = 0.205$; one-tailed t-test), ending positions ($t(45) = -0.08$; $p = 0.525$; one-tailed t-test), or a model combining both starting and ending positions in that space ($t(45) = 0.49$, $p = 0.314$; one-tailed t-test; see Methods for details). **C** Pattern similarities were extracted separately for each hemisphere using ROI masks and showed a significant 6-fold periodicity effect in the left EC ($t(45) = 2.41$, $p = 0.017$; one-tailed t-test; Bonferroni corrected for tests in both ROIs) but not in the right EC ($t(45) = 0.47$, $p = 0.322$; one-tailed t-test). **D** None of the control ROIs showed the statistically significant 6-fold periodicity effect (HPC: $t(45) = -0.53$, $p = 0.705$; SMA: $t(45) = -4.97$, $p = 1$; PMC: $t(45) = -3.44$, $p = 0.999$; LG: $t(45) = -1.29$, $p = 0.897$; one-tailed t-tests). * $p < 0.05$.

5. It is stated that the mean distances across angles was similar but I did not find an accompanying figure or table. This is an important control. What is the distribution of distances across angles?

We apologize for not providing detailed information regarding the mean distances across different directions (angles). The distances and their frequencies were identical for all non-cardinal directions (2 occurrences of 4.5, 4 occurrences of 3.6, 6 occurrences of 2.2) and for all cardinal directions (3 occurrences of 4, 4 occurrences of 3, 5 occurrences of 2). The distances across both cardinal and non-cardinal directions, as well as their means, were closely matched (see Supplementary Table X below). We calculated the mean distance for each direction based on the frequency of each distance covered by that direction (e.g., average of 2 occurrences of 4.5, 4 occurrences of 3.6, 6 occurrences of 2.2 is 3). The actual mean distances for each direction were also equal among non-cardinal directions (3.4) and among cardinal directions (3), and were approximately equal to each other ($3.4 \approx 3$). We now provide a Supplementary Figure 3 showing the distribution of distances across directions.

Due to the constraints of the action-outcome space architecture and our goal to maximize variation in the starting and ending positions of each direction covering the space, the longer distances were of two different lengths, while the shorter distances were all of the same length. To balance the means across non-cardinal and cardinal directions, we sampled a slightly greater number (one more) of longer distances for cardinal directions, as the equivalent distances in non-cardinal directions were slightly longer.

Non-cardinal directions

Angle	30°	60°	120°	150°	210°	240°	300°	330°
2 x Dist	4.5	4.5	4.5	4.5	4.5	4.5	4.5	4.5
4 x Dist	3.6	3.6	3.6	3.6	3.6	3.6	3.6	3.6
6 x Dist	2.2	2.2	2.2	2.2	2.2	2.2	2.2	2.2
Mean (Count x Dist)	3.0	3.0	3.0	3.0	3.0	3.0	3.0	3.0

Cardinal directions

Angle	0°	90°	180°	270°
3 x Dist	4.0	4.0	4.0	4.0
4 x Dist	3.0	3.0	3.0	3.0
5 x Dist	2.0	2.0	2.0	2.0
Mean (Count x Dist)	2.8	2.8	2.8	2.8

Supplementary Figure 3. Distribution of distances covered by each direction in the abstract action-outcome space. The top panel shows the distances covered by the non-cardinal directions, while the bottom panel shows the distances for the cardinal directions. The first column of both panels shows the number of times each of the three distances was covered by each direction. It also shows the mean distance calculated for each direction based on the frequency of every distance travelled by that direction. For example, the average of 2 x 4.5, 4 x 3.6, 6 x 2.2 is 3. Other columns show that the distances in both cardinal and non-cardinal directions, as well as their means, were closely matched.

Revised methods:

Assuming that the shortest distance between locations in the action-outcome space is the unit 1, the space was sampled using 3 different distances for each direction. The distances covered 4, 3 and 2 units for the cardinal directions (0°, 90°, 180° and 270°) ($M=2.8$ units for each direction, **calculated based on the frequency of each distance covered by that direction. see Supplementary Figure 3 for details**), while 4.5, 3.6 and 2.2 units were covered for all other directions ($M=3$ units for each direction).

6. I could not find a whole-brain map of the grid-like RSA results. Were there grid-like effects in mPFC and PCC, as found in several previous studies (Doeller et al., 2010; Constantinescu et al.,

2016; Bao et al., 2019; Park et al., 2022; Veselic et al., bioRxiv, 2024)? The medial wall isn't visible in Fig. S4C.

We thank the reviewer for this suggestion and now added the whole-brain map of the searchlight RSA results in our Supplementary Figure 5. To summarize our findings, we found the grid-like code in the mPFC, and some clusters in the PCC. However, please note that none of the clusters survive whole-brain correction.

Supplementary Figure 5: Grid-like representation of the abstract action-outcome space.

.... C Whole-brain searchlight RSA using 5 voxel radius spheres revealed a cluster in bilateral EC (Uncorrected; MNI peak voxel coordinates: 27, 1.8, -48; peak voxel Spearman's rho = 1.78) as well as in the medial prefrontal cortex (Uncorrected; MNI peak voxel coordinates: 8, 29, -23; peak voxel Spearman's rho = 1.77). Results are uncorrected for multiple comparisons. $*p < 0.05$; Bonferroni corrected for tests in both ROIs.

7. In the distance adaptation analysis, in addition to HC there is prominent adaptation in lateral OFC. This is interesting in light of the OFC literature when considering these pairings are associated with distinct predictions about outcomes (visibility and catch p) (see e.g. O'Reilly et al., PLoS Biology, 2013).

We agree with the reviewer that this is an important point, as previous literature has indeed shown that the OFC is involved in predicting outcomes. Furthermore, as pointed out by Reviewer 1, the connectivity between the hippocampus and the OFC supports the use of cognitive maps for goal-directed behaviour. Similarly, Reviewer 3 noted that a map-like representation is often found in the anterior mPFC. Therefore, we decided to discuss this finding in the Discussion section as follows (see pages 19 and 20 of the revised manuscript):

“Evidence for a cognitive map representation in mPFC

In addition to the hippocampal-entorhinal system, our analyses also revealed the involvement of the medial prefrontal cortex (mPFC) and the lateral orbitofrontal cortex (IOFC) in the representation of the abstract action-outcome space (see Supplementary Figure 5C and Supplementary Tables 1 and 2). Neural representations consistent with a cognitive maps have been found in the mPFC in humans (Doeller et al., Nature, 2010; Constantinescu et al., 2016) and non-human primates (Bongioanni et al., Nature, 2021). IOFC has been shown to be specifically involved in the construction of such a map (e.g., Costa et al., 2023, rodent study), perhaps reflecting the direct projections from the hippocampus to these prefrontal regions (Reznik et al., 2024; Eichenbaum, 2017; Preston & Eichenbaum, 2013). Other work has suggested that the OFC, particularly its lateral part, represents the expected outcomes based on environmental statistics (O'Reilly et al., 2013) in many tasks (Rudebeck & Murray, 2014) and is involved in disambiguating the unobservable states in a cognitive map of task space (Wilson et al., 2014). The mPFC and OFC, together with the hippocampal-entorhinal system, could therefore participate in acquiring and structuring associative representations (Stalnaker et al., 2015). These representations may play an important role in guiding behaviour when multiple action alternatives are possible.”

Reviewer comments 8-9:

8. How can we be certain SMA is representing 'action plans' per se without using reverse inference? Outcomes and visual stimuli are highly correlated with the specific action plan. Can these be decoupled in the design? For example, would this be possible to dissociate by restricting analysis to trials where the action was the same but the joystick identity and associated outcome differed (i.e. when they switched joysticks)?

9. For the action similarity–based adaptation analysis: Is there evidence for adaptation when two of the same actions are presented for the same option? E.g. a left, left option? Similarly, this brings up the possibility of more than 3 trial types (e.g., if left is used twice in the first option and once in the second option), or were those trials from this analysis?

Comment 8 and 9 of the reviewer are related and we therefore decided to answer them together with a set of new analyses, described here.

In the 8th comment, the reviewer points out, that to be sure whether participants represent action plans per se, we should dissociate (1) the visual stimulus (in this case, the identity of the joystick) from the action and (2) the action-related representations from the outcome-related representations.

1. We question the necessity to dissociate visual stimuli from the actions. In our adaptation analysis, we observed the modulation of SMA activity depending on the overlap in visually cued actions. It is common to use perceptual judgements as a source of information about action representations. For example, the effects of actions on temporal experience have only been studied through their secondary effects on perceptual judgement (Hagura et al., 2017; Stetson et al., 2006); Furthermore, images of tools are often used to investigate the tool-related action understanding (Federico et al., 2023).
2. We agree with the importance to dissociate the actions from their outcomes. Indeed, the actions of each of the two joysticks, controlling two different outcomes, are fully correlated with associated outcomes. However, our analysis is still partially based on trials with decoupled actions and outcomes. In the original version of the manuscript, we were not explaining exact details of the trial categories in the main analysis. Specifically, trials with overlapping action plans include trials where an action could be shared between the same joystick (e.g., Joystick 1) in a pair of action combinations (e.g., Combination 1: Left Up, Combination 2: Left Right) or between the different joysticks in the pair (e.g., Combination 1: Left Up, Combination 2: Up Right). Since (i) the two joysticks control two different outcomes, and (ii) in action-outcome mapping, even the same probabilities of two outcomes are rarely associated with the same actions across joysticks, the stimuli were already partially decoupled from actions. We now modified Figure 5 and clarified this part in the methods.

In addition, as the reviewer suggested, we could fully disentangle actions from their outcomes in the control analysis. The reviewer is right, it would have been very convenient to run the analysis on trials where the actions were 'swapped' between joysticks (e.g. the first action combination of a pair consists of the actions Left and Right for joystick 1 and 2, and the second action combination of a pair consists of the actions Right and Left for joystick 1 and 2). Unfortunately, due to the small number of trials of this type, we could not perform the analysis on this subset of trials. However, the main analysis includes trials where the action shared between the two action combinations is the action of joystick 1 in the first combination and the action of joystick 2 in the second combination, and vice versa (e.g., Combination 1: Left Up, Combination 2: Up Right). We used these trials to decouple actions from outcomes. We restricted our analysis to these trials by including them as overlapping action plans and comparing them with non-overlapping action plans. The logic here is the same as in the main analysis (see Methods). Thus, this control analysis could show the outcome-independent representation of actions. Here is the table showing the mean number of these trials for each of the 6 runs across subjects:

	Mean	Standard Deviation
1	5.6531	2.2131
2	6.0204	1.6645
3	5.5102	1.7809
4	5.5306	2.0525
5	5.9796	1.6893
6	5.5510	1.8715

Apart from trials with non-overlapping actions, we excluded any other type of trial. We found a significant modulation of the BOLD activity in SMA ($t(45)=-2.29$, $p=0.024$, two-tailed t-test) but not in the PMC ($t(45)=-0.29$, $p=0.775$, two-tailed t-test), suggesting that the SMA contains action-related representations. We now report these results in the Supplementary Figure 8.

In the 9th comment, the reviewer asked whether there is evidence for BOLD modulation in the action similarity-based adaptation analysis when the same action is presented three times in the two action combinations. Our trials indeed included a fair number of trials where the two action combinations share an overlapping action, but the action is presented twice in either the first or second combination of the pair (e.g. the first action combination of a pair consists of the actions Left and Left for joystick 1 and 2, and the second action combination of a pair consists of the actions Right and Left for joystick 1 and 2). Similar to previous analyses, we created a parametric regressor, but this time we categorised trials into 3 types based on the number of times the shared action was present in each of the combinations of a pair. This resulted in shared actions being presented either once in each of the combinations, or once in one combination and twice in the other combination of actions. We excluded the small number of trials with switched joysticks (see above). The analysis yielded significant results (SMA: $t(45)=-3.66$, $p=0.0001$; PMC: $t(45)=-3.21$, $p=0.003$; two-tailed t-tests; not corrected for multiple comparisons). Although there was no significant difference between the results of this analysis and the results of the main analysis for the SMA ($t(45)= 0.58$, $p=0.57$, paired-sample t-test), the PMC showed a strong significant modulation in contrast to the main analysis. Thus, there is indeed evidence for BOLD modulation for the shared action when presented in total of three times within two successive action combinations, suggesting that the SMA may represent the action plans. This new analysis is now included as the Supplementary Figure 8.

Figure 5: Representation of individual actions in the SMA. A,B Logic of analysis. Trials from the first Comparison Task were categorized as similar (Trial type 1) or dissimilar (Trial type 2) depending on whether the two action combinations of a pair shared the common action: in the current example of

overlapping action plans, the common action would be moving the joystick to the right (A). Action could be shared between two action combinations by the same or different joysticks.

Supplementary Figure 8. Representation of individual actions in the SMA. **A** Trials from the first Comparison Task were categorized into three different trial types. The first type, shown in the top panel, consisted of trials where the two action combinations of a pair shared the common action, which occurred three times: in the current example of overlapping action plans, the common action would be moving the joystick to the right with both joysticks in the first action combination and with one joystick in the second action combination. The middle panel shows a second trial type, where the shared common action occurs only once in each of the combinations, shared by the same or by the different joysticks. The third trial type with non-overlapping action plans is shown on the bottom panel. **B** The logic of the analysis is the same as of the main analysis (see Methods). The constructed parametric regressor based on the three trial types (**A**) yielded significant results for both ROIs of interest (SMA: $t(45) = -3.66, p = 0.0001$; PMC: $t(45) = -3.21, p = 0.003$; two-tailed t-test). **C** The action similarity-dependent increase in activity in the SMA was confirmed on the whole-brain level (FDR-corrected using a voxel-level threshold of $p < 0.01$; MNI peak voxel coordinates: $-2, -3, 61$; peak voxel $t(45) = -4.586$; two-tailed test). **D** To fully disentangle actions from their outcomes, we performed a control analysis that differed from the previous analysis only in the types of trials entered as 'overlapping action plans'. Specifically, the analysis was restricted to the trials where the common action between the two action combinations was shared by different joysticks (e.g., joystick 1 in the first and joystick 2 in the second combination cued the movement to the right). The modulation of the BOLD activity remained significant in the SMA ($t(45) = -2.29, p = 0.024$, two-tailed t-test) but not in the PMC ($t(45) = -0.29, p = 0.775$, two-tailed t-test), suggesting that the SMA represents action-related information. * $p < 0.05$; ** $p < 0.01$; two-tailed t-tests.

Revised Results:

Taken together, the results of this analysis indicate that cueing actions by observing the corresponding joystick settings elicits the activation of distinct action plans in the SMA. Given that actions and outcomes of each joystick are fully correlated in the task, we cannot completely separate action- from outcome-specific representations within the SMA. However, additional analyses show that SMA activity is modulated by similar actions of two different joysticks that do not produce the same outcomes, suggesting that the SMA contains action-related representations (see Supplementary Figure 8). Furthermore, in contrast to the hippocampal-entorhinal system, SMA showed no indication of a two-dimensional representation of action-outcome space based on the dimensions supplied by our task (Supplementary Figure 6B). These findings, together with positive evidence for SMA activation by observational cueing of action plans (see Figure 1H), is consistent with SMA representation of individual action-outcome links, without the relational organization of multiple such links into an abstract space.

10. I did not follow the logic for why the correlation between HC and SMA would be predicted to increase with the distance between action-outcome points. The motivation needs to be better articulated.

We apologize for not making this point immediately clear in the manuscript. The reviewer may be asking two points: (1) why does the distance-dependent modulation of HC also affect the relation between HC and SMA? (2) why does this relation increase with distance, rather than decrease?

1. In our adaptation analysis, we observed the modulation of hippocampal activity as a function of distance within the abstract action-outcome space – we interpret this as a mapping of action-outcome space. We hypothesized that cognitive maps in the hippocampal system do not operate in isolation, but in coordination with other brain regions from different functional domains. This led us to explore whether the interaction between the hippocampus and SMA follows this same pattern of modulation seen in hippocampal activity.
2. To clarify, our hypothesis did not specifically predict an *increase* in connectivity (measured as covariation of BOLD signals) between the hippocampus and the SMA. Instead, we expected a modulation of connectivity that might manifest as either an increase or a decrease in connectivity depending on the *distance* in the action-outcome space. Therefore, we used a two-tailed test to remain agnostic with respect to the specific direction of such modulation.

Revised results:

The existence of two parallel systems, (i) a map-like representation of action-outcome relations in the hippocampal-entorhinal system and (ii) a representation of individual actions in the SMA, allowed us to test the hypothesis that the hippocampal-entorhinal abstract map interacts with individual action representations in the motor system to relate multiple individual action-outcome associations.

Motivated by the distance-based modulation of hippocampal activity in the action-outcome space, we aimed to investigate whether the connectivity between the HPC and SMA follows the same pattern of modulation.

We conducted a generalized psychophysiological interaction analysis⁵⁸ (gPPI) on the data from the first Comparison Task (Figure 1H) with the HPC and SMA as seed ROIs and the distance between action combinations in the abstract action-outcome space (Figure 1E) as a psychological component (Figure 6A,B; see Methods). Importantly, we did not formulate a hypothesis about the direction of modulation of the connectivity between HPC and SMA. The averaged results for both seeds showed that functional connectivity between the left HPC and SMA was increased as

a function of distance between action combinations in the action-outcome space ($t(45) = 2.42$, $p = 0.021$; two-tailed t -test; Figure 6C).

11. The adaptation results in Fig 4 and supplementary tables of significant clusters use FDR correction. While this is a bit unconventional, the t -stats look respectable. The authors should upload their contrast images (I assume this will be done at some stage).

The contrast images for fMRI analyses are uploaded and can be accessed via the following link: https://osf.io/uz83d/?view_only=f54b8f6247b24193ac29393bed91c17d

12. Are the example fits for the MDS and Procrustes analysis from day one or day two? It would be interesting to see what these fits look like for a good, bad, and median participant from day 2 to get a better sense of how accurately these spaces are represented.

We apologize for the lack of clarity. The MDS and Procrustes fits shown in Figure 2 illustrate the data of an example participant from day two, which we now specify in the figure legend. We followed the suggestion of the reviewer and defined the best, median and worst participant based on their fits of similarity judgments to the actual positions of stimuli in the action-outcome space, as resulting from the procrustes analysis. The plots showing MDS and Procrustes fits for these three example participants for both Rating Tasks are now added to the Supplementary Figure 12.

Supplementary Figure 12. Reconstructed map-like representation obtained from the two Rating Tasks. A Abstract action-outcome space. B, C The Procrustes distances obtained by fitting the multidimensional scaling (MDS) coordinates of the action combinations (**B**) and coloured balls (**C**) to the true coordinates of the respective stimuli in the action-outcome space were smaller than the critical distances obtained from the permutation tests (see Figure 2; see Methods). **D, E** The Procrustes distances from Rating Task 1 (**B**) and Rating Task 2 (**C**) were used to define three example participants with best, median and worst fit. The corresponding upper (**D**) and lower (**E**) panels demonstrate the data from day 2.

Reviewer #3 (Remarks to the Author):

I enjoyed reading this manuscript arguing for a form of abstract coding of actions in two parts of the medial temporal lobe. The results are both novel and compelling and they will be of broad interest to cognitive scientists, neuroscientists, and motor scientists not least because they challenge the prevailing belief that such medial temporal lobe structures are not concerned with action representation (although as the authors correctly note, the importance of the medial temporal lobe is indicated by often overlooked studies such as Brasted et al., *Brain*, 2003). The analyses that have been used are ones that are relatively novel but they have now been used in other contexts and there is broad agreement on how they should be performed.

We thank the Reviewer for the positive feedback and for the interest in our manuscript! Below please find our point-by-point responses to the comments raised by the Reviewer.

1. The right hand panels of both parts A and B of figure 2 each contain keys with elements called “critical distance”. I am not sure what this label means. It looks as if there are lines that correspond to these labels in right hand panel A but not in right hand panel B? Is that right? Are they, however, overlapping with one another in panel A? Is there perhaps a line in panel B but which overlaps with the continuous dark blue/purple line? I wonder if a revised manuscript might clarify what is being shown in the two right hand panels and also explain why it is being shown and how it relates to the interpretation of the results?

We apologize to the reviewer for not explaining the notion of ‘critical distance’ in the plot. By applying MDS, we first extracted coordinates from participants’ judged similarity scores between stimuli. These coordinates, representing the structure of similarity scores, were mapped to the actual positions of stimuli in the action-outcome space using Procrustes analysis. The procrustes distance, a measure of goodness of fit, is assessed as the normalized sum of squared errors between the reconstructed and actual positions. To evaluate the significance of the mapping, we created a distribution of Procrustes distances, obtained by mapping the MDS coordinates to the sets of coordinates with shuffled stimulus – position assignment. The two right panels of B and D show that at the group level, the fit between the reconstructed and actual positions of stimuli in the action-outcome space is better than the critical fit (indicated by critical Procrustes distance). The critical Procrustes distance is defined as the 5th percentile of each participant’s distribution from the permutation analysis. The threshold corresponds to the threshold for statistical significance at $\alpha = 0.05$. This permutation analysis supports the claim that the participants organize multiple action-outcome associations in an abstract two-dimensional map.

The reviewer is right, the lines representing critical distance are overlapping with lines representing mean data distance. We now overlaid the lines representing critical distance to the mean distance, thus making them more visible. The legend of the figure now explicitly states the meaning of the ‘critical distance’. Together with Methods section: ‘Analysis: Rating Tasks’, it should give a better understanding for the right panels of the figure parts **B** and **D**.

Revised version of panels B and D: B,D To visually evaluate how participants' responses reflected the structure of the action-outcome space, we reconstructed their implied mental space from the pairwise similarity scores by applying multidimensional scaling (MDS) and obtained coordinate positions (indicated by snowflakes) that represent the structure of these similarity estimates. The resulting coordinates were mapped using Procrustes analysis to match the original positions (indicated by circles) of action combinations or coloured balls in the action-outcome space (see Bellmund et al.⁴⁴ and see Methods). The corresponding left and middle panels demonstrate the data of an example participant from day 2. To evaluate the significance of mapping, we fitted MDS coordinates using the same approach to the sets of coordinates with shuffled stimulus – position assignments (curved solid lines). The corresponding right panels show the data for all participants. The vertical solid lines represent the mean Procrustes distances between original and reconstructed positions for each training day. The vertical dashed lines indicate mean critical Procrustes distances obtained from the shuffled distributions for each training day. The shaded areas indicate the standard error of the mean, calculated across participants. The results revealed that the mapping was indeed better than would be expected by chance for action combinations (Day 1: data distance: $M = 0.368$, $SD = 0.273$; critical distance: $M = 0.526$, $SD = 0.009$; Day 2: data distance: $M = 0.257$, $SD = 0.261$; critical distance: $M = 0.528$, $SD = 0.011$) as well as coloured balls (Day 1: data distance $M = 0.371$, $SD = 0.219$; critical distance: $M = 0.382$, $SD = 0.028$; Day 2: data distance: $M = 0.247$, $SD = 0.205$; critical distance: $M = 0.372$, $SD = 0.04$), pointing towards organization of multiple action-outcome associations in an abstract two-dimensional map. **** $p < 10^{-4}$; Bonferroni corrected for tests on both days.

Revised methods:

This resulted in a distribution of Procrustes distances from 40320 possible permutations, which we thresholded at 5th percentile (a threshold, we termed critical distance) corresponding to the threshold for statistical significance at $\alpha = 0.05$.

To assess the significance of the mapping, we performed permutation tests using shuffled balls - position assignment and tested whether participants' fit of two maps was better than the critical

distance. The critical distance was obtained from 720 possible permutations (see Figure 2D and Supplementary Figure 12C).

2. Figure 3A. I think that it might help some readers to make clear in the legend for figure 3a that what is being shown is the trajectory in the abstract action representation space between one action and the second action when they are presented sequentially one after the other in the action comparison task. I realise that this is in the main text but it would be easy to come away from the figure 3A and its legend thinking that it is something about the trajectory of a given action on each trial that is being analysed.

We thank the reviewer for the comment and agree that it is indeed very important to explain the analysis logic better in Figure 3A. We now provide a revised version of Figure 3, where we clarify that the panel A shows the directions in the abstract action-outcome space, relating the first to the second action combination of a pair.

Figure 3: Grid-like representation of the abstract action-outcome space. A,B,C Logic of analysis. The marked positions in the abstract action-outcome space correspond to different combinations of actions presented sequentially as pairs in the first Comparison Task. Each arrow corresponds to a direction in this abstract space, relating the first to the second action combination of a pair. The action- outcome space was sampled in 30⁰ steps using 12 different directions in the first Comparison Task (A).

3. Line 272, figures 3 and supplementary 4. Is there any evidence of hexadirectional grid-like encoding in anterior medial prefrontal cortex? While the focus on the entorhinal cortex in the current report is clearly warranted, there is evidence from some of the authors' own previous work (Doeller et al., Nature, 2010) as well as the work of others (Constantinescu et al., 2016 Science, Bongioanni et al., Nature, 2021) that such patterns of activity are sometimes almost more easily identifiable in anterior medial prefrontal cortex. It is difficult to assess this from the supplementary figure because of the placement of the sections.

We thank the reviewer for the comment on the evidence for hexadirectional grid-like encoding in the medial prefrontal cortex (mPFC) as shown in human and animal studies. Similarly, Reviewers 1 and 2 also pointed out the BOLD adaptation to the distance in the action-outcome space in the OFC. We apologize for not showing all the slices resulting from our whole brain analysis in Supplementary Figure 5 and now provide the slices where the mPFC is clearly visible (see the Figure below). In

addition, we now mention this finding in the Results section and discuss it in the Discussion section as follows (see pages 9, 19 and 20 of the revised manuscript):

Revised Results:

Finally, the effect appeared to be primarily **localized** to the EC: this was evident both from the absence of a **statistically significant** effect in (i) a control ROI analysis conducted in the hippocampus (HPC), supplementary motor area (SMA), premotor cortex (PMC), and lingual gyrus (LG) (all p -values > 0.625 ; Supplementary Figure 5B; see Methods for selection of ROIs) and from the results of (ii) a whole-brain searchlight analysis (Supplementary Figure 5C). **The whole-brain analysis additionally revealed a large cluster in the medial prefrontal cortex (mPFC) (Supplementary Figure 5C).**

Supplementary Figure 5: Grid-like representation of the abstract action-outcome space.

...C Whole-brain searchlight RSA using 5 voxel radius spheres revealed a cluster in bilateral EC (Uncorrected; MNI peak voxel coordinates: 27, 1.8, -48; peak voxel Spearman's rho = 1.78) as well as in the medial prefrontal cortex (Uncorrected; MNI peak voxel coordinates: 8, 29, -23;

peak voxel Spearman's rho = 1.77). Results are uncorrected for multiple comparisons. * $p < 0.05$; Bonferroni corrected for tests in both ROIs.

Revised Discussion:

“Evidence for a cognitive map representation in mPFC

In addition to the hippocampal-entorhinal system, our analyses also revealed the involvement of the medial prefrontal cortex (mPFC) and the lateral orbitofrontal cortex (IOFC) in the representation of the abstract action-outcome space (see Supplementary Figure 5C and Supplementary Tables 1 and 2). Neural representations consistent with a cognitive maps have been found in the mPFC in humans (Doeller et al., Nature, 2010; Constantinescu et al., 2016) and non-human primates (Bongioanni et al., Nature, 2021). IOFC has been shown to be specifically involved in the construction of such a map (e.g., Costa et al., 2023, rodent study), perhaps reflecting the direct projections from the hippocampus to these prefrontal regions (Reznik et al., 2024; Eichenbaum, 2017; Preston & Eichenbaum, 2013). Other work has suggested that the OFC, particularly its lateral part, represents the expected outcomes based on environmental statistics (O'Reilly et al., 2013) in many tasks (Rudebeck & Murray, 2014) and is involved in disambiguating the unobservable states in a cognitive map of task space (Wilson et al., 2014). The mPFC and OFC, together with the hippocampal-entorhinal system, could therefore participate in acquiring and structuring associative representations (Stalnaker et al., 2015). These representations may play an important role in guiding behaviour when multiple action alternatives are possible.”

4. Line 450-470, figure 6. I think that the abstract motor space in which the PPI is conducted is the same one that is explored in figure 4 as opposed to the ones explored in figures 3 or 5. If that is the case then maybe that could be made clearer somewhere in lines 450-470. On a related note, I wondered if although connectivity between hippocampus and motor association reflects the abstract motor space, does connectivity between different motor association regions reflect the action similarity space explored in figure 5? In other words, while PPIs between hippocampus and motor association areas in premotor and parietal cortex might reflect the abstract motor space, do the motor association areas themselves show evidence, in a PPI, of interacting with one another in the same abstract space (illustrated in figure 4) or the motor similarity space (illustrated in figure 5)?

We apologize to the reviewer for not making it immediately clear in the text that we used the distances in the action-outcome space to perform the gPPI analysis. We would also like to clarify that the analyses shown in Figures 3 and 4 are based on the same abstract action-outcome space, which is defined by the relations between action-outcome associations. Specifically, the arrangement of joystick actions along their outcome dimensions can be thought of as a two-dimensional mapping of the action-outcome associations that we observed in the complementary analyses of Figures 3 and 4. The reviewer correctly points out that the analysis in Figure 5 explores the similarity between actions themselves, which is different from the distribution of action combinations in the two-dimensional space defined by their outcomes.

The gPPIs between the hippocampus and SMA are based on the distances in the abstract action-outcome space, which correspond to the similarity between the outcomes of the two action combinations (first comparison task). In this sense, the gPPI analyses are the follow-up of the distance-related modulation of the BOLD signal observed in the hippocampus in Figure 4. We now refer to the action-outcome space panel from Figure 1 in all related main analyses that do not have the action-outcome space panel in the corresponding figure.

We also thank the reviewer for the comment on whether the motor association areas themselves show evidence of interaction based on distances in the action-outcome space as well as on action similarity. We chose the SMA and PMC as our seed ROIs. We first replicated our gPPI analysis in the CONN toolbox, where the psychological component was the distance between action combinations in the abstract action-outcome space. However, we did not observe a significant modulation between any of the selected ROIs (PMC to SMA: $t(45) = -0.068$, $p = 0.942$; SMA to PMC: $t(45) = 0.612$, $p = 0.548$; two-tailed t-tests).

Next, we conducted a new gPPI analysis where the psychological component was the shared action between two action combinations. Specifically, we again used the data from the first Comparison Task and divided the task trials into two conditions: condition 1 consisted of trials in which the two action combinations shared a common action (e.g., move joystick to the right), while condition 2 consisted of trials in which the action combinations of a pair did not share an action. We observed a significant modulation of activity between PMC and SMA when PMC was used as the seed region (PMC to SMA: $t(45) = 2.06$, $p = 0.047$; two-tailed t-test), but not when SMA was used as the seed region (SMA to PMC: $t(45) = -0.799$, $p = 0.422$; two-tailed t-test) and not when the results were averaged across both seeds ($t(45) = 1.511$, $p = 0.138$; two-tailed t-test). Overall, the results of these analyses are inconclusive. For this reason, and considering that the focus of the study did not include connectivity within the motor cortex, we would refrain from including additional analyses so as not to increase the length of the manuscript. However, if the reviewer feels it is important, we can discuss these additional analyses.

Revised Results:

We found that the estimated subjective similarity scores of the sampled stimuli correlated significantly and positively with the distances between the actual positions of these stimuli in the abstract action-outcome space (Figure 1E; Figure 2A,C; see Methods).

*The existence of two parallel systems, (i) a map-like representation of action-outcome relations in the hippocampal-entorhinal system and (ii) a representation of individual actions in the SMA, allowed us to test the hypothesis that the hippocampal-entorhinal abstract map interacts with individual action representations in the motor system to relate multiple individual action-outcome associations. **Motivated by the distance-based modulation of hippocampal activity in the action-outcome space, we aimed to investigate whether the connectivity between the HPC and SMA follows the same pattern of modulation.***

*We conducted a generalized psychophysiological interaction analysis⁵⁸ (gPPI) on the data from the first Comparison Task (Figure 1H) with the HPC and SMA as seed ROIs and the distance between action combinations in the abstract **action-outcome** space (Figure 1E; Figure 4A) as a psychological component (Figure 6A,B; see Methods). **Importantly, we did not formulate a hypothesis about the direction of modulation of the connectivity between HPC and SMA.** The averaged results for both seeds showed that functional connectivity between the left HPC and SMA was increased as a function of distance between action combinations in the action-outcome space ($t(45) = 2.43$, $p = 0.019$; two-tailed t-test; Figure 6C).*

5. Notwithstanding the novelty of the current report, some mention of the pioneering work of Passingham, Petrides, and Murray and Wise on conditional motor mappings / arbitrary sensorimotor mappings, as opposed to reaching and grasping seems necessary.

We thank the reviewer for pointing out these very relevant studies, which we unfortunately omitted. We now refer to these in the Introduction and discuss in the Discussion section as follows (see page 2, 20 and 21 of the revised manuscript):

Revised introduction:

However, humans (but also some other animals, see **Passingham 1988, Wise & Murray, 1999, and Yamazaki et al.³**), have a remarkable ability to rapidly learn and then exploit arbitrary associations between actions and outcomes (**Toni, Rushworth, Passingham 2001**): think, for example, of the **different** keypress actions required to get a subway ticket from the ticket machine. Although this may be difficult when you first arrive in a new city, it rapidly becomes fluent and automatic.

Revised discussion:

“Our findings now show that action-outcome representations in the hippocampal-entorhinal system are in fact well-suited for flexible action selection, since they represent action-outcome information in a format that supports relations between multiple options. Interestingly, earlier work had suggested hippocampal involvement in novel arbitrary mapping of visual stimuli to motor actions in human patients (Petrides, 1985) as well as in Macaque monkeys (Wise & Murray, 1999).”

“Classically, cognitive motor control studies focused on procedural aspects of representing an individual action, recalling the concepts of the procedural memory literature. However, more recent studies recognized an additional major contribution of explicit learning processes in action representation and control (Taylor et al., 2014; Krakauer et al., 2019; Areshenkoff et al., 2022), but did not consider how many such representations might be organized into an overall structure. Here, we have investigated this process of relating multiple actions (and their linked outcomes) to each other, as opposed to simply planning an individual action. We show that relations between multiple action-outcome pairs are supported by map-like representations in the hippocampal-entorhinal system. Our findings suggest that multiple arbitrary action-outcome associations are represented in a similar way to other forms of semantic knowledge (Wise & Murray 1999), requiring a continuous circumstantial learning and updating of these associations.”

BIBLIOGRAPHY

Zhang, H., & Maloney, L. T. (2012). Ubiquitous log odds: A common representation of probability and frequency distortion in perception, action, and cognition. *Frontiers in Neuroscience, JAN*. <https://doi.org/10.3389/FNINS.2012.00001>

Doeller, C. F., Barry, C., & Burgess, N. (2010). Evidence for grid cells in a human memory network. *Nature 2010 463:7281, 463(7281), 657–661*. <https://doi.org/10.1038/nature08704>

Constantinescu, A. O., O'Reilly, J. X., & Behrens, T. E. J. (2016). Organizing conceptual knowledge in humans with a gridlike code. *Science, 352(6292), 1464–1468*. <https://doi.org/10.1126/SCIENCE.AAF0941>

Bongioanni, A., Folloni, D., Verhagen, L., Sallet, J., Klein-Flügge, M. C., & Rushworth, M. F. S. (2021). Activation and disruption of a neural mechanism for novel choice in monkeys. *Nature* 2021 591:7849, 591(7849), 270–274. <https://doi.org/10.1038/s41586-020-03115-5>

Costa, K. M., Scholz, R., Lloyd, K., Moreno-Castilla, P., Gardner, M. P. H., Dayan, P., & Schoenbaum, G. (2023). The role of the lateral orbitofrontal cortex in creating cognitive maps. *Nature Neuroscience*, 26(1), 107. <https://doi.org/10.1038/S41593-022-01216-0>

Reznik, D., Margulies, D. S., Witter, M. P., & Doeller, C. F. (2024). Evidence for convergence of distributed cortical processing in band-like functional zones in human entorhinal cortex. *Current Biology*. <https://doi.org/10.1016/J.CUB.2024.10.020>

Eichenbaum, H. (2017). Prefrontal–hippocampal interactions in episodic memory. *Nature Reviews Neuroscience* 2017 18:9, 18(9), 547–558. <https://doi.org/10.1038/nrn.2017.74>

Preston, A. R., & Eichenbaum, H. (2013). Interplay of hippocampus and prefrontal cortex in memory. *Current Biology*, 23(17), R764–R773. <https://doi.org/10.1016/J.CUB.2013.05.041>

O'Reilly, J. X., Jbabdi, S., Rushworth, M. F. S., & Behrens, T. E. J. (2013). Brain Systems for Probabilistic and Dynamic Prediction: Computational Specificity and Integration. *PLoS Biology*, 11(9), e1001662. <https://doi.org/10.1371/JOURNAL.PBIO.1001662>

Rudebeck, P. H., & Murray, E. A. (2014). The Orbitofrontal Oracle: Cortical Mechanisms for the Prediction and Evaluation of Specific Behavioral Outcomes. *Neuron*, 84(6), 1143–1156. <https://doi.org/10.1016/J.NEURON.2014.10.049>

Wilson, R. C., Takahashi, Y. K., Schoenbaum, G., & Niv, Y. (2014). Orbitofrontal cortex as a cognitive map of task space. *Neuron*, 81(2), 267–279. <https://doi.org/10.1016/J.NEURON.2013.11.005>

Stalnaker, T. A., Cooch, N. K., & Schoenbaum, G. (2015). What the orbitofrontal cortex does not do. *Nature Neuroscience* 2015 18:5, 18(5), 620–627. <https://doi.org/10.1038/nn.3982>

Bellmund, J. L. S., Deuker, L., Schröder, T. N., & Doeller, C. F. (2016). Grid-cell representations in mental simulation. *ELife*, 5(AUGUST). <https://doi.org/10.7554/ELIFE.17089>

Bao, X., Gjorgieva, E., Shanahan, L. K., Howard, J. D., Kahnt, T., & Gottfried, J. A. (2019). Grid-like Neural Representations Support Olfactory Navigation of a Two-Dimensional Odor Space. *Neuron*, 102(5), 1066-1075.e5. <https://doi.org/10.1016/j.neuron.2019.03.034>

Viganò, S., Rubino, V., Soccio, A. Di, Buiatti, M., & Piazza, M. (2021). Grid-like and distance codes for representing word meaning in the human brain. *NeuroImage*, 232, 117876. <https://doi.org/10.1016/J.NEUROIMAGE.2021.117876>

Viganò, S., Bayramova, R., Doeller, C. F., & Bottini, R. (2023). Mental search of concepts is supported by egocentric vector representations and restructured grid maps. *Nature Communications* 2023 14:1, 14(1), 1–14. <https://doi.org/10.1038/s41467-023-43831-w>

Morgan, L. K., MacEvoy, S. P., Aguirre, G. K., & Epstein, R. A. (2011). Distances between Real-World Locations Are Represented in the Human Hippocampus. *Journal of Neuroscience*, 31(4), 1238–1245. <https://doi.org/10.1523/JNEUROSCI.4667-10.2011>

Theves, S., Fernández, G., & Doeller, C. F. (2020). The Hippocampus Maps Concept Space, Not Feature Space. *Journal of Neuroscience*, *40*(38), 7318–7325. <https://doi.org/10.1523/JNEUROSCI.0494-20.2020>

Viganò, S., & Piazza, M. (2020). Distance and Direction Codes Underlie Navigation of a Novel Semantic Space in the Human Brain. *Journal of Neuroscience*, *40*(13), 2727–2736. <https://doi.org/10.1523/JNEUROSCI.1849-19.2020>

Chouinard, P. A., & Goodale, M. A. (2009). fMRI adaptation during performance of learned arbitrary visuomotor conditional associations. *NeuroImage*, *48*(4), 696–706. <https://doi.org/10.1016/j.neuroimage.2009.07.020>

Persichetti, A. S., Avery, J. A., Huber, L., Merriam, E. P., & Martin, A. (2020). Layer-Specific Contributions to Imagined and Executed Hand Movements in Human Primary Motor Cortex. *Current Biology: CB*, *30*(9), 1721–1725.e3. <https://doi.org/10.1016/j.cub.2020.02.046>

Behrens, T. E. J., Muller, T. H., Whittington, J. C. R., Mark, S., Baram, A. B., Stachenfeld, K. L., & Kurth-Nelson, Z. (2018). What Is a Cognitive Map? Organizing Knowledge for Flexible Behavior. *Neuron*, *100*(2), 490–509. <https://doi.org/10.1016/j.neuron.2018.10.002>

Taylor, J. A., Krakauer, J. W., & Ivry, R. B. (2014). Explicit and implicit contributions to learning in a sensorimotor adaptation task. *The Journal of Neuroscience: The Official Journal of the Society for Neuroscience*, *34*(8), 3023–3032. <https://doi.org/10.1523/JNEUROSCI.3619-13.2014>

Areshenkoff, C., Gale, D. J., Standage, D., Nashed, J. Y., Flanagan, J. R., & Gallivan, J. P. (2022). Neural excursions from manifold structure explain patterns of learning during human sensorimotor adaptation. *ELife*, *11*. <https://doi.org/10.7554/ELIFE.74591>

Krakauer, J. W., Hadjiosif, A. M., Xu, J., Wong, A. L., & Haith, A. M. (2019). Motor Learning. *Comprehensive Physiology*, *9*(2), 613–663. <https://doi.org/10.1002/cphy.c170043>

Hagura, N., Haggard, P., & Diedrichsen, J. (2017). Perceptual decisions are biased by the cost to act. *ELife*, *6*. <https://doi.org/10.7554/ELIFE.18422>

Stetson, C., Cui, X., Montague, P. R., & Eagleman, D. M. (2006). Motor-sensory recalibration leads to an illusory reversal of action and sensation. *Neuron*, *51*(5), 651–659. <https://doi.org/10.1016/j.neuron.2006.08.006>

Passingham, R. E. (1988). Premotor cortex and preparation for movement. *Experimental Brain Research*, *70*(3), 590–596. <https://doi.org/10.1007/BF00247607>

Wise, S. P., & Murray, E. A. (1999). Role of the hippocampal system in conditional motor learning: Mapping antecedents to action. *Hippocampus*, *9*(2), 101–117. [https://doi.org/10.1002/\(SICI\)1098-1063\(1999\)9:2<101::AID-HIPO3>3.0.CO;2-L](https://doi.org/10.1002/(SICI)1098-1063(1999)9:2<101::AID-HIPO3>3.0.CO;2-L)

Barlas, Z., & Obhi, S. S. (2013). Freedom, choice, and the sense of agency. *Frontiers in Human Neuroscience*, *7*(AUG), 60625. <https://doi.org/10.3389/FNHUM.2013.00514/BIBTEX>

Corbit, L. H., & Balleine, B. W. (2000). The Role of the Hippocampus in Instrumental Conditioning. *Journal of Neuroscience*, *20*(11), 4233–4239. <https://doi.org/10.1523/JNEUROSCI.20-11-04233.2000>

Kim, J., Joshi, A., Frank, L., & Ganguly, K. (2022). Cortical–hippocampal coupling during manifold exploration in motor cortex. *Nature* 2022 613:7942, 613(7942), 103–110. <https://doi.org/10.1038/s41586-022-05533-z>

Hindy, N. C., Ng, F. Y., & Turk-Browne, N. B. (2016). Linking pattern completion in the hippocampus to predictive coding in visual cortex. *Nature Neuroscience* 2016 19:5, 19(5), 665–667. <https://doi.org/10.1038/nn.4284>

Mattfeld, A. T., & Stark, C. E. L. (2015). Functional contributions and interactions between the human hippocampus and subregions of the striatum during arbitrary associative learning and memory. *Hippocampus*, 25(8), 900–911. <https://doi.org/10.1002/HIPO.22411>

Federico, G., Osiurak, F., Ciccarelli, G., Ilardi, C. R., Cavaliere, C., Tramontano, L., Alfano, V., Migliaccio, M., Di Cecca, A., Salvatore, M., & Brandimonte, M. A. (2023). On the functional brain networks involved in tool-related action understanding. *Communications Biology* 2023 6:1, 6(1), 1–11. <https://doi.org/10.1038/s42003-023-05518-2>

Toni, I., Rushworth, M. F. S., & Passingham, R. E. (2001). Neural correlates of visuomotor associations spatial rules compared with arbitrary rules. *Experimental Brain Research*, 141(3), 359–369. <https://doi.org/10.1007/S002210100877>

Petrides, M. (1985). Deficits on conditional associative-learning tasks after frontal- and temporal-lobe lesions in man. *Neuropsychologia*, 23(5), 601–614. [https://doi.org/10.1016/0028-3932\(85\)90062-4](https://doi.org/10.1016/0028-3932(85)90062-4)